# High-resolution [14]C bomb-peak dating and climate response analyses of subseasonal stable isotope signals in wood of the African baobab – A case study from Oman

Franziska Slotta[1,2], Lukas Wacker[3], Frank Riedel[1], Karl-Uwe Heußner[4], Kai Hartmann[1], and Gerhard Helle[1,2]

[1] Freie Universität Berlin, Institute of Geolgical Sciences, Berlin, Germany
[2] GFZ German Research Centre for Geosciences, Section 4.3 Climate Dynamics and Landscape Evolution, Potsdam, Germany
[3] ETH Zürich, Laboratory of Ion Beam Physics, Zürich, Switzerland
[4] Deutsches Archäologisches Institut, Scientific Department of the Head Office, Berlin, Germany

*Correspondence to:* Gerhard Helle, ghelle@gfz-potsdam.de and Lukas Wacker, wacker@phys.ethz.ch

**Abstract.** The African baobab, *Adansonia digitata* L., has great paleoclimatological potential because of its wide distributional range and millennial length lifespan. However, dendroclimatological approaches are hampered by dating uncertainties due to its unique, parenchyma-dominated stem anatomy. Here, securely-dated time series of annual wood increment growth and intra-ring stable isotopes of carbon and oxygen of cellulose for a baobab tree from Oman covering 1941 to 2005 were established and tested for relationships to hydroclimate variability. Precise dating with the atomic bomb peak (ABP) using highly resolved [14]C measurements confirmed the annual character of the baobab's growth rings. F[14]C values of tree-ring cellulose were found up to 8.8 % lower than in the corresponding atmospheric $CO_2$ for the period around the ABP, that in conjunction with a considerable autocorrelation of the $\delta^{13}C$ series, points to the incorporation of previous year's carbon contributing to the average age of intra-ring wood samples. F[14]C of terminal parenchyma bands, marking the tree-ring boundaries, were found to be considerably younger than their corresponding tree ring indicating that parenchyma tissue is alive for many years probably undergoing cell division, structural reorganization and contributing to secondary growth. In contrast to the $\delta^{13}C$ time series, no significant autocorrelation was found in the $\delta^{18}O$ series of tree-ring cellulose despite the enormous water storage potential of this stem-succulent tree species. Year-to-year variability in tree-ring width and stable isotope ratios revealed radial stem growth and the geochemistry of wood cellulose are influenced by fluctuations in the hydroclimate. In particular, $\delta^{18}O$ was found to be a good climate proxy, followed by tree-ring width and $\delta^{13}C$. Tree-ring width and intra-ring $\delta^{18}O_{min}$ correlated well with each other and with precipitation amount for the period from pre-monsoon May to the end of the monsoon season in September/October. Intra-annual stable isotope courses were found rather similar for both $\delta^{13}C$ and $\delta^{18}O$. Years with particularly low monsoon rain were reflected by increased stable isotope values in the mid-section of intra-annual courses. Distinct patterns with low subseasonal isotope values seem indicative for years with heavy rainfall events from pre-monsoonal cyclones. Rain events from post-monsoonal cyclones may also be recorded, however, only two years of observation prevented a more conclusive evaluation.

## 1 Introduction

The African baobab, *Adansonia digitata* L., has received growing interest from the paleoclimatology community due to its widespread distribution and age longevity. As revealed by radiocarbon dating individuals can reach ages of 1,000 up to 2,500 years (Swart, 1963; Riedel et al., 2014; Patrut et al., 2017, 2018). Radiocarbon dating has not just helped to reveal baobabs to be the longest-living angiosperm tree species on earth, but also resolved the age of specific multi-stem tree structures of the biggest, monumental individuals in Africa (Patrut et al., 2015a,b, 2017, 2018). Deciduous *A. digitata* inhabits (semi-)arid areas

across the African continent and the south-eastern Arabian Peninsula (Wickens, 1979; Wickens and Lowe, 2008) and is considered an important archive for past rainfall information stored in the layers of its wood (Robertson et al., 2006; Slotta et al., 2017; Woodborne et al., 2015, 2016, 2018). Facing the potential impacts of climate change, *A. digitata* can, on the one hand, help in understanding past climate patterns in order to better model future climate scenarios across the African continent, a great challenge considering the current lack of high-resolution trans-regional proxy data (IPCC, 2007, 2013). On the other

hand, baobabs may be threatened by recent climate warming as the mortality of monumental baobabs in southern Africa appears to have increased over the last decade (Patrut et al., 2018), despite the high probability that the species has endured more severe climate conditions during earth history. In this respect conclusive evidence is lacking, and a better understanding of baobab ecophysiology and how they generally respond to temperature and rainfall variations, and to extremes like drought but also to heavy, cyclone-related rainfall events that can occur in parts of their distribution area (e.g. South Africa,

Mozambique or Oman) is required.

At a few baobab locations, tree-ring parameters such as stable isotopes ($\delta^{13}C$, $\delta^{18}O$) of wood cellulose and also tree-ring width were shown to correlate significantly with climate parameters (Robertson et al., 2006; Woodborne et al., 2015, 2016; Slotta et al., 2017). However, initial dendrochronological and subsequent dendroclimatological analyses are hampered by the baobab's complex stem anatomy (Fig. 2 A-D). A baobab tree ring generally consists of a diffuse porous wood section dominated by

woody fibers and vessels followed by a terminal parenchyma band (TPB). Since the stems are specialized for water storage, parenchyma cells make up 69-88 % of their total wood content (Chapotin et al., 2006a). Apart from TPBs, parenchyma cells form a sheath around associated vessels (paratracheal vasicentric), as well as in fine bands, clusters or rays (apotracheal) that are irregularly dispersed among fibers and lack an association with vessels (Fisher, 1981; Neumann et al., 2001; Rajput, 2004; Wickens and Lowe, 2008). Except from abundant parenchyma cells dispersed among wood fibers and vessels, the different

forms of baobab parenchyma are illustrated in the schematic drawing of Figure 2 D. The extraordinary high abundance of parenchymal tissue creates difficulties in distinguishing TPBs from other tangentially orientated non-annual parenchyma bands throughout the woody tissue, especially because the thickness of TPBs can vary greatly around the stem (Chapotin et al., 2006c; Slotta et al., 2017). Ring counting on baobabs of known age can result in about 98 % accuracy of the actual tree age (Guy, 1970), but even opposing radii of the same individual may differ in absolute numbers (Johansson, 1999). UV light

induced fluorescence was found to be very helpful in identifying the wood anatomical details of baobab trees (Slotta et al., 2017). Nevertheless, false and missing rings together with differences in tree age and site of origin have caused conventional cross-dating to fail (Slotta et al., 2017). In order to allow accurate comparison of baobab stable isotope data with meteorological data, highly resolved $^{14}C$ measurements around the atomic bomb peak (ABP) could be utilized to ensure proper identification, counting and dating of growth rings.

Radiocarbon dating of organic material has long been applied to obtain reliable age estimates of tropical and subtropical trees with unique and complex stem anatomy (Swart, 1963; Worbes, 1989). Parallel to the development of modern accelerator mass spectrometry (AMS) techniques the number of radiocarbon-aided dendrochronological studies on trees with indistinct growth boundaries has increased in recent years (Fichtler et al., 2003; Hua et al., 2003; Robertson et al., 2006; Andreu-Hayles et al., 2015; Ohashi et al., 2016; Baker et al., 2017; Linares et al., 2017; Witt et al., 2017). The vast majority of atomic bomb tests

have been carried out in the Northern Hemisphere, resulting in a delay in atmospheric mixing and five different zones separated by atmospheric cell boundaries (three in the Northern Hemisphere, two in the Southern Hemisphere) (Fig. 1A) (Hua et al., 2013). Despite an increasing number of studies that reconstruct the atmospheric radiocarbon content of the bomb peak period from tree rings (e.g. Hua et al., 2013; Turney et al., 2018), tree-ring radiocarbon studies in zones close to the equator (NH3, SH3) remain scarce and need to be refined. In this regard, our study contributes to the growing body of tree-ring radiocarbon

records from these zones, however, because of the peculiar stem anatomy and its high content of parenchyma, long-lived baobab trees may not be the best canditates for high precision reconstruction of the atmospheric radiocarbon content.

The baobab site selected for this study is in the Dhofar region of Oman and is located along the north-eastern edge of the species natural distribution (Wickens and Lowe, 2008). This follows a common approach in dendroecology and -climatology to select sites close to the ecological limit of a species to ensure a clearer signal of the limiting environmental and climatic

factors (Fritts, 1976; Smith, 2008). The woodlands and forests along the Dhofar mountains are a paleo-African relict of a moist vegetation belt that once stretched beyond the Southern Arabian Peninsula into Asia (Kürschner et al., 2004; Hildebrandt and Eltahir, 2006). Upwelling along the southern coast of Oman lowers the Arabian Sea's surface temperature and consequently cools the moisture rich air masses of the southwest monsoon winds. Once the dew point is reached, fog and low clouds form and then drift inshore towards the mountain range (Kürschner et al., 2004). Compared to other baobab populations, this site is

therefore characterized by mostly horizontal precipitation (i.e., cloud water interception within the canopy; Hildebrandt and Eltahir, 2006). Nevertheless, precipitation seasonality within a calendar year is defined by a single wet period and the deciduous habit of the species in general should favor the formation of annual tree rings.

Where tree rings are anatomically non-distinct, intra-annual analysis of stable isotopes ($\delta^{13}C$, $\delta^{18}O$) can help to identify annual growth increment provided that a general seasonally recurring pattern prevails in the intra-annual stable isotope records (e.g.

Poussart et al., 2004; Anchukaitis et al., 2008; Pons and Helle, 2011; Xu et al., 2014; Ohashi et al., 2016). Hence, we wanted to test for recurring patterns in the intra-annual stable isotope data from a well-dated baobab tree-ring record consolidated by radiocarbon dating. After discering intra-annual trends, or referred to in this study as isotope courses, we aimed at establishing the relationship between the inter-annual variability of stable isotope parameters, minimum, maximum and mean values with

monthly climate data. Usually, tree-ring stable isotopes show a high sensitivity to external drivers, such as weather conditions,

making them a valuable tool to reconstruct the environmental and climatic influences during tree growth (e.g. Fichtler et al., 2010; Hartl-Meier et al., 2015; Szejner et al., 2016; van der Sleen et al., 2017). The climate signature of tree-ring width and $\delta^{13}C$ basically originates from internal climate response mechanisms, driven by species-specific physiology, as well as carbon isotope fractionations during photosynthesis at leaf level. Elevated $\delta^{13}C$ values in tree-rings are mainly caused by a reduction in stomatal conductance (as a result of drought stress) and/or an increase in photosynthetic activity (Scheidegger et al., 2000).

In both cases the leaf internal $CO_2$ concentration declines and leads to a reduced enzymatic discrimination against $^{13}C$ (Farquhar et al., 1982; Farquhar and Cernusak, 2012). However, since increased photosynthetic activity is unlikely under drought stress, high $\delta^{13}C$ values are typically interpreted as a drought signal in areas where growth is limited by moisture availability (Brienen et al., 2011; Schollaen et al., 2013; Woodborne et al., 2015, 2016). Following this approach, the higher the $\delta^{13}C$ values become, the more severe is the drought episode. $\delta^{18}O$ of tree-ring cellulose is first related to the $\delta^{18}O$ of the precipitation source via soil

water and then later by internal processes. $\delta^{18}O$ of soil water constitutes the $\delta^{18}O$ input to the arboreal system and usually represents an average $\delta^{18}O$ over several precipitation events modified by partial evaporation from the soil (depending on soil texture and porosity) and by a possible time lag, depending on rooting depth (Sprenger et al., 2017; Treydte, et al. 2014). $\delta^{18}O$ of tree-ring cellulose is dependent on two internal processes: evaporative $^{18}O$-enrichment of leaf water via transpiration and leaf-to-air vapor pressure deficit (VPD), as well as biochemical fractionations and isotopic exchange of $^{18}O$ with trunk water

during cellulose biosynthesis (Roden et al., 2000; Kahmen et al., 2011; Treydte et al., 2014 and citations therein). In this regard, $\delta^{18}O$ in tree rings from (sub-)tropical timber has been proven to strongly represent the isotopic composition of rain, i.e. source water (e.g. Evans and Schrag, 2004; Brienen et al., 2012, 2013; Boysen et al., 2014; Baker et al., 2015; Baker et al., 2016; van der Sleen et al., 2015, 2017). Variation of $\delta^{18}O$ in rainfall is determined by several factors where the amount of rain (Araguas et al., 1996) plays a key role producing a strong inverse correlation. Apart from isotopic composition of the source water,

drought conditions with increasing VPD can lead to increasingly unfavorable growth conditions (with reduced tree-ring widths) paralleled by high $\delta^{18}O$ values of corresponding tree ring cellulose (e.g. Treydte et al., 2014).

We performed highly resolved inter- and intra-annual radiocarbon and stable isotope measurements on tree-ring cellulose of a baobab tree from Oman in order to (1) to produce a time series of tree-ring width and intra-annual stable isotope ratios ($\delta^{13}C$, $\delta^{18}O$), and (2) to compare $^{14}C$ data of baobab tree-ring cellulose with published reconstructions of atmospheric $^{14}C$ data ($F^{14}C$)

between the African continent and India, primarily aiming at testing the annual nature of ring formation in baobab trees and securing the accuracy of ring-width and stable isotope time series. The accurate time series of tree-ring width index (RWI), minimum ($\delta^{13}C_{min}$, $\delta^{18}O_{min}$), maximum ($\delta^{13}C_{max}$, $\delta^{18}O_{max}$) and mean ($\delta^{13}C_{mean}$, $\delta^{18}O_{mean}$) stable isotope values were then (3) compared to instrumental climate data for assessing their sensitivity to monthly hydroclimate variables that included the comparison of intra-ring stable isotope courses during dry years, as well as to cyclone-related pre- and post-monsoonal heavy

rainfall events.

## 2 Materials and Methods

### 2.1 Study site and climate

The baobab tree analyzed for this study is one of more than 60 individuals growing in Wadi Hinna (17°03' N, 54° 36' E, 318 meters above sea level (m asl)) at the edge of the Dhofar Mountains in southern Oman (Fig. 1B, D). The Dhofar Mountains reach elevations of about 1,450 m asl and are separated from the Arabian Sea by a narrow coastal plain. The prevailing wind system is defined by four different seasons (Walters and Sjoberg, 1990; Charabi et al., 2011): (1) winter (December-March), in which the region is dominated by the northeast monsoon, (2) the spring transition (April-May), during which the northeasterly circulation breaks down while south-westerly trade winds enter the region; (3) summer (mid June - mid September), in which damp air masses of the southwest monsoon (also *khareef*) encounter the mountain range and bring dense fog and light rain (Kwarteng et al., 2009), and (4) the autumn transition (October-November), during which the northeasterly flow is re-established.

The closest meteorological station to the study site is located at Salalah airport (17° 01' N, 54° 04' E, 20 m asl), 57 km west from Wadi Hinna. Due to the distance and altitudinal difference between station and study site, the use of Climate Research Unit (CRU TS4.01; modified after Harris et al. 2013) interpolated grid data was considered more appropriate for defining the monthly climatology, and for climate response analysis. The mean annual temperature for the reference period 1945-2013 was 23.1 °C. Precipitation data (also CRU TS4.01) and the Global Precipitation Climatology Centre (GPCC V7_05, (Schneider et al., 2015)) covering the study site (17.0-17.5° N, 54.0-54.5° S) was tested for statistical similarity against data from the Salalah station (1945-2013) in order to confirm which data set best represents the temporal variability of the study region. GPCC data showed the highest correlation ($r_{GPCC} = 0.91$, $p < 0.0001$; $r_{CRU} = 0.70$, $p < 0.0001$) and was therefore used in further analysis. The grid cell indicated an average annual precipitation of 130 mm with large interannual variation (range: 28-534 mm; 1945-2013). The light drizzle during southwest monsoon in summer (mid June – mid September) accounts for about 70 % (= 90 mm, range: 21-231 mm) of the average annual rainfall. Apart from the monsoon rain, precipitation is rare and erratic, originating either from low pressure systems moving across Oman or from cyclonic systems moving westward from the Arabian Sea during the spring (May - June) and autumn (October - November) transitions (Clark et al., 1987). These tropical storms with heavy rainfall occur about once every three years (Pedgley, 1969; Hildebrandt and Eltahir, 2006; Strauch et al., 2014). Aside from intensity and duration, rainfall from the monsoon and cyclones usually differ distinctively in its respective isotopic signature (Clark et al., 1987; Strauch et al., 2014; Wushiki, 1991) (Fig. 1C, Table 1). Monsoon precipitation from the Dhofar mountains generally correlates with the oxygen isotopic composition of the Indian monsoon rainfall along the southeast coast of Oman (Strauch et al., 2014) shows marginal variation, between -0.5 ‰ and +0.7 ‰, with values close to the mean $\delta^{18}O$ of 0.61 ± 0.02 ‰ reported for the Arabian Sea water (Lambs et al., 2011). Minimal difference exists in the oxygen isotopic signature of monsoon rain, fog and interception waters (Table 1). In contrast, water samples collected in the Dhofar mountains from cyclones that sometimes bring enormous quantities of rainfall (e.g., one or more years of rainfall amount in a single day) show highly variable but significantly low $\delta^{18}O$ signatures down to -11 ‰ (Dansgaard, 1964).

## 2.2 Field work

The sampling took place in early April 2015 when the trees were still without leaves. Ten baobabs were sampled in total. Increment cores of 12 mm in diameter and up to 80 cm in length were taken at breast height from four different orientations (NE, SE, SW, and NW). While in the field, core samples were stored in an airtight and cool container, and later stored in a freezer to prevent mold growth and distortion of wood anatomical features that could be caused by the dehydration of parenchyma tissue.

**2.3. Sample preparation and wood increment dissection**

The baobab tree showed rather wide growth rings favorable for intra-annual analysis, possibly due its immediate proximity and interaction to an ephemeral stream. The SW oriented core was chosen because its parenchyma bands were very distinct and the perpendicular orientation of vessels and fibers with respect to the long axis of the core allowed for precise subsampling. The core surface was prepared with a razor blade to expose the anatomical features of each tree ring and photographs were

taken with a fluorescence microscope (Nikon MULTIZOOM AZ100M, V-2A: 380-420 nm; camera head: Nikon digital sight DS-Fi1c; light source: C-HGFI HG Precenteres Fiber Illumiator (130 W mercury lamp); program: NIS Elements 4.30.01© 1991-2014 Laboratory Imaging). Individual micro-pictures were assembled to give one complete stitched panorama picture of the wood core, that was printed for further sampling documentation (scale 1:7.6) and provided the basis for the tree-ring width measurements (Fig. 2A).

For radiocarbon and stable isotope analyses, tangential wood slices of approximately 1 mm were cut parallel to the fiber orientation and in radial direction from the cambial zone towards the pith (Fig 2B, C, D). Since the tree ring structure for the outermost wood section was often hard to interpret (Johansson, 1999; Slotta et al., 2017), we did a five-point screening across a region of the core that based on our ring-counting was estimated to be the location of the bomb peak. The entire bomb peak and the decade before was dissected amounting to a total of 360 samples with resolutions between 1 and 11 samples per year.

Because parenchyma cells can undergo cell division within the stem independent from the vascular cambium (Chapotin et al., 2006a; Spicer, 2014), rays and other parenchymatous structures (see Fig. 2D for illustration) that may be carrying a different, non-contemporaneous radiocarbon and stable carbon isotope signal than the surrounding wood vessels and fibers were removed from the wood slices to avoid possible contamination. A potential fraction of parenchyma cells in between vessels and fibers was considered being small of higher order. Two thirds of each wood slice were separated for radiocarbon dating

while the remaining third was kept for stable isotope analyses (Fig. 2C). Each terminal parenchyma band (TPBs) was also retained and used for radiocarbon analysis.

## 2.4 Radiocarbon dating

The samples were prepared for radiocarbon analysis at ETH Zurich according to a protocol for high-precision measurements
(Sookdeo et al., 2019). Oxalic acid II (OXII, NIST SRM 4990C) was used for standard normalization. Internal wood reference materials from AD 1515 (pine from Switzerland) and two different radiocarbon free wood blanks (New Zealand Kauri and lignite from Reichwalde Germany) were repetitively analyzed together with the annual wood samples. The wood blanks were used for blank subtraction in the data evaluation process and the AD 1515 references were used for quality control only (Brehm et al., 2021). In total wood 110 samples were extracted for holocellulose with a base-acid-base-acid-bleaching procedure
(Němec et al., 2010). Ten samples were further purified to alpha-cellulose with an additional base treatment (17.5 % NaOH for 2 h at room temperature) followed by washing and freeze-drying. The cellulose samples were then graphitized using an Automated Graphitization Equipment (AGE III) coupled to an elemental analyzer (EA: Vario MICRO cube; Wacker et al., 2010b). Once graphitized, the samples were immediately pressed into targets to minimize their interaction with air and their $^{14}$C content was measured with a MICADAS mass spectrometer (Wacker et al., 2010a). Since the tree ring structure for the
outermost wood section was often hard to interpret (Johansson, 1999; Slotta et al., 2017), we did a five-point screening for the bomb peak before samples were selected from each of the clearly-identifiable growth rings onwards from what we thought to be 1942. We avoided developing a longer record by including older tree rings because of a lack of reliable precipitation data beyond this date that could be used in calibration. With intention to minimize possible carry-over effects from the implementation of previous years' non-structural carbohydrates (NSC) into the current year's wood cellulose, the samples for
dating were selected from the last third of each growth structure while steering clear of the TPB by at least one sample, where possible (Fig. 2B). For comparison, cellulose samples of five individual TPBs were analyzed for 1960-1963 (n = 4) and 2005. For the period between 2005 and 2015, for which by visual inspection annual rings could not be identified, 19 samples were dissected of which 17 samples were dated. In addition, intra-ring radiocarbon analysis was performed on the tree rings of 1962 (6 samples) and 1963 (8 samples).
Comparison of $^{14}$C of the assumed tree rings with atmospheric $^{14}$C data comprising the bomb peak values (see different zones in Fig. 1A) allowed for annual dating between 1956 and 1980; before and after these dates the differentiation between the atmospheric $^{14}$C values of consecutive years is not sufficient enough to securely designate the age of wood increments (see Hua et al., 2003). The normalized $^{14}$C activity ratio (F$^{14}$C; Reimer et al., 2004) of the assumed tree rings was plotted against and correlated peak-to-peak (± 5 years) with the calibration curve for the corresponding hemispheric zone NH3 (Hua et al.,
2013). Because of an observed offset in the data sets, the calibration curves for SH3 and SH1-2 (Hua et al., 2013) were also included in the comparison (Fig. 3). Each intra-ring baobab F$^{14}$C sample position on the time axis is relative to their position within the respective tree ring of a growing season reflecting approximately June until September. In case of suspected missing rings, the wood anatomy was reviewed and additional samples were radiocarbon dated. No properly dated tree-ring width chronology (e.g. from *J. excelsa* (Sass-Klaassen et al., 2008)) exists from the study region that could be used as a reference
was available for comparison. Calendar ages were calculated from the F$^{14}$C values of the TPBs using the program OxCal

v4.3.2 (Bronk Ramsey, 2017) and the post-bomb atmospheric NH3 and SH1-2 curves (Hua et al., 2013). Furthermore, we visually and statistically cross-dated (COFECHA, Holmes, (1983)) time series of ring width and stable isotopes with the seasonally and annually resolved precipitation amount (May-Sep, Jun-Sep, and Jan-Dec) for the grid cell 17-17.5° N, 54-54.5° E (GPCC V7.05, Schneider et al. 2015).

**2.5 Stable isotope analysis**

For the stable isotope analyses ($\delta^{13}C$, $\delta^{18}O$), all 360 samples (excluding TPBs) were holocellulose extracted using the devices and procedures described by Wieloch et al. (2011), homogenized by ultrasonic treatment (Laumer et al., 2009) and freeze-dried for at least 48 h. A thorough description of the background and details on the methodology is given by (Helle et al., 2021). Prior to mass spectrometric analysis, 130-180 µg of cellulose and reference material (IAEA CH-3, CH-6 and Merck cellulose), respectively, were packed in silver capsules and stored over night at 100 °C in a vacuum drying oven. Simultaneous dual measurements of $\delta^{13}C$ and $\delta^{18}O$ were performed on carbon monoxide derived from high temperature TC/EA pyrolysis (1400 °C) coupled online via a Conflo IV to an IRMS Delta V Plus (Thermo Fisher Scientific, Bremen, Germany). Values are referred to VPDB and VSMOW, respectively with a reproducibility of $\leq 0.15‰$ for $\delta^{13}C$ and $\leq 0.25‰$ for $\delta^{18}O$.

In order to use $\delta^{13}C$ of baobab tree-ring cellulose ($\delta^{13}C_{raw}$) as a measure for plant response to weather conditions, the $\delta^{13}C_{raw}$ values have been corrected ($\delta^{13}C*$) for the anthropogenic changes in the long-term trend of the atmospheric $CO_2$ source. Rising atmospheric $CO_2$ concentrations ($pCO_2$) and correspondingly declining $\delta^{13}C$ of atmospheric $CO_2$ ($\delta^{13}CO_2$) prevailing since the onset of industrialization (1850 AD) were addressed by subtracting for each tree-ring stable isotope value the annual changes in $\delta^{13}CO_2$, obtained from direct measurements (White et al., 2015). Since trees respond to increased atmospheric $pCO_2$ with higher discrimination against $^{13}C$, an additional moderate correction of 0.073‰ per 10 ppm change (Kürschner, 1996 ) was applied producing a corrected $\delta^{13}C*$ record.

The intra-annual isotopic signatures were plotted against their relative position within the tree ring normalized to 100 % of the respective tree-ring width. Annual tree-ring stable isotope values were obtained from averaging intra-ring $\delta^{13}C*$ and $\delta^{18}O$ values.

**2.6 Statistical analysis**

Ring widths were measured for the entire core and detrended using an exponential fit (R program, package dplR; Bunn et al., 2018) resulting in a ring-width index chronology (RWI). Due to different intra-annual resolution of the stable isotope time series (ranging from 2 to 11 samples per year) the mean annual signature might be blurred and thus may weaken the climate response of intra-ring isotopic records. Hence, besides mean isotopic signatures ($\delta^{13}C*_{mean}$, $\delta^{18}O_{mean}$), we also tested minimum ($\delta^{13}C*_{min}$, $\delta^{18}O_{min}$) and maximum ($\delta^{13}C*_{max}$, $\delta^{18}O_{max}$) stable isotope values for their climate sensitivity.

The obtained time series (RWI, $\delta^{13}C*_{min/max/mean}$, $\delta^{18}O_{min/max/mean}$) were tested for autocorrelation. Moving mean, variance and correlation tests (window size = 15 years) were performed to discretize the timeseries in terms of central tendency, variation

and trend behavior. The response of these parameters to weather conditions was then evaluated by calculating the Pearson correlation with monthly and seasonally averaged data of precipitation (GPCC V7.05, Schneider et al. 2015), cloud cover, temperature ($T_{min/max/mean}$), and vapor pressure (CRU TS4.01) for the grid cell 17-17.5° N, 54-54.5° E, as well as relative humidity (CRU_blendnewjul08_RH_7303cf) for the area covering 15-20° N, 20-25° E. The impact of drought on the wood parameters was estimated using the Standardized Precipitation Evapotranspiration Index (SPEI) for the grid cell 17-17.5° N, 54-54.5° E. SPEI assesses droughts by using both precipitation and temperature variability at different timescales (Vicente-Serrano et al., 2010a; 2010b; 2015). Where applicable, the coefficient of determination (COD = $r^2 \cdot 100$ %) has been added for explanatory meaning.

## 3 Results

### 3.1 Wood anatomy

The TPBs were generally well defined in this ca. 100-year old tree, with a mean width of 1.5 mm (range: 0.5-2.4 mm). The only exceptions were the outermost 15.3 mm of wood and three rings with TPB widths ≤ 0.1 mm (i.e. 1973, 1981, 1982) which were finally confirmed by the radiocarbon dating. In general, the width of the TPBs was significantly correlated to the width of the corresponding tree ring ($r_0 = 0.45$, $COD_0 = 20.3$ %, $p < 0.001$), as well as the rings that were built 1 to 4 years earlier (highest correlation: $r_{-4} = 0.36$, $COD_{-4} = 13.0$ %, $p = 0.005$) and those built 1-2, as well as 6 years later (highest correlation: $r_{+1} = 0.47$, $COD_{+1} = 22.0$ %, $p < 0.001$).

The mean tree-ring width for the analyzed period (TRW measured without TPBs and excluding missing rings) was $4.8 \pm 2.2$ mm with a slightly significant negative trend over time ($r = -0.32$, COD = 10.2 %, $p = 0.011$). After detrending, significant autocorrelations were observed for lags 1-3 and lag 6. The moving mean test revealed major changes in the central tendency of the RWI time series for the periods 1962-1965, 1976-1977, and 1985-1992. No major changes were revealed in the variance homogeneity and linear dependency.

### 3.1 Radiocarbon measurements and dating

#### 3.1.1 Holocellulose vs. alpha cellulose

The values of six out of ten holocellulose and alpha cellulose pairs were equal within two standard deviations, and nine out of ten within three standard deviations. Based on these indications it was not necessary to perform an alpha-cellulose extraction procedure for this species.

#### 3.1.2 Inter-annual $F^{14}C$ variability

The $F^{14}C$ values of the assumed tree rings (excluding the younger, outer stem without clearly identifiable TPBs) fit the overall shape of the bomb peak curve relative well, thus demonstrating their annual character (Fig. 3). Anomalies in the $F^{14}C$ record

led to a review of the wood anatomy and consequently the identification of three more tree rings (boundaries marked by dashed lines in Fig. 2), while the year 1975 was regarded as absent from our time series. Our interpretation of the intra-annual $F^{14}C$ data was confirmed by visual and statistical comparison of the TRW chronology with precipitation data. Comparison with precipitation data also indicated missing rings in 1951 and 1983, that could not be confirmed with year-to-year variability of atmospheric $F^{14}C$ data was not sufficient for unambiguous detection. The last identifiable TPB corresponds to the tree ring of 2005, thus, the outermost 15.3 mm of wood represented nine years of growth (2006-2014) that could not be distinguished. Due to the gentle slope of the calibration curve during this period, we refrained from assigning the measured $F^{14}C$ values to a specific year. Hence, the final annually dated baobab time series investigated further covers the period from 1941 till 2005.

The highest correlation between the revised baobab time series and calibration curves from different hemispheric zones (Fig. 3A, B) was reached for a peak-to-peak fit with SH1-2, although the study site is located in zone NH3 (Fig. 1A) (r = 0.996, $p < 0.0001$; SH3: r = 0.995, $p < 0.0001$; NH3: r = 0.989, $p < 0.0001$, all CODs > 97.8 %). For the peak period (1964-1966) baobab $F^{14}C$ values were found notably lower than the mean values of the zone NH3 ($\approx$ -8.8 %) and even slightly lower than the mean values of SH1-2 ($\approx$ -2.6 %) (Fig. 3A, B).

### 3.1.3 Intra-annual $F^{14}C$ variability

Intra-annual baobab $F^{14}C$ values analyzed for the two consecutive years, 1962 and 1963, were found positioned between higher NH3 and lower SH1-2 atmospheric values (Fig. 3B). The $F^{14}C$ values for the TPBs ($F^{14}C_{TPB}$) for 1960-1962 (n = 3) are significantly higher than the (mean) values for the corresponding tree rings (Fig. 3, Table 2) and also higher than expected for NH3. For 1963, the $F^{14}C_{TPB}$ value is in the range of the tree ring's value, and the last clearly visible TPB (2005) shows a slightly lower $F^{14}C$ value than the corresponding tree ring. Based on radiocarbon analysis TPBs appear to be on average younger than the fibers and vessels of their corresponding tree ring (Table 2).

### 3.2 Statistical characteristics of tree-ring parameter time series

Despite the corrections made for atmospheric change in $\delta^{13}C$ and pCO$_2$, the $\delta^{13}C*$ time series shows a slightly negative trend over time (r = -0.32, COD = 10.2 %, $p = 0.011$) (Fig. 4). The mean $\delta^{13}C*$ value was -26.99 $\pm$ 1.21 ‰ (min: -29.77 ‰, max: -23.43 ‰) with an observed mean range per year of 2.27 ‰ (maximum range: 4.55 ‰ in 1963). $\delta^{13}C*_{max}$ correlated with RWI (r = 0.31, COD = 9.6 %, $p = 0.014$) and strongly with $\delta^{18}O_{max}$ (r = 0.61, COD = 37.2 %, $p < 0.0001$). Significant autocorrelation was found for $\delta^{13}C*_{max}$ (lag 3) and for $\delta^{13}C*_{mean}$ (lag 1-2 and $\delta^{13}C*_{min}$ (lag 1). The moving tests revealed major changes in the central tendency of $\delta^{13}C*_{min}$ for the periods 1948-1953, 1960-1961, and 1984-1985. Major changes in the linear dependency of $\delta^{13}C*_{max}$ occurred in 1964-1966 and 1984-1985 and its variance homogeneity changed significantly for 1985-1987.

The mean $\delta^{18}O$ value was 30.44 $\pm$ 0.77 ‰ (min: 27.77 ‰, max: 32.67 ‰) with an observed mean range per year of 1.38 ‰ (maximum range: 3.16 ‰ in 2002). A rather strong negative correlation was found between $\delta^{18}O_{min}$ and RWI (r = -0.60,

$p < 0.0001$). No significant autocorrelation was found for any of the $\delta^{18}O$ time series. Major changes in the central tendency occurred for $\delta^{18}O_{min}$ in 1984-1987, and for $\delta^{18}O_{max}$ in 1968-1969. The variance homogeneity of $\delta^{18}O_{min}$ changed significantly for 1957-1959, 1976-1977, 1985-1988, and 1991-1993; the variance homogeneity of $\delta^{18}O_{max}$ changed significantly for 1959-1961.

### 3.3 Intra-annual stable isotope courses

The intra-annual signature was found similar for both $\delta^{13}C^*$ and $\delta^{18}O$ (all years with > 1 value; n = 59; Fig. 5A). The first half of the tree ring was characterized by a majority of minimum values per year, whereas the second half comprised mainly maximum values. This separation was clearer for $\delta^{13}C^*$ where 86 % of all $\delta^{13}C^*_{min}$ occurred within the first half (64 % within the first quarter) and 80 % of all $\delta^{13}C^*_{max}$ were found in the second half of the tree ring (58 % within the last quarter). $\delta^{18}O$ showed a generally higher variability, with the minimum and maximum values more scattered throughout the year. Only 68 % of the $\delta^{18}O_{min}$ were found within the first half of a tree ring (44 % within the first quarter) and 71 % of the $\delta^{18}O_{max}$ occurred within the second half (44 % within the last quarter). The mid-section showed the highest variation for both $\delta^{13}C^*$ and $\delta^{18}O$, the values at the very beginning and the very end varied less. Due to different resolutions of the intra-annual time series ranging from 2 to 11 samples per year, the actual intra-annual isotope courses might be blurred. The time series with at least 7 samples per year (n = 20) resulted in a distinct intra-ring pattern (Fig. 5B). For $\delta^{18}O$, the values were still more scattered than for $\delta^{13}C^*$ with 75 % of $\delta^{18}O_{min}$ occurring in the first half (35 % within the first quarter) and 85 % of $\delta^{18}O_{max}$ occurring in the second half of the tree ring (55 % within the last quarter). The center of the tree ring revealed a short-term dip in $\delta^{18}O$ before rising towards the end of the tree ring, which is in contrast to $\delta^{13}C^*$ that showed slightly lowered, but not declining values in the mid-section (Fig. 5B).

### 3.3.1 Intra-annual isotope course in years with heavy rainfall events due to spring or autumn cyclones

In order to estimate the influence of heavy rainfall events on intra-annual $\delta^{13}C^*$ and $\delta^{18}O$, we separated years with pre- to early-, and late- to post-monsoonal storms (starting in Apr/May/Jun or Sep/Oct/Nov, respectively) $\geq 90$ mm (i.e. average monsoon precipitation) with > 1 or $\geq 7$ observations, respectively (Fig. 5A, B). As shown in Figure 5C this revealed a pronounced difference in the mid-section of the intra-ring isotope courses, causing a significant decline. In the five years with spring cyclones (1959, 1963, 1977, 1989, 2002), $\delta^{13}C^*$ and $\delta^{18}O$ values declined during the first half of a tree ring and steeply increased within the second half, reaching their maximum close to or at the end of each tree ring. Although $\delta^{13}C^*$ and $\delta^{18}O$ courses closely resemble each other, $\delta^{18}O$ showed a more pronounced decline in the mid-section and a greater range, with maximum values found in two out of the five years at the very beginning of a tree ring (Fig. 5C). This indicates that minimum values are shifted from the beginning of an intra-ring isotope course to its center in years with spring cyclones and rainfall $\geq 90$ mm.

In the two years with autumn cyclones (Fig. 6A) intra-annual $\delta^{13}C^*$ showed no distinct pattern and after reaching the maximum values indicated a final decrease at the end of a tree ring, but showed no distinct pattern. $\delta^{18}O$ responds largely

similar to autumn cyclones with the lowest values of the intra-ring course in the last quarter of the tree ring. Additionally, $\delta^{18}O$ values were found above normalized average in the first half of the intra-ring $\delta^{18}O$ course, and below average at the end (Fig. 6A, right). The tree ring of 1948 showed a conspicuously strong increase in $\delta^{18}O$ towards the very end. It was rather wide and is represented by 8 intra-annual isotope samples.

### 3.3.2 Intra-annual isotope course in years with exceptionally low monsoon precipitation

Years with less than 50 % of the average monsoon precipitation (Jun-Sep: $\leq 45$ mm) and no spring/autumn heavy rainfall due to cyclones were selected for inspecting the influence of drought on the intra-ring stable isotope courses yielding six years with reduced average monsoon precipitation, 1968, 1972, 1974, 2000, 2001 and 2003. Except for the year 2000, all intra-ring $\delta^{13}C^*$ courses start with a distinct increase reaching $\delta^{13}C^*_{max}$ or values close to $\delta^{13}C^*_{max}$ within the first half of the tree ring (Fig. 6B). In general, intra-ring $\delta^{13}C^*$ for the years 1968, 1972 and 2003 showed an increasing trend with $\delta^{13}C^*_{max}$ reached towards the end of the ring, whereas in 1974, 2000 and 2001, $\delta^{13}C^*_{max}$ occurred in the early half of the ring, followed by declining $\delta^{13}C^*$ (with the exception of 2000 that showed a complex multimodal pattern). The intra-annual course for the year 2000 showed a much higher intra-ring stable isotope variation, but it was also the year with the most observations ($= 6$) for a year with low monsoon rain. In sum, both isotope species generally showed rather similar intra-ring trends in response to drought with highest values frequently located in the mid-section of the intra-ring course (Fig. 6B).

### 3.4 Statistical relations between weather conditions and tree-ring parameters

Significant correlations between RWI, $\delta^{13}C^*_{min/mean/max}$, and $\delta^{18}O_{min/mean/max}$ and selected monthly climate variables reflecting moisture conditions at the site were observed for the current and the previous year (Table 3, Fig. 7). RWI, comprising annual radial growth, showed significant positive correlations ($p < 0.001$) with August and May-September precipitation (Table 3, Fig. 7A). $\delta^{13}C^*_{mean}$ generally revealed no relation to precipitation and only weak correlations with other climate variables. In contrast, $\delta^{18}O_{mean}$ correlated significantly and negatively with May precipitation, as well as the monthly aggregates, May-September and January-December.

Intra-ring minimum isotope values displayed only weak correlations between $\delta^{13}C^*_{min}$ and precipitation from previous year January and current year November, while no relations were found with other climate variables. Conversely, $\delta^{18}O_{min}$ correlated significantly with $T_{max}$ and all moisture related climate variables with $\delta^{18}O_{min}$ showing a strong response to pre-monsoonal May rainfall (Table 3, Fig. 7B).

$\delta^{13}C^*_{max}$ showed more and better relationships with climate quantities than $\delta^{13}C^*_{mean}$ and $\delta^{13}C^*_{min}$. Significant correlations were found for May temperature, vapor pressure, and SPEI (1 month). The highest correlation to cloud cover was revealed for $\delta^{18}O_{max}$ in pre-monsoon April (Table 3, Fig. 7C). Further significant correlations of $\delta^{18}O_{max}$ were also found with the climate conditions of the previous year (previous Jan-Dec, Feb, Oct and Nov; c.f. Table 3 for details).

## 4 Discussion

### 4.1 $F^{14}C$ values in baobab wood are very likely altered by carbohydrate metabolism

Situated in Oman, the baobab $F^{14}C$ values were expected to fit the post-bomb atmospheric curve NH3 (Hua et al., 2013). However, they were found notably lower than NH3 and even SH3, around the bomb peak period (1964-1967, Fig. 3A). A number of external factors can potentially be responsible for the dilution of atmospheric $^{14}C$: 1) fossil fuel dilution from neighboring industrial and urban areas (Chakraborty et al., 2008; Chakraborty et al., 1994), 2) volcanic emanations (e.g. Lefevre et al., 2018), and 3) upwelling of carbon rich waters causing outgassing of 'older' $CO_2$ into the atmosphere (Takahashi et al., 2009; Sreeush et al., 2018). We can exclude atmospheric $^{14}C$ dilution from $CO_2$ released by volcanic activities. This effect is usually confined to obvious volcanic activity, e.g., fumaroles. Furthermore, we consider regional fossil fuel $CO_2$ mixing as unlikely at this tree site in the Dhofar mountains. Because of its remoteness to any city or industrial complex the site rather reflects atmospheric background. Due to the close proximity to the coast of the Arabian Sea with significant upwelling we cannot fully exclude atmospheric $^{14}C$ dilution. However, other studies report only subtle effects of aged $CO_2$ released from ocean upwelling (e.g. Beramendi-Orosco et al., 2018). Since the measurements for the 1940s are consistent with other records for the northern hemisphere, we suggest that dilution from $^{14}C$-free $CO_2$ from upwelling cannot be responsible for the large offset of 8.8% (1964-1967) between $F^{14}C$ of baobab tree-ring cellulose record and atmospheric $F^{14}C$ of NH3.

The baobab record of the bomb peak is further characterized as a period of major changes in the central tendency of RWI (1962-1965) as well as in the linear dependency of $\delta^{13}C^*_{max}$ (1964-1966). It is difficult to assess the driving force(s) behind these major changes without further investigations on other tree species and other sites in Oman, however, the notably lowered values of baobab $F^{14}C$ during the bomb peak period may to some extent be explained by the baobabs' carbohydrate turnover. The observed autocorrelation in the $\delta^{13}C^*$ signature points to the use of previous years' carbon for part of the growth, which leads to a dampening in the carbon isotope record of environmental signals, including the atmospheric $F^{14}C$ content. The dramatic deviation of baobab $F^{14}C$ values and atmospheric NH3 data around the bomb peak area points out that an immediate allocation of leaf assimilates to wood growth is very unlikely and transient storage or carbon recycling from respiration has to be considered.

In an attempt to explain the observed baobab $F^{14}C$ values, we applied a simple conceptual modeling approach assuming that baobab wood formation is fueled by pools of non-structural carbon (NSC) that carry the atmospheric $F^{14}C$ source values for the NH3 zone. From radiocarbon dating, Richardson et al. (2013) found a mean age of stemwood NSC of about 10 years in temperate forest trees with individual dates of up to 31 years. They presented lines of evidence for 'fast' and 'slow' cycling reserves and proposed a two-pool model structure to explain the age of stemwood total NSC. A follow-up study on the mixing of new and previously stored NSC supported this model, and the 'last in, first out' hypothesis (Richardson et al., 2015). We have no $^{14}C$ data or other information about the age of different NSC pools in baobabs, however, we attempted to address these considerations by assuming two NSC pools, and tested for annual renewal rates ranging between 2 and 10% ('slow' cycling reserves) and 98 to 90% (low contribution of old reserves; 'fast' cycling), respectively. Taking the 'last in, first out' hypothesis

(Richardson et al., 2015) into account we further presumed that wood formation is fueled by a mix from these two NSC pools and visually tested the agreement with the observed baobab $F^{14}C$ values for relative contributions between 95% and 70% from the `fast` cycling pool, and 5 to 30% from the `slow` cycling NSC pool. Best approximation to the baobab $F^{14}C$ time series was achieved by holding stakes of 85% `fast` cycling and 15% 'slow' cycling NSCs with annual renewal rates of 95% (renewal `fast` cycling pool) and 5% (renewal `slow` cycling pool) of the two respective pools (Fig. 8). The `slow` cycling NSC pool may predominantly represent starch the major long-term carbohydrate reserve in trees, although starch has not always been found older than sugars which should represent the `fast` cycling NSC pool (Richardson et al. 2015). We suggest that the `slow` cycling NSC might originate from considerable cortical photosynthesis (Kotina et al., 2017) of older respiratory $CO_2$. Additionally, the significant autocorrelation found for $\delta^{13}C^*$ strongly indicates the incorporation of previous years' carbon into tree-ring cellulose from either carbohydrate reserves, and/or from current carbohydrates derived from old stem respiratory $CO_2$ recycled in cortical photosynthesis.

Despite our simplistic approach with assumptions based on literature data from very different tree species and not taking into account pool size dynamics, the offset in $F^{14}C$ between baobab cellulose and NH3 data during the time around the bomb peak is well captured. Nonetheless, baobab $F^{14}C$ for 1945, 1952-1954, 1956 and 1957 lie slightly above the calculated range of $F^{14}C$ of the NSC pool. Although to a much lesser extent, this is in common with $F^{14}C$ values measured for the TPBs strongly indicating that the wood of these rings may have contained an unrecognized high fraction of intraxylary parenchyma cells and did not consist of pure xylem fibers and vessels.

### 4.2 $F^{14}C$ of TPBs indicates ongoing formation and/or restructuring of parenchyma tissue for several years

Terminal parenchyma bands (TPBs), marking the visual tree-ring boundaries, were found to be at least up to 1.6 years younger than the corresponding tree ring when calibrated with post bomb NH3 and assigned to the youngest possible age (Table 2, Fig. 8). In addition, the highest correlation between the width of TPBs and RWI was found for the ring built one year after the corresponding tree ring. Initial parenchyma bands are being produced each year initiated by the cambium between wood (xylem) and bark. However, well-defined TPBs could only be found deeper in the xylem, marking well the actual tree-ring boundaries, but also explaining previously described difficulties in cross-dating the outer part of baobab core samples (Johansson, 1999; Slotta et al., 2017).

Since parenchyma cells retain their ability to divide throughout their lifespan (Spicer, 2014) it is thus possible that the proliferation, i.e., post-cambial parenchyma growth, starts once the existing NSC storage volume is sufficiently refilled. This point in time may vary, depending on the tree's current water and nutrient budget, and may sometimes not be reached at all. However, parenchyma tissue may carry a generally different radiocarbon signal than the surrounding wood vessels and fibers, although both types of xylem are likely being fed by the same mixed NSC pool with the only difference being that fibers and vessels are annual and parenchyma cells are perennial and can undergo structural changes for an unknown number of years. Fractions of parenchyma tissue in between vessels and fibers might be easily overlooked during the sample dissection procedure causing a potential shift in $F^{14}C$ towards younger ages. The assumed lifespan is based on the significant correlation

found between the width of TPBs and the width of rings built six years later. However, the TPBs seem to be more variable in their $^{14}$C content than vessels and fibers. Probably, because their growth is not continuous, but rather discontinuous during their life following the year of initialization. Presumably, parenchyma growth depends on NSC availability and the demand of water, as well as the supply of water by the soil. A significant post-cambial growth, i.e. widening of TPBs could not be observed for 1973, 1981, and 1982, which is two years before the missing tree ring in 1975 and one and two years before the missing tree ring in 1983. This leads to the assumption that baobabs prioritize growth of fibers and vessels over storage enlargement, i.e. formation of new parenchyma cells. When there is not enough energy to invest in growth, baobabs seem to refrain from broadening the initial TPBs. However, the ten TPBs prior to the missing ring in 1951 were all notably thickened; the reason for this missing ring remains unclear. The fact that TPBs accounting for nine years (2006-2014) were not well-developed in the outer part of the wood core potentially indicates a particularly high stress level over the last decade, which is in line with the meteorological data, i.e. increasing $T_{max}$ and decreasing precipitation (Fig. 4).

In sum, the relatively high fraction (69-88 %) (Chapotin et al., 2006c) and perennial character of parenchyma tissue in stem succulent baobabs lead to shifts of the atmospheric $F^{14}C$ signal in their wood. The relative content and lifetime of intraxylary parenchyma may be also considered when reconstructing atmospheric $F^{14}C$ from tree rings of other woody plant groups. Conifers were found to have the lowest parenchyma fractions, followed by angiosperm trees (including shrubs), woody climbers (lianas) and succulents (Morris et al., 2016). Major differences in the relative parenchyma content of angiosperms were also found between tropical species ($36.2 \pm 13.4$ %) and temperate species ($21.1 \pm 7.9$ %) (Morris et al., 2015), i.e. the atmospheric $F^{14}C$ signal in tree-ring cellulose may be altered in a species specific and/or even plant individual way. This, however, only plays a role in radiocarbon dating exercises utilizing the bomb peak with its dramatic changes over a few years.

**4.3 Baobabs have annual growth rings, but visual recognition depends on tree-ring width and shaping of the TPBs**

Besides the issues discussed above, the high number of radiocarbon dates allowed to confirm the annual nature of the baobab growth zones. Very thin TPBs ($\leq 0.1$ mm) cannot not be identified as such with a solely wood anatomical approach. From this study, only xylem increments of $\geq 2$ mm in combination with TPBs of $\geq 0.5$ mm thickness can be considered an annual tree ring with confidence. Thus, a high number of radiocarbon dates is advisable to date time series of centuries-old baobab wood samples for climate reconstruction purposes, not only because of varying external growth conditions but also because baobabs show varying growth rates during the different phases within their stem development (Breitenbach, 1985), and because very old individuals are in fact multi-stemmed conglomerates (e.g. Patrut et al., 2015b). However, annual time resolution appears to be possible particularly for wood sections with ample ring width and distinct TPBs. Here, we worked with fresh samples, that were stored airtight and cool in the field to avoid distorting the original wood anatomical characteristics. When air-drying baobab, stem tissues shrink and, in some cases reveals fissures between the TPBs and the part of a tree ring that is dominated by fibers and vessels. Assuming those fissures to mark actual tree-ring boundaries, individual growth rings for stable isotope and/or radiocarbon sampling have been peeled off for paleoclimatic studies (e.g. Woodborne et al. 2015, 2016). This sampling strategy waives the elaborate securing of freshly taken increment cores (c.f. Sampling strategy) on the expense of not obtaining

exact growth increment data. Another constraint is that formation of more than one parenchyma band per year can easily

introduce substantial uncertainty in tree-ring counting. In this study, 3 missing rings were detected in a time series of 65 years from a rather well-grown young individual suggesting that dating errors of 5 % or more have to be considered. It has also to be noted, that the sampling procedure based on dry, deformed wood will inhibit the possibility to deduce intra-ring isotope signals, as from this study design. Nonetheless, precipitation reconstructions of good quality from $\delta^{18}O$ of baobab tree-ring cellulose seem to be possible, as $\delta^{18}O_{mean}$ revealed highly significant correlations with the May-September period (Table 3).

### 4.4 Intra-annual stable isotope courses are in line with isotope fractionation theory

The intra-annual stable isotope courses were rather similar for both $\delta^{13}C^*$ and $\delta^{18}O$ (Fig. 5). The first half of the tree ring was characterized by a majority of minimum values per year, whereas the second half comprised mainly maximum values. In terms of intra-annual $\delta^{13}C^*$ the observed seasonal course is very different to the recurring pattern described for deciduous tree species from temperate (Helle and Schleser, 2004) and tropical zones (Schollaen et al., 2013), where maximum $\delta^{13}C$ values were

490 generally observed during early wood formation indicative for the use of [13]C-enriched reserves like starch at the beginning of the vegetation period. Conversely, the baobab's seasonal $\delta^{13}C$ pattern generally showed low- to lowest values at the beginning of each tree ring which points to the use of other, [13]C-depleted, non-structural carbohydrates that might be originating from cortical photosynthesis (Kotina et al., 2017) producing assimilates with lighter $\delta^{13}C$ compared to those from leaf photosynthesis (Cernusak et al., 2001). Additionally, the significant autocorrelation found for $\delta^{13}C^*$ strongly indicates the incorporation of

495 previous years' carbon into tree-ring cellulose from either carbohydrate reserves and/or from current carbohydrates derived from old stem respiratory $CO_2$ recycled in cortical photosynthesis. Unlike for $\delta^{13}C^*$, no autocorrelation was present in $\delta^{18}O$. A lack of significant autocorrelation in $\delta^{18}O$ series from tree-ring cellulose is quite common (Treydte et al., 2007), however, this could not be expected here per se considering the relatively large water saving capacity of baobab stemwood potentially storing varying amounts of mixed precipitation waters from several monsoon seasons. Baobabs draw upon their water reserves

when flushing leaves prior to the onset of the monsoon season (Chapotin et al., 2006b) which may lead to a very high annual turnover diminishing potential autocorrelation in $\delta^{18}O$ of stem, i.e. source water. Despite the differences in autocorrelation, $\delta^{18}O$ intra-ring values revealed a higher variability than those of $\delta^{13}C^*$. This may point to the fact that the source water (precipitation) isotopic signature already does contain a climatic signal related to air temperature and precipitation amount, whereas $\delta^{13}C$ of atmospheric $CO_2$ does not. Highly variable rainfall amounts from pre- or post-monsoonal cyclones comprise

correspondingly variable $\delta^{18}O$ signatures (Fig. 1C). The significant correlation (r= -0.61, $p < 0.0001$, n = 56) between RWI and $\delta^{18}O_{min}$ (occurring mainly at the beginning or within the first half of a tree ring) may be an indication for a significant influence of moisture supply on tree growth at the beginning of the wet season. Low $\delta^{18}O$ of tree-ring cellulose values reflect higher rainfall and less evaporative [18]O enrichment of leaf water, i.e. lower VPD evoking favorable growth conditions and vice versa.

Climate relevance of tree-ring $\delta^{13}C$ is originating only from fractionation processes during photosynthetic $CO_2$ uptake. Nonetheless, the overall high similarity between the average intra-annual $\delta^{13}C^*$ and $\delta^{18}O$ courses suggests that stable isotope

fractionations at leaf level (Farquhar et al., 1989; Roden et al., 2000; Kahmen et al., 2011), where water and carbon cycles meet during photosynthesis, are of predominant importance. In accordance with fractionation theories, maximum values at or close to the end of the tree ring likely do correspond to the end of the monsoon period, when relative humidity is getting low and leaf-to-air vapor pressure deficit is rising, thus causing corresponding leaf water $^{18}$O enrichment to increase while enzymatic discrimination against $^{13}$C during photosynthetic uptake of atmospheric $CO_2$ is declining.

### 4.5 Intra-annual stable isotope courses are indicative for years with heavy rainfall events from cyclones

Starting first with low to intermediate values, intra-ring $\delta^{13}$C* and $\delta^{18}$O of the mid-section either increase or decline in dry or wet years, respectively. In particular, heavy pre-monsoonal rainfall from spring cyclones did not significantly alter start and end values but apparently causes a distinct decline in the mid-section of the general stable isotope course (Fig. 5C). In general, pre-monsoonal precipitation can help to replenish the stem water reserves during and/or shortly after leaf flush, and the monsoonal influence on the baobabs' ecophysiology was reflected by the significant correlations between $\delta^{18}O_{min}$ and RWI with April and May rainfall (Table 3, Fig. 7A, B). However, strongly depleted $\delta^{18}$O signatures of heavy pre-monsoonal rainfall events are not imprinted into the intra-ring $\delta^{18}$O pattern as a deep, short-term dip, but rather lead to a distinct trough in the mid-section. This points to a lagging, or more indirect influence by diluting the $\delta^{18}$O signature of the stem water and/or by causing wetter (micro-)site conditions evoking reduced leaf water $^{18}$O enrichment, but also increased $^{13}$C discrimination at the beginning and during the main monsoon season. Note, isotopically lighter values for the mid-section were observed also with increasing number of observations per year. $\delta^{18}$O courses from wide tree rings with more than 7 observations per year revealed a short-term mid-section decline (Fig. 5B, right) resulting from higher monsoon precipitation as suggested by the significant correlations with RWI (Table 3). The expression of this dip within the intra-annual $\delta^{18}$O pattern is relatively weak, however, it may hamper the clear detection of unknown spring cyclonic events in potential future reconstruction studies.

Only two years (1948, 2004) with autumn cyclones bringing more than 90 mm of rain could be investigated (Fig. 6A). Due to the lack of replication, a conclusive evaluation of intra-ring stable isotope response to post-monsoonal cyclones is not possible, especially with respect to the observed indistinct $\delta^{13}$C* courses (Fig. 6A, left). However, $\delta^{18}$O seems to largely respond to autumn cyclones with a strong decline to very low values in the last quarter of the intra-ring course. When compared to the average seasonal $\delta^{18}$O course and those for years with spring cyclones, intra-ring $\delta^{18}$O values of years with autumn cyclones were found above in the first half, and below at the end of the tree-ring (Fig. 6A, right). Furthermore, post-monsoonal rain seems to prolong the growing season as indicated by the rather wide tree ring of 1948 that provided 8 intra-annual isotope samples. Based on theoretical considerations and studies on temporal dynamics of soil water stable isotopes at the soil-plant-atmosphere interface of the critical zone (Sprenger et al., 2017; Treydte et al., 2014), the conspicuously strong increase in $\delta^{18}$O towards the very end might reflect evaporative $^{18}$O enrichment of the soil water taken up by the tree, and/or effects due to increased evapotranspiration at leaf level during the prolonged growing period after a heavy autumn rainfall event. In sum, the intra-ring stable isotope courses are modified by pre- and post-monsoonal rainfall events from cyclones. Significantly negative

correlations of $\delta^{18}O_{max}$ with cloud cover for previous October and the current April also indicate an influence of non-monsoonal precipitation (Table 3, Fig. 7C). Hence, baobabs might well take advantage of rainfall outside their actual vegetation period.

### 4.6 Increased stable isotope values in mid-section of intra-annual courses reflect low monsoon precipitation

A clearer picture with more constrained intra-ring isotope patterns was obtained for dry years with less than 50 % of the average monsoon precipitation and no pre- or post-monsoonal cyclones. Except for the year 2000, all intra-ring $\delta^{13}C^*$ and $\delta^{18}O$ courses reveal increased values for the mid-section of a tree ring (Fig. 6B). The very low variability of $\delta^{18}O$ in rain or fog within and between different monsoon seasons (Table 1) suggests that the observed $\delta^{18}O$ pattern has to be largely attributed to the effects of fractionation processes at leaf level. Vapor pressure deficit between leaves and ambient air is higher during a dry monsoon season causing enhanced $^{18}O$ enrichment in leaf water, and lower stomatal conductance with decreased leaf internal $CO_2$ concentrations causing photosynthetic $^{13}C$ discrimination to decline. Decreasing values at the very end of an intra-annual stable isotope course can be observed in $\delta^{13}C^*$ of three years (1974, 2000, 2001) and in all but one $\delta^{18}O$ year (2003), which might be attributed to the decreasing transpiration and reduced photosynthesis towards the end of the growing period, i.e. during leaf senescence. $\delta^{13}C^*$ of 1968, 1972 and 2003 did not reach the high values in the tree rings of the other dry years, however, in the second half they continue to increase (at a slower rate) to the end. This trend may or may not be an artefact of the low resolution (3 values per year) causing a more or less linear increasing trend (c.f. Fig. 6A). The year 2000 showed an unexpected early decline in $\delta^{18}O$, as seen for spring storm years, that might be due to above normal precipitation from a fall storm in October 1999. However, the $\delta^{18}O$ time series shows no autocorrelation and this finding could not be verified for other years because the tree rings following other fall storms were only half as wide and the signal is probably lost in the resolution.

### 5 Conclusions

This study comprises 62 years of highly resolved inter- and intra-annual analyses of $F^{14}C$, $\delta^{13}C$ and $\delta^{18}O$ on baobab tree-ring cellulose. The fairly high number of radiocarbon dates has confirmed that baobabs have annual growth rings that are separated by terminal parenchyma bands. However, unambiguous visual recognition of tree-rings was possible only for xylem increments of $\geq 2$ mm in combination with TPBs of $\geq 0.5$ mm thickness. Situated in Oman, the baobab $F^{14}C$ values were expected to fit the post-bomb atmospheric curve NH3. However, they were found notably lower than NH3 and even lower than SH3 around the atomic bomb peak period (1964-1967). A certain spread between tree species and sites is well known (cf. Fig. 3 such large offsets are uncommon, but have been reported earlier (e.g. Chakraborty et al., 1994)). Potential consequences should be considered when dating tropical trees with indistinct growth pattern and high abundance of parenchyma tissue. Normally, three radiocarbon samples placed within the bomb peak period result in a very precise dating of tree rings. But, with the bomb peak represented by clearly lower $F^{14}C$ values compared to the corresponding atmospheric zone, a lower sampling rate on parenchyma-rich tree species like the baobab can lead to a misinterpretation of the time series' resolution and duration. Usually, external factors like e.g. fossil fuel dilution from neighboring industrial or urban areas are considered likely causes

for deviations of tree-ring data from atmospheric $F^{14}C$. However, the large extent of the $F^{14}C$ offset observed here for *A. digitata*, with its unique stem anatomy, puts the focus on potential tree physiological explanations, namely carbohydrate turnover and longevity of certain stem tissue (parenchyma). A conceptual modeling approach suggests that the observed shift between atmospheric and baobab $F^{14}C$ is caused by a mixture of non-structural carbohydrates fueling baobab wood formation with 85% contribution from a 'fast' cycling and 15% from a 'slow' cycling pool with respective annual renewal rates of 95%

(renewal `fast`cycling pool) and 5% (renewal `slow`cycling pool). Furthermore, radiocarbon dates of terminal parenchyma bands, marking the visual tree-ring boundaries, were found to be at least up to 1.6 years younger than their corresponding tree ring. This can be explained by the well-known ongoing restructuring and/or formation of new intraxylary parenchyma tissue years after initial tree-ring formation. Stem succulent baobabs have a very high fraction of long-lived parenchyma tissue that is not only arranged in tangential bands but also irregularly dispersed within the wood modifying its $F^{14}C$ values. Hence, the

relative content and lifetime of intraxylary parenchyma may be also considered when reconstructing atmospheric $F^{14}C$ from tree rings of other woody plant groups or when dating particularly parenchyma-rich succulent plants. Nonetheless, the high-resolution bomb-peak dating approach together with visual inspection of TPBs allows the development of precise time series useful for investigating the ecophysiology of baobabs and to assess their vulnerability to recent climate change. Well dated longer time series for paleoclimatic reconstructions are still more difficult to establish, although annual time resolution appears

to be possible for wood sections with ample ring width and distinct TPBs. Tree-ring width indices (RWI) and stable isotope ratios have revealed significant climate sensitivity. RWI and intra-ring $\delta^{18}O_{min}$ correlated well with each other (r = -0.6, p < 0.0001) and with precipitation amount for the period from pre-monsoon May to the end of the monsoon season in September-October. Intra-annual stable isotope courses were found rather similar for both $\delta^{13}C$ and $\delta^{18}O$. The distinct pattern seems indicative for years with heavy rainfall events from pre-monsoonal cyclones. Rain from post-monsoonal cyclones may

also be recorded, however, a conclusive evaluation was not possible because of only two years of observation. Years with particularly low monsoon rain were reflected by increased stable isotope values in mid-section of intra-annual courses. When intra-ring isotope signals, as from this study, cannot be deduced, precipitation reconstructions of good quality from cellulose $\delta^{18}O$ seem to be possible, as $\delta^{18}O_{mean}$ revealed highly significant correlations with the May-September period. Overall, in the baobab archive $\delta^{18}O$ appeared to be a good climate proxy followed by tree-ring width and $\delta^{13}C$.

**Acknowledgements**

We are grateful to the Director General of Nature Conservation Suleiman Bin Nasser Al Akhzami and the Director of Biodiversity D. Thuraya Said Al-Sariri at the Ministry of Environment & Climate affairs for granting us permission (Permit No. 11/2015) to sample baobab trees in Wadi Hinna and to the Ministry employees involved in the collection process for their field assistance. Furthermore, we would like to extend our gratitude to Bernhard Pracejus and Frank Mattern (Sultan Qaboos

University) for collaboration. Frank Mattern and his wife Sarah as well as Annette Kossler and Heidemarie Pettauer also assisted in the field. We are also grateful to Quan Hua for making available published radiocarbon data. We would further like

to thank Silvia Bollhalder, and Philip Gautschi (ETH Zurich) for their support in the laboratory and Andreas Hendrich for his help with the figure layout. We acknowledge the fruitful discussions with Arne Ramisch and thank Daniel Balanzategui and Patrick Lynch for the linguistic revision of the manuscript. Furthermore, we thank the reviewers Guaciara M. Santos and Supryo Chakraborty for their valuable criticism. This study was funded by the DFG (HE3089/10-1, RI 809/32-1) and supported with $^{14}$C measurements performed at the Laboratory of Ion Beam Physics (Prof. H.A. Synal) at ETH Zurich.

## Data availability

The data produced in this publication will be available from the PANGAEA database.

## Supplement

There is supplement related to this article.

## Author contributions

FS, FR, GH and LW conceived the study. Field work for this study was conducted by FS and FR. FS and LW performed the F$^{14}$C analyses. FS and GH performed chemical extraction of tree-ring cellulose and measurements of stable isotope ratios of carbon and oxygen. GH and FS prepared the manuscript with contributions from all co-authors.

## Competing interests

The authors declare that they have no conflict of interest.

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

## Tables and Figures

**Table 1: Compilation of published oxygen stable isotope data of monsoon rainfall, fog, stemflow, throughfall, and cyclone water from the Dhofar mountains for different years. Site coordinates are as follows: Salalah (17° 01' N, 54° 04' E), Qairoon Hairitti (17° 15' N, 54° 05' E), Gogub (17° 12' N, 54° 06' E), Tawi Atir (17° 06' N, 54° 31' E), Qarhanout (17° 07' N, 54° 37' E).**

| Year | Sampling period | type | $\delta^{18}O$ ‰ VSMOW | sd | n | Site | Altitude m asl | Reference |
|---|---|---|---|---|---|---|---|---|
| 1988 | 06/01-08/12 | rainfall | 0.16 | 0.37 | 30 | Salalah | 20 | Wushiki, 1991 |
| 1989 | 06/03-05/08 | rainfall | 0.31 | 0.57 | 25 | Salalah | 20 | Wushiki, 1991 |
| 1989 | 07/01-08/08 | rainfall | -0.52 | 0.26 | 42 | Qairoon Hairitti | 880 | Wushiki, 1991 |
| 2008 | Jun-Aug | rainfall | 0.03 | 0.02 | 2 | Gogub | 450 | Strauch et al., 2014 |
| 2009 | Aug | rainfall | -0.22 | 0.07 | 4 | Gogub | 450 | Strauch et al., 2014 |
| Average | | rainfall | -0.05 | 0.26 | 103 | | | |
| 2009 | 08/17-08/19 | fog | 0.09 | 0.08 | 3 | Gogub | 450 | Strauch et al., 2014 |
| 2009 | 08/18-08/20 | fog | -0.08 | 0.16 | 2 | Tawi Atir | 650 | Strauch et al., 2014 |
| Average | | fog | 0.01 | 0.12 | 5 | | | |
| 2009 | 08/17-08/19 | throughfall | 0,02 | 0,02 | 3 | Gogub | 450 | Strauch et al., 2014 |
| 2009 | 08/18-08/20 | throughfall | -0.01 | 0.25 | 2 | Tawi Atir | 650 | Strauch et al., 2014 |
| 2011 | 08/09-08/11 | throughfall | 0.04 | 0.32 | 8 | Tawi Atir | 650 | Strauch et al., 2014 |
| Average | | throughfall | 0.02 | 0.20 | 13 | | | |
| 2009 | 08/17-08/19 | stemflow | 0.09 | 0.07 | 3 | Gogub | 450 | Strauch et al., 2014 |
| 2009 | 08/18-08/20 | stemflow | 0.03 | 0.24 | 2 | Tawi Atir | 650 | Strauch et al., 2014 |
| Average | | stemflow | 0.06 | 0.16 | 5 | | | |
| 1988 | Apr | cyclone | -2.79 | | 1 | Salalah | 20 | Wushiki, 1991 |
| 2011 | 11/04 | cyclone | -10.76 | | 1 | Qarhanout near Jabal Samhan | ~ 940 | Strauch et al., 2014 |

**Table 2: Comparison of the $F^{14}C$ values of terminal parenchyma bands (TPBs) and the corresponding tree ring (TR). In case of multiple measurements per ring, the mean value was taken for comparison. Considering the youngest possible calibrated age, most TPBs are significantly younger than the corresponding tree ring. End of growth assumed to be end of September, i.e. at 0.75 of each year. Please note that the calibration always yields age ranges on the rising side of the bomb peak as well as on the falling part. For years ≤ 1964 only the rising side and for years ≥ 1965 only the falling side was considered for consistency. Underlined calibrated ages are viable, others are theoretical, only, and given for sake of completeness.**


| Sample ID | Year of wood formation | $F^{14}C_{TB}$ | (mean) $F^{14}C_{TR}$ | difference TB-TR (%) | TB ages calibrated with SH1-2 (95.4 %) | TB ages calibrated with NH3 (95.4 %) |
|---|---|---|---|---|---|---|
| 242.PB | 1960 | 1.281780 | 1.227284 | 4.4 | (6.1 %) 1962.6 AD to 1962.96 AD <br> (4.8 %) 1963.18 AD to 1963.42 AD <br> (4.7 %) 1979.6 AD to 1979.88 AD <br> (76.6 %) 1980.12 AD to 1981.34 AD <br> (3.2 %) 1982 AD to 1982.2 AD | (5.7 %) 1962.34 AD to 1962.64 AD <br> (89.7 %) 1979.3 AD 1980.7 AD |
| 237.PB | 1961 | 1.381735 | 1.286768 | 7.4 | (35.5 %) 1963.52 AD to 1964.08 AD <br> (59.9 %) 1975.2 AD to 1976.12 AD | (9.8 %) 1962.76 AD to 1963.12 AD <br> (85.6 %) 1974.74 AD to 1976.08 AD |
| 231.PB | 1962 | 1.349989 | 1.266081 | 6.6 | (13.7 %) 1963.34 AD to 1963.76 AD <br> (81.7 %) 1976.3 AD to 1977.48 AD | (6.2 %) 1962.58 AD to 1963 AD <br> (89.2 %) 1976.1 AD to 1977.66 AD |
| 223.PB | 1963 | 1.460845 | 1.414688 | 3.3 | (7.3 %) 1964 AD to 1964.2 AD <br> (88.1 %) 1972.48 AD to 1973.38 AD | (8.7 %) 1962.98 AD to 1963.4 AD <br> (72.2 %) 1972.16 AD to 1973.06 AD <br> (14.5 %) 1973.28 AD to 1973.96 AD |
| 19.PB | 2005 | 1.056946 | 1.061611 | -0.4 | (5.0 %) 1957.6 AD to 1958.08 AD <br> (90.4 %) 2008.18 AD to 2011.1 AD | (2.3 %) 1957.62 AD to 1967.88 AD <br> (1.2 %) 2006.22 AD to 2006.36 AD <br> (91.9 %) 2006.84 AD to … |

Table 3: Significant correlations between the obtained baobab time series and climate parameters for the area around Wadi Hinna. Correlations within months of the current year are marked by upper case abbreviations, correlations within months of the previous year by lower case abbreviations and a grey background. Significance levels: * $p < 0.05$, ** $p < 0.01$, *** $p < 0.001$, **** $p < 0.0001$. Positive (negative) correlations are indicated with "+" ("-"). Note: $\delta^{13}C$ vales show some significant but weak correlations compared to RWI and $\delta^{18}O$.

| Baobab/ climate parameter | RWI | $\delta^{13}C^*_{min}$ | $\delta^{13}C^*_{mean}$ | $\delta^{13}C^*_{max}$ | $\delta^{18}O_{min}$ | $\delta^{18}O_{mean}$ | $\delta^{18}O_{max}$ |
|---|---|---|---|---|---|---|---|
| precipitation | apr*-<br>jul*-<br>jan-dec*-<br>May**+<br>Aug***+<br>May-Sep***+<br>Jan-Dec*+ | jan*-<br>Nov**+ | | oct*-<br>jan-dec*- | Apr*-<br>May****-<br>May-Sep**-<br>Jan-Dec**- | Apr*-<br>May****-<br>May-Sep***-<br>Jan-Dec***- | jan-dec*- |
| $T_{max}$ | Sep/Oct*- | | May*- | oct*-<br>May*- | dec*-<br>Feb/Mar*- | | |
| vapor pressure | | | Mar*-<br>May*- | jan-dec*-<br>feb/mar/apr*-<br>oct*-<br>May*- | Feb/Mar*- | | feb*- |
| cloud cover | | | Apr*- | | Feb*- | Apr*- | oct*-<br>Apr***-<br>Aug**- |
| rH anomality | | | aug*+ | | jan*+<br>jul/aug**+<br>Mar*+ | jul*+<br>aug**+<br>Mar*+ | nov*+<br>Mar*+ |
| SPEI (1 month) | oct*-<br>dec**+<br>Jan*+<br>May*+<br>Aug**+<br>May-Sep*+<br>Jan-Dec*+ | | nov*+<br>Nov*+ | May*+<br>Nov*+ | apr*+<br>dec*-<br>May**-<br>Aug**-<br>May-Sep*- | dec**-<br>May**-<br>Aug*-<br>May-Sep*- | Oct*+ |

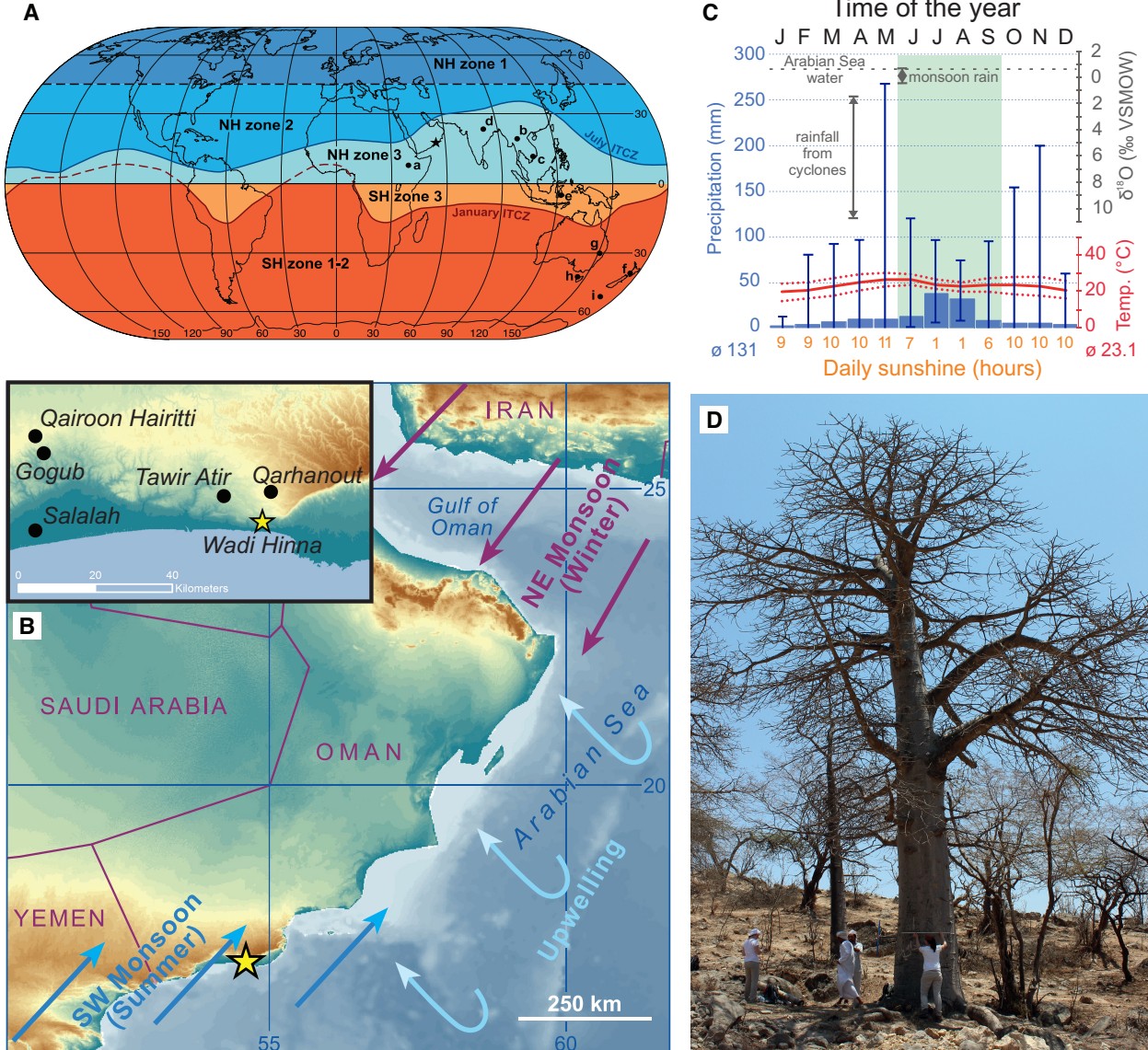

**Figure 1: Context of the study site. (A)** World map showing the northern hemispheric (NH) and southern hemispheric (SH) zones of atmospheric bomb ¹⁴C (after Hua, et al. 2013). Zones close to the equator (NH3, SH3) are still underrepresented in studies reconstructing the atmospheric radiocarbon content. Star indicates study site; sites labeled with letters a-i correspond to sample locations of F¹⁴C studies referred to in figure 5. **(B)** Map showing the location (yellow star) of the tree site Wadi Hinna (318 m asl) at the edge of the Dhofar mountains in southern Oman (17° 03' N, 54° 36' E) (map topography based on SRTM and bathymetry data from Becker et al. 2009). Prevailing seasonal wind patterns are shown by blue (summer monsoon, rainfall season) and purple arrows (winter monsoon, dry season). Light blue arrows indicate upwelling of deep waters from the Arabian Sea. **(C)** Climate diagram for the area around Wadi Hinna (17.0-17.5° N, 54.0-54.5° S) with average mean precipitation (bars indicating minima and maxima; source: GPCC, gpcc_V7_05) and temperature for 1942-2013 (dotted lines representing Tmin and Tmax, respectively; source: CRU TS4.01, GPCC V7_05). The isotopic signature of the Arabian Sea water and the precipitation around the study site is indicated in dark grey and varies significantly between monsoonal and non-monsoonal precipitation (Wushiki, 1991; Strauch et al., 2014). Average daily sunshine hours are given in orange (1979-1990, NOAA: ftp://ftp.atdd.noaa.gov/pub/GCOS/WMO-Normals/RA-II/OM/41316.TXT) and the timing of the summer monsoon and the approximate growing season of the baobab trees is indicated in green. **(D)** Sampled baobab tree at Wadi Hinna.

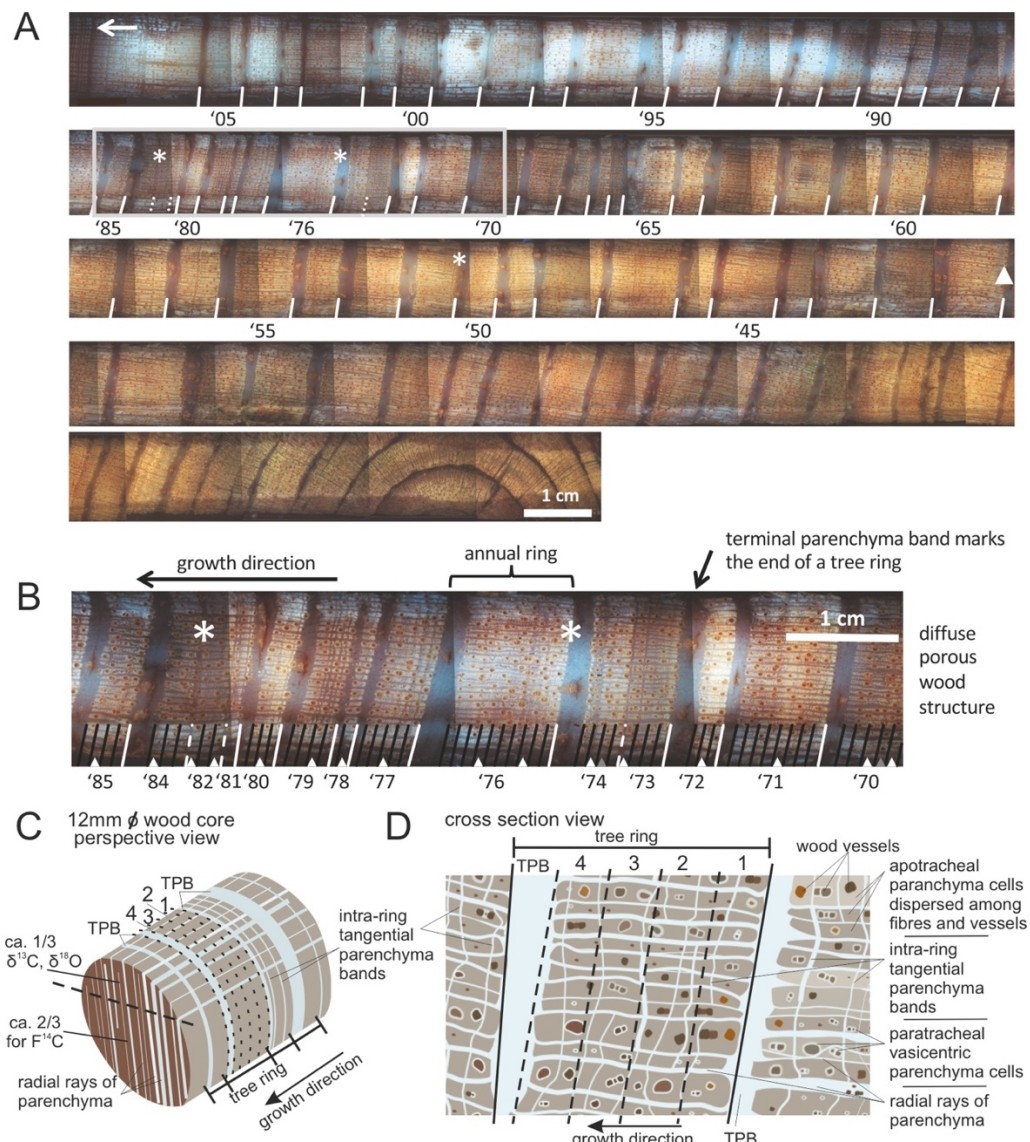

**Figure 2: Wood anatomy of the sampled baobab core. The right side of each photograph is unintentionally darker, causing vertical cut-offs in the stitched panorama. White asterisks mark positions of missing rings identified by radiocarbon dating. A) Stitched core picture photographed under ultra-violet light with automatic white balance showing alternating diffuse porous wood sections and clearly defined terminal parenchyma bands. Tree-ring boundaries and years of growth are indicated for the analyzed core section, i.e. from cambium (white arrow, top left) until 1941 (white triangle). B) Enlarged core section (grey frame on top) with indicated tree-ring boundaries (white lines), intra-annual subsamples (black lines), radiocarbon dated samples (white triangles: holocellulose, grey triangles (1970, 1974): alpha cellulose) and years of growth. Boundaries indicated by dashed white lines were only identified after additional $^{14}$C analyses. C) Schematic drawing of an increment core section indicating the high abundance of radial and tangential parenchhyma bands. Tree rings end with a terminal parenchyma band (TPB), however, they also contain intra-annual tangential parenchyma bands of various thickness. Intra-annual tangential slices of ca. 1mm thickness were dissected with a scalpel (dashed lines). 2/3 of each slice was kept for $F^{14}$C, 1/3 was used for stable isotope analysis. D) Schematic drawing of a cross section view illustrating different percularities of parenchyma tissue in baobabs (parenchyma cells dispersed among wood fibers and vessels not shown). For details cf. text.**

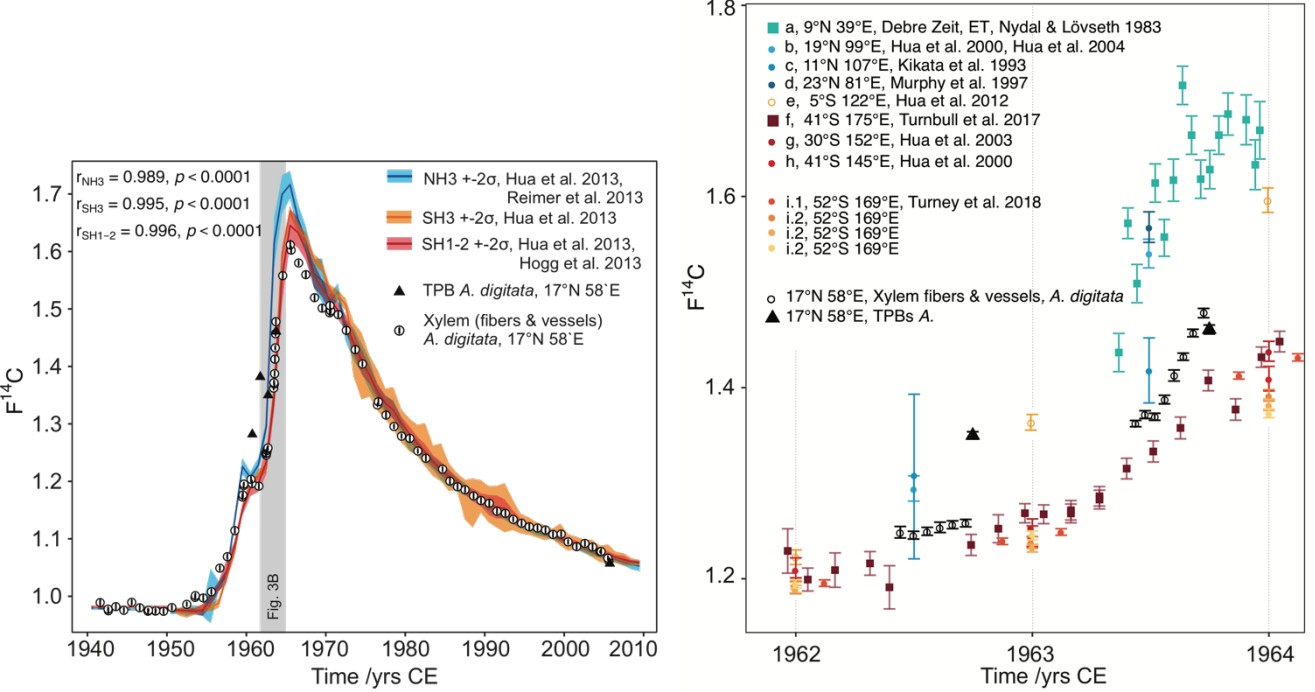


**Figure 3: (A)** Comparison of baobab $F^{14}C$ values with $F^{14}C$ curves for both hemispheres. Data prior to 1950 taken from IntCal13 and SHCal13 (Reimer et al., 2013, Hogg et al., 2013). 1950-2010 originates from Hua et al., 2013 (line: mean value, shade: ± 2 standard deviations (σ)). The southern hemispheric increase in atmospheric $F^{14}C$ lags behind the northern hemisphere by about 1 year and its peak is considerably shallower. The overall conformity of the baobab measurements (white dots with 2 σ error bar)

with the course of the calibration curves proves the annual character of the growth zones. The values for baobab tree-ring cellulose (xylem fibers and vessels) lie between the values for both hemispheres for the steep increase in $F^{14}C$ and then apparently follow the course of SH1-2, although temporary shifted due to different vegetation periods. Four out of five terminal parenchyma bands (TPB, black triangles, error bars are smaller than the symbol size) show either much higher $F^{14}C$ values (1960-1962) or a lower value (2005) than the corresponding tree ring. **(B)** Baobab measurements in comparison to atmospheric (squares) and tree-ring data

(circles) for NH3 (a-d), SH3 (e), and SH1-2 (f-i) between 1962 and 1964. These years show strongly increasing atmospheric $F^{14}C$ values and were chosen to investigate the intra-annual baobab $F^{14}C$ values. For this period, the baobab $F^{14}C$ values lie between the higher NH3/SH3 and the lower SH1-2 values. A dip within the first half of the tree ring is followed by a slight (1962) or steep (1963) increase of the values. Note, the inter- and intra-species' variability from the same site (i.1 and i.2, Turney et al., 2018).

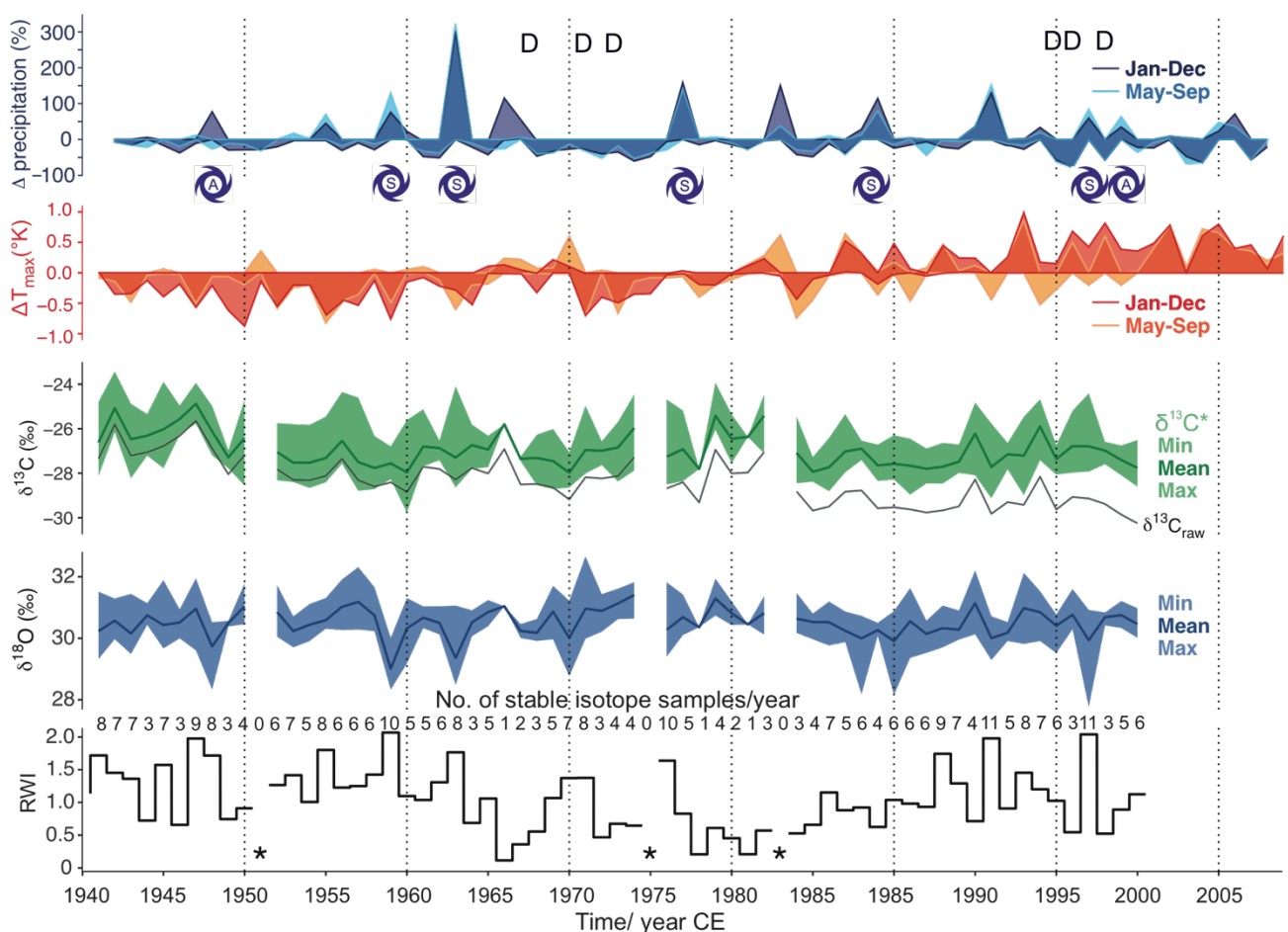

Figure 4: Analyzed baobab tree-ring parameters (ring width indices (RWI), $\delta^{18}O$, $\delta^{13}C*$) together with available precipitation and $T_{max}$ data for the period 1941-2014. Precipitation is presented as percentage differences from the mean, $T_{max}$ as differences in Kelvin. Darker colors reflect differences on an annual basis, lighter colors for the monsoon period Jun-September. The stable isotope values ($\delta^{13}C*$, $\delta^{18}O$) are presented with minima, maxima and mean values for each year. Mean $\delta 13C$, uncorrected for effects of changing atmospheric $CO_2$ concentration and corresponding $\delta^{13}C_{CO2}$ is given as black line ($\delta^{13}C_{raw}$). Dry years with $\leq 45$ mm of rain during the monsoon period are marked with D. Years with spring (S) or autumn (A) storms with $\geq 90$ mm of rain are indicated by cyclone symbols. Missing rings caused gaps in the time series and are marked by asterisks.

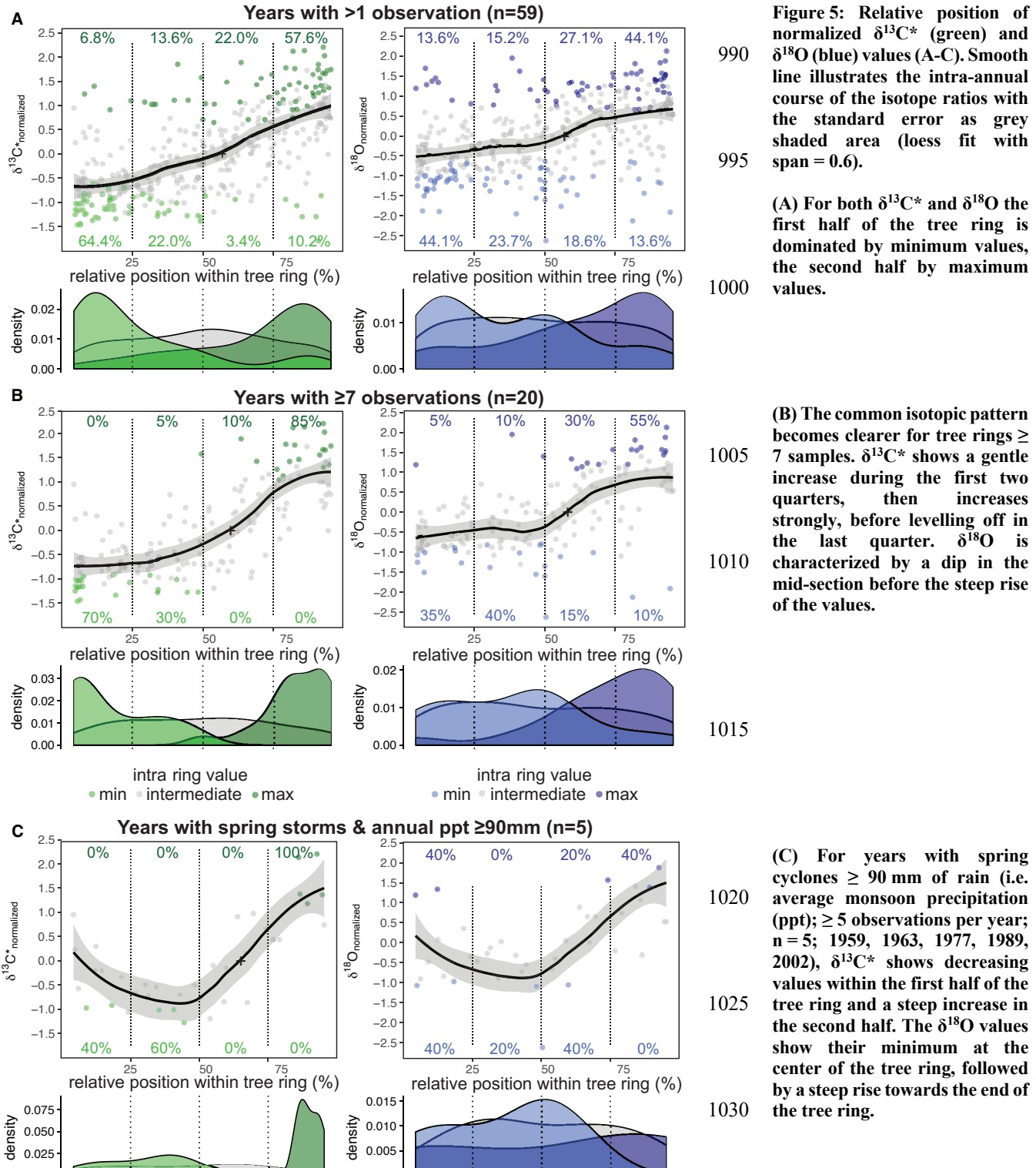

**Figure 5: Relative position of normalized δ13C* (green) and δ18O (blue) values (A-C). Smooth line illustrates the intra-annual course of the isotope ratios with the standard error as grey shaded area (loess fit with span = 0.6).**

**(A) For both δ13C* and δ18O the first half of the tree ring is dominated by minimum values, the second half by maximum values.**

**(B) The common isotopic pattern becomes clearer for tree rings ≥ 7 samples. δ13C* shows a gentle increase during the first two quarters, then increases strongly, before levelling off in the last quarter. δ18O is characterized by a dip in the mid-section before the steep rise of the values.**

**(C) For years with spring cyclones ≥ 90 mm of rain (i.e. average monsoon precipitation (ppt); ≥ 5 observations per year; n = 5; 1959, 1963, 1977, 1989, 2002), δ13C* shows decreasing values within the first half of the tree ring and a steep increase in the second half. The δ18O values show their minimum at the center of the tree ring, followed by a steep rise towards the end of the tree ring.**


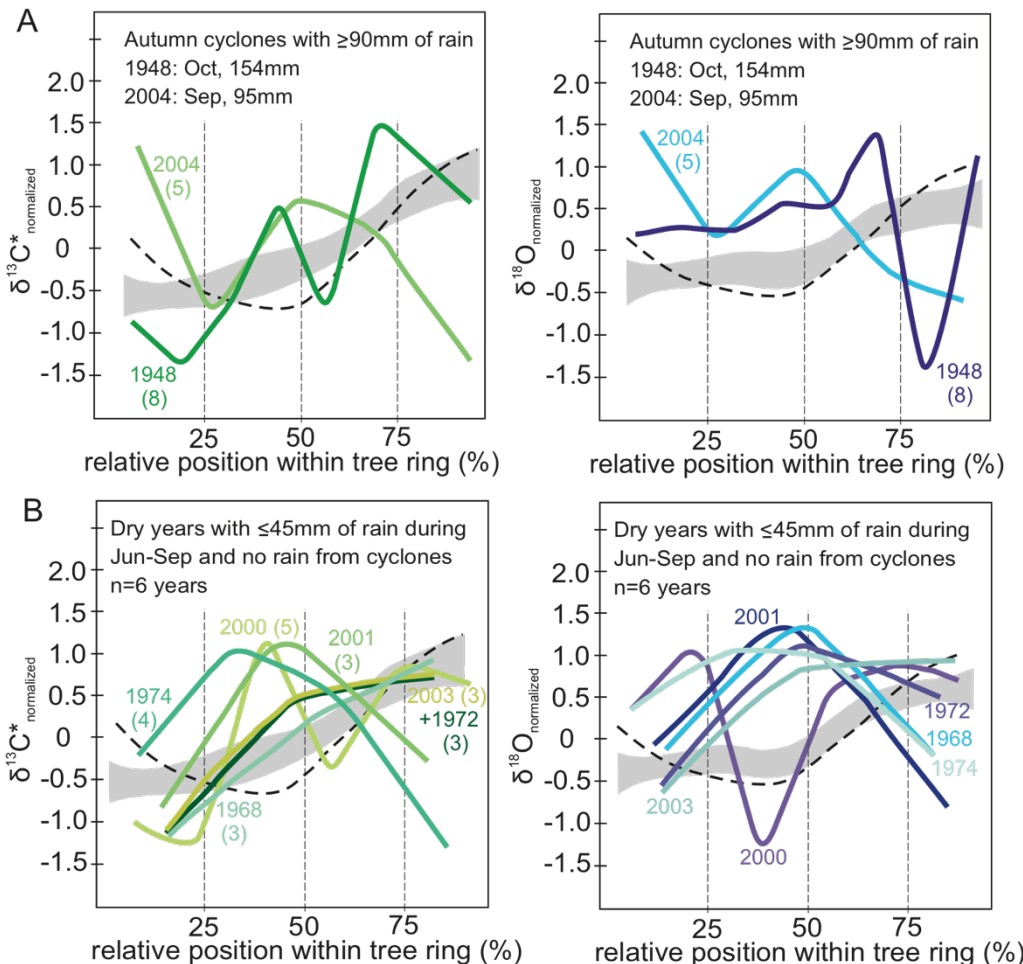

**Figure 6: (A)** Intra-annual course of normalized δ13C* (green) and δ18O (blue) values for years with autumn cyclones bringing ≤ 90 mm of rain (n = 2) and **(B)** dry years with ≥ 45 mm monsoonal precipitation and no rain from cyclones (n = 6). Numbers in brackets indicate number of isotope samples analyzed per year. Grey shaded area marks intra-annual course of isotope ratios with standard error for all years with ≥ 3 observations. Dashed line indicates intra-annual course of isotopes for years with spring cyclones, illustrated in detail in Fig. 5C. Rain from autumn cyclones may affect the intra-annual isotope signature by more strongly declining values towards the end of a tree ring. Years with less than 45 mm of monsoon precipitation show an increase in δ13C* towards the middle of each tree ring, then decline or increase only slightly towards the end. Rather similar seasonal trends can be observed for δ18O values. Hence, dry monsoon seasons usually cause higher isotope values in the middle part of each tree ring than during normal years. Please see text for further details on the conspicuous seasonal trends in 1948 and 2000.

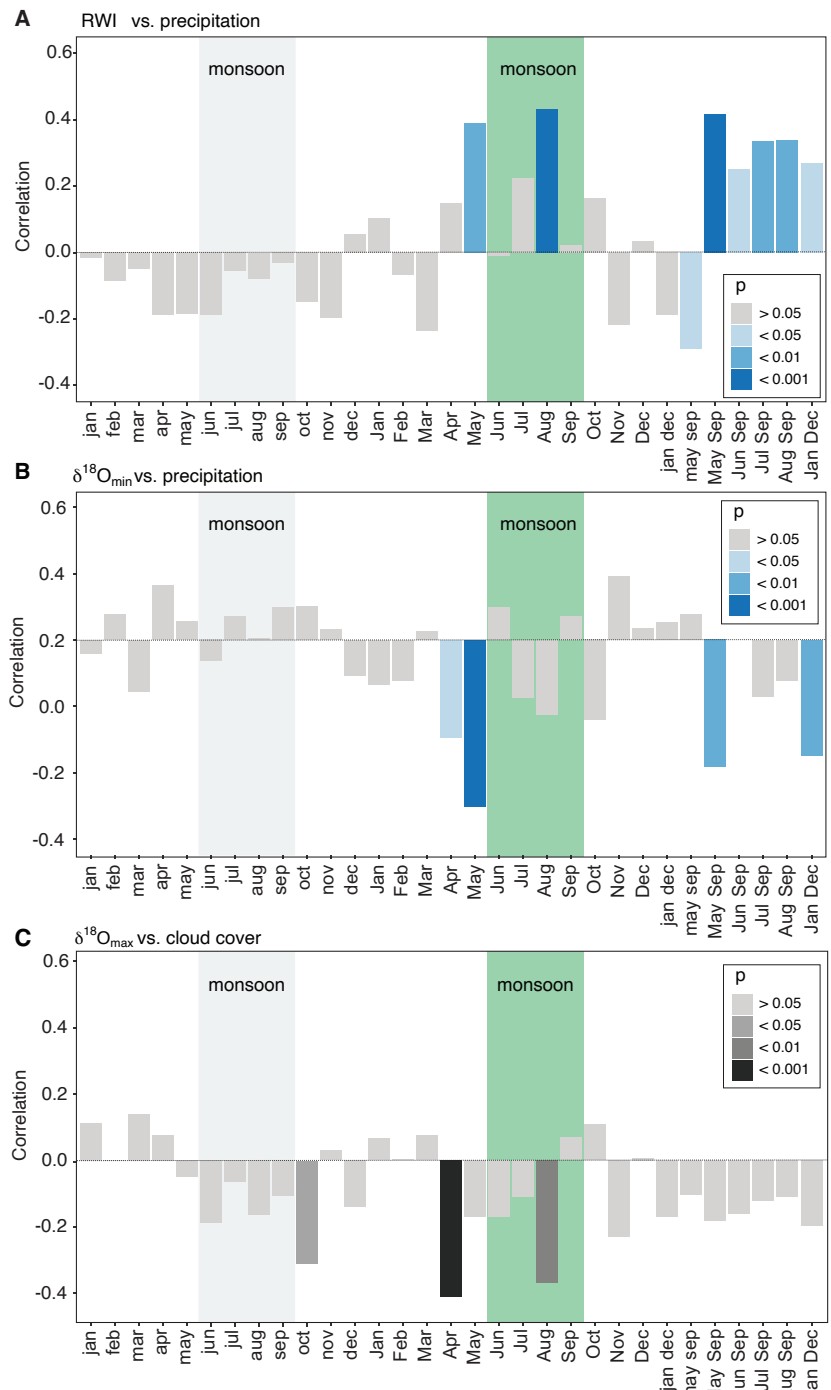

**Figure 7: Diagrams for the strongest correlations of baobab tree-ring parameters with monthly and seasonally resolved climate parameters. Months of the previous year are marked by lower case abbreviations, months of the current year by upper case abbreviations. The green box marks the average monsoon period. (A) Ring-width indices versus precipitation amount. (B) $\delta^{18}O_{min}$ versus precipitation. (C) $\delta^{18}Omax$ versus cloud cover. Apart from the actual monsoon season (Jun-Sep) also the pre-monsoon (Apr-May) has a significant influence on wood increment growth and $\delta^{18}O$ isotope signature.**

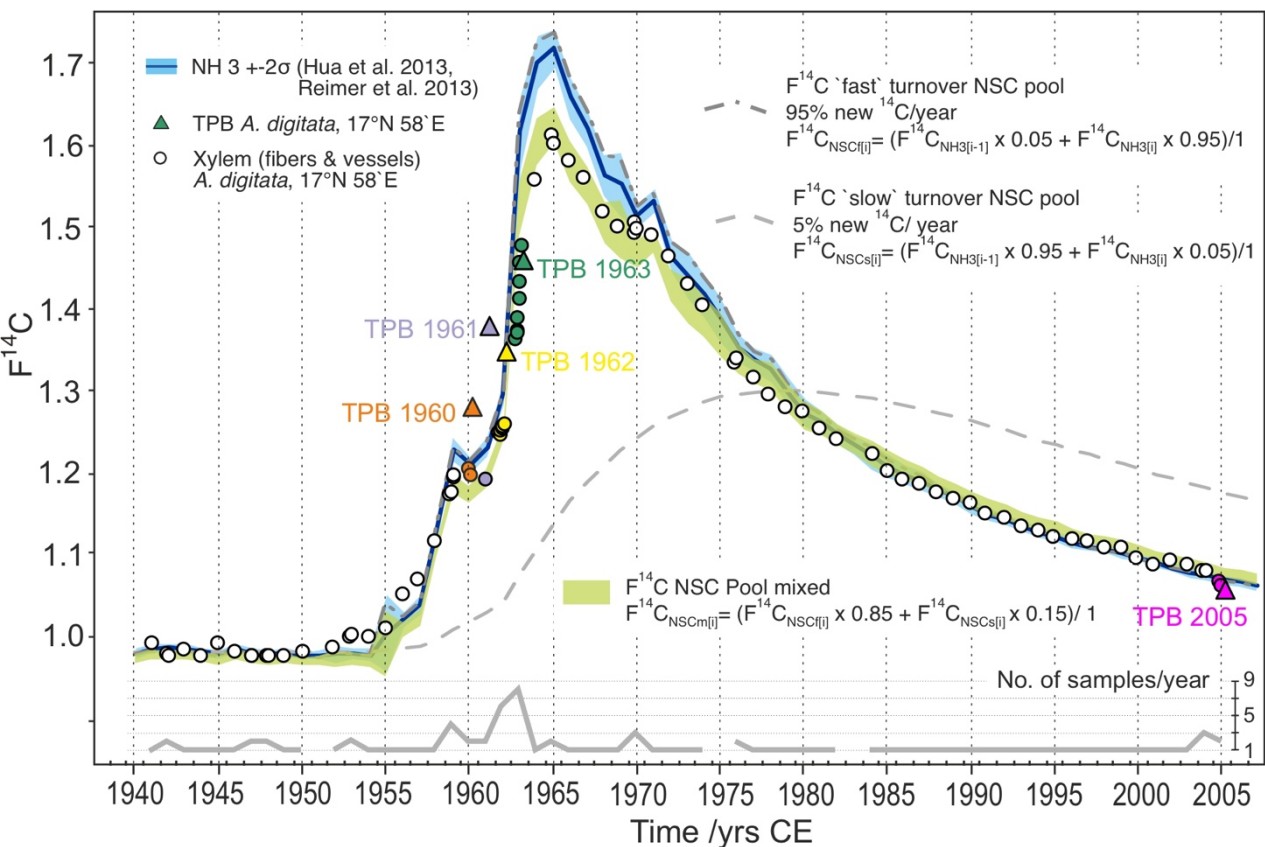

**Figure 8: Baobab F$^{14}$C values in comparison with the F$^{14}$C curve for NH3 (blue line/shade) and modelled pools of non-structural carbohydrates (NSC) fueling wood formation. The observed F$^{14}$C values of baobab xylem fibers and vessels (circles) are well approximated by assuming polymerization from a mixed NSC pool (F$^{14}$C$_{NSCm[i]}$) with contributions from a fast (F$^{14}$C$_{NSCf[i]}$) and slow (F$^{14}$C$_{NSCs[i]}$) turnover NSC pool, respectively. Range of F$^{14}$C$_{NSCm[i]}$ (green shade) is given by the 2σ range of atmospheric F$^{14}$C source values (F$^{14}$C$_{NH3[i]}$) for each year ($_{[i]}$). 1940 was chosen as starting year for the model calculations. Terminal parenchyma bands (TPB, colored triangles) show higher (1960-1962) and lower (2005) F$^{14}$C values than F$^{14}$C of NH3 and xylem fiber and vessel tissue of corresponding tree rings. This indicates distinct younger ages of TPBs and points to structural changes and ongoing cell division of parenchyma tissue for a number of years after initial formation. Some F$^{14}$C values of fiber and vessel samples (e.g. 1945, 1953, 1954, 1957, 1958) were also found above corresponding NH3 values. This might allude to the presence of intraxylary parenchyma tissue that could not be detected during the tree-ring dissection process.**