# Peer review of "High resolution 14C bomb-peak dating and climate response analyses of subseasonal stable isotope signals in wood of the African baobab – A case study from Oman"

_Biogeosciences, 2019_

## Author Comment (AC1) · 11 Sep 2019

Datasets can be downloaded from the PANGAEA data base:

Slotta, F. et al. (2019): Normalized 14C activity ratios (F14C) of an African baobab (Adansonia digitata) tree from Oman. https://doi.pangaea.de/10.1594/PANGAEA.905621

Slotta, F. et al. (2019): Subseasonal $\delta$13C and $\delta$18O of tree-ring cellulose of an African baobab (Adansonia digitata) tree from Oman.

https://doi.pangaea.de/10.1594/PANGAEA.905625

Slotta, F. et al. (2019): Tree-ring width indices (RWI) of an African baobab (Adansonia digitata) tree from Oman. https://doi.pangaea.de/10.1594/PANGAEA.905619

Best, Gerd Helle

---

## Short Comment (SC1) · Review of MS Slotta et al. · 22 Sep 2019

Radiocarbon analysis of the annual rings of trees has been carried out by several investigators to study a variety of natural processes. Such kind of records, especially in the extra-tropical region of the north hemisphere are widely available. The tropical region, however, is not well represented. To fill this gap Slotta et al. attempted to reconstruct an atmospheric 14C record from southern Oman based on the radiocarbon

analysis of tree rings.

The atmospheric radiocarbon activity showed anomalous enrichment during the early to mid-1960s, which is well documented in various atmospheric measurements as well as observed in several tree rings based proxy records. The authors have made high frequency sampling of a baobab tree during the bomb peak interval in order to study the nature of the 14C variability and the underlying mechanism that caused the observed variability. One of the main observations of their analysis is that the 14C variability in this region is characterised by a significant low value (ca 9%) compared to the expected value across the similar latitude belt. The authors opine that the internal cause, such as plant physiological processes are primarily responsible for this depletion. Apparently, they ignore the external factors, such as the fossil fuel dilution of atmospheric 14C variability, which may also produce such kind of anomalous signal. I would suggest the authors to discuss this aspect as well, to systematically rule out this possibility before coming to a definitive conclusion. My concerns are detailed below.

Section 4.1 The authors observed 14C activity in their tree ring that was noticeably lower than the NH3 and SH3 around the bomb peak (1964-1967). The authors explain that the anomalously low values were driven by plant physiological activities, the carbohydrate turn over time. But this hypothesis suffers from some limitation, because such kind of low tree ring 14C activities has been reported by some investigators in the Asian region without invoking the tree physiological process. For example, Kikata et al. observed a bomb peak around D14C = 692‰ in Vietnam. Hua et al. (2000) found 694‰ in northern Thailand. Chakraborty et al. (1994) found 630‰ in an urban area in west India. Murphy et al. observed slightly higher value of 705‰ in central India, which was also supported by Chakraborty et al (2008)'s observation of 708‰ in another site in central India.

Some of these authors have attributed the lower value of atmospheric 14C activity in a specific region in terms of fossil fuel dilution. For example, Chakraborty et al. (1994) analysed a teak sample from a West Indian urban area and found a somewhat

low value of 630‰ but the same species of another teak sample obtained from a central Indian but forested environment showed a bomb peak of 708%. The lowering of bomb peak (approx. 11%) in the urban area was attributed to fossil fuel dilution of atmospheric 14C. Chakraborty et al. (2008) did not invoke the idea of tree physiological process in this case, though the possibility, in principle, may not be ruled out. But, the occurrence of two different 14C values in the same tree species at two different places seem to be driven by external factor(s) rather than the tree physiological processes.

There may be other reasons to doubt the tree physiological process affecting the tree ring 14C activity. The mechanism explained by the authors involves the incorporation of previous year's carbon that significantly affects the average age of the current year wood. If that be the case, then a similar effect should have been observed by other investigators. Hua et al. (2003) analyzed a Pinus Radiata tree sample collected at Armidale in New South Wells, Australia and found excellent agreement with the atmospheric observation for the period of 1952 to 1967. But these authors observed higher 14C values in their Armidale tree ring samples for the period of 1968-1975. Obviously, the increase in radiocarbon activity cannot be explained by tree physiological processes. So either an increase or a decrease in 14C activity is likely to be driven by external factors.

Using a numerical exercise and auto-correlation analysis of d13C data Slotta et al. estimate that approx 85% of the fast cycling carbon pool and 15% of slow cycling carbon pool are contributing the lower values of bomb peak. If this explanation is true, then this effect should have been manifested in the entire record the authors have reported, which is not apparent from their results. Rather, the authors admit that the baobab F14C values for 1945, 1952-1954, 1956 and 1957 are indeed higher than the calculated range. This observation casts doubt in their explanation of old carbon turn over mechanism in explaining the negative excursion of 14C activity during the bomb peak period.

There may be another explanation of lower 14C activity in this region. Cember (1989)

analysed coral 14C activity from across the Red Sea to estimate the gas exchange rate. Cember observed very high air-sea exchange process over the Red Sea region. If this process is also operative in the southern Oman region, which is not very far from the Red Sea, then a viable explanation of anomalous 14C activity in the atmosphere may be provided.

2.4 Radiocarbon dating The analytical description provided by the authors is not up to the mark. For example, radiocarbon dating requires 13C correction and age correction; there is no mention whether such kind of corrections has been done. The reporting of 14C activity, especially in case of sequential samples (tree ring, corals) is typically done in cap delta notation (D14C), but the authors have preferred normalised activity. For comparison purposes with the published records, the authors are suggested to report the 14C activity in cap delta notation. Finally, the error in 14C measurement should be mentioned in terms of cap delta as well as the corresponding temporal value.

Minor issues: Line 92: The rainfall amount and its isotopes, usually show weak inverse correlation. Pls provide reference for evidence of "strong" correlation.

Line 141: very heavy rainfall in a single day producing high negative d18O "due to amount effect" is not technically right. Many studies (Lawrence and Gedzelman, 1996; Gedzelman et al., 2003; Lawrence et al. 1998; Chakraborty et al., 2016; Xu et al., 2019), showed that extreme precipitation events such as cyclonic activities produced very low values of precipitation d18O.

Line 181: pls provide a zoomed figure for the 1962-63 record of bomb 14C.

Line 199: What are the reference materials used?

Line 202: Please provide the permil sign after 0.15 and 0.25.

Line 217: 'weakening' should be "weaken".

Line 251-252: How the interpretation of the F14C data was confirmed by visual and statistical comparison of the TRW chronology with precipitation data should be explained

in detail.

Line 255: 'shallow' should be replaced by "gentle".

Line 268: 'radiocarbon' should be followed by "analysis".

Line 275: 'considerably declining' meaning is not clear.

Line 280: What is the physical basis of getting a strong correlation between d18O and RWI? Also mentioned in Line 334. Please provide the value of correlation and state the sample number.

Line 309: the lag between cyclonic events and the corresponding d18Omin should be provided on a monthly time scale.

Line 355: 'extend' should be replaced by "extent".

Line 494: "evaporative enrichment in 18O...".Please provide supportive evidence of enhanced soil evaporation, say by means of observed or reanalysis data in support of this speculation.

Line 505: "Vapor pressure deficit ...18O enrichment in leaf water", and "lower stomatal conductance...13C discrimination to decline" require supporting literature.

Line 514: the authors argue that the decline in d18O...might be due to the previous year's October precipitation. If so, then d18O is also expected to be auto-correlated.

Line 524: likely in 'would have likely..." should be deleted.

References:

Cember 1989 Bom radiocarbon in the Red Sea: a medium scales gas exchange experiment. JGR Ocean 94:2111-2123.

Lawrence, J. L. & Gedzelman, S. D. Low stable isotope ratios of tropical cyclone rains. Geophys. Res. Lett. 23, 527–530 (1996).

Gedzelman, S., Lawrence, J., Gamache, J., Black, M., Hindman, E., Black, R., Dunion, J., Willoughby, H., Zhang, X., 2003. Probing hurricanes with stable isotopes of rain and water vapor. Mon. Weather. Rev. 131 (6), 1112–1127.

Hua, Q., Barbetti, M., Zoppi, U., Chapman, D. M., and Thomson, B. 2003 Bomb Radiocarbon in Tree Rings from Northern New South Wales, Australia: Implications for Dendrochronology, Atmospheric Transport, and Air-Sea Exchange of CO2, Radiocarbon, 45, 431-447.

Xu et al. 2019 Stable isotope ratios of typhoon rains in Fuzhou, Southeast China, during 2013–2017.

Kikata Y, Yonenobu H, Morishita F, Hattori Y. 1992. 14C concentrations in tree stems. Bulletin of the Nagoya University Furukawa Museum 8:41–6. In Japanese.

---

## Short Comment (SC2) · 30 Oct 2019

I found the article very interesting. However, I have some comments and questions regarding the data and discussions.

The paper begins by pointing out the lack of atmospheric radiocarbon (14C) datasets that defined the inter- and intra- hemispheric division zones closer to the equatorial line (NH3 and SH3; Hua et al . 2013). Therefore, building a new dataset appeared to be the main motivation, which the paper wishes to address or attempt to fill in. But as

soon as the newly produced 14C data based on the baobab tree rings did not match with the average curve used as benchmark for zone NH3, an alternative explanation was offered. Mixed non-structural carbon (NSC) pools incorporation in the structural ring cellulose fraction - a new tree species functional trait - Maybe (?!).

I appreciate that in view of the perplexing results of the 14C data of the baobab tree rings, an alternative explanation should be considered. But for the mixed NSC-ring cellulose hypothesis works, all other possible bias must be ruled out, and a through-out discussion on the scarcity of the previous records across this zones (NH3 and SH3), and the possibility of multiple sources of air-14CO2 influencing this particular site should be offered. We cannot ignore the fact that the original atmospheric 14C records across NH3 and SH3 are quite incomplete, temporally and spatially.

The stable isotope measurements, although very complex, gave insights of tree water usage. Overall it seems to indicate the tree was not water limited. This brings us to the second issue. Why the baobab tree would incorporate constants amounts of slow turnover NSC into its ring cellulose structural carbon fraction year-after-year, regardless of the environment stress conditions surrounding it? Richardson et al. (2013) stated that even though they found very old pools of starch and sugars in aboveground temperate forest trees, stressed trees would still use up first all available present-day fast cycling carbon pool to support growth and metabolism. This would include even the most recently added starch molecules. Therefore, the usage of "older" NSC reserves was set for times of stress. I think it will be important to make this distinction in the text. Richardson et al. (2013) did not mentioned that ring cellulose 14C results were off from expected values after direct comparison with the northern hemisphere atmospheric record, just the NSCs extracts (sugars and starches) were.

In this article, the baobab tree ring cellulose extracts 14C results are unusually off from its expected zonal averaged record or records (if direct comparisons are done to independent datasets). If a constant slow turnover NSC incorporation to ring cellulose is to blame, is the article implying that this is a functional trait for all parenchymarich succulent species?! If so, it would be imperative to test other parenchyma-rich succulent trees before even suggesting a physiological effect. Additionally, if a novel physiological effect has been found, it would be important to distinguish it from non-parenchyma-rich succulent species. This trait would not necessarily be mimicked by another tree species elsewhere, so that the use of tree rings as a proxy of atmospheric 14C would still be valid. This point should be made clear, so that the reader(s) can notice the difference. Note that a large percentage of the data in the NH3 and SH3 zones were based on tree ring data.

Regarding reliability of the data produced, the article mentioned that:

1) "To avoid carry-over effects from previous years' NSC into the current year's wood cellulose, the samples for dating were selected from the last third of each growth structure while steering clear of the terminal parenchyma band (TPB) by at least one sample, where possible". While is important do not include material from the neighboring rings, losing material from the actual growing season should also affect the 14C results. So, is the statement in double quotation marks correct?! Just the last fraction of the full growing season per calendar year was selected for 14C measurements?! This is relevant and should be explained.

2) Hollocellulose and alpha-cellulose extractions have been widely used for isolating of the structural carbon fraction of tree rings for 14C analysis, with alpha-cellulose being considered superior then holocellulose in some cases. Here, it is stated that comparisons between hollocellulose vs alpha-cellulose extractions were tested on 10 samples/results, and that they were indistinguishable within 2-3 sigma of each other. Thus, holocellulose was adopted as the main chemical extraction procedure. However, these results were not clearly indicated as well as the calendar years from where they belong. This information is relevant, as the calendar years belonging to the steep slopes of the bomb-peak would be more sensitive to unremoved labile NSC (if any) affecting 14C results. Second, I am sure that the laboratories that performed the analyses run a quality control and quality assurance program based on combustion/graphitization of

reference materials. However, there is no mentioning on reproducibility and accuracy of any present-day wood-control sample undergoing the holocellulose or alpha-cellulose procedures mentioned here, so that exogenous contamination of any sort from the full procedure could be rejected. Moreover, to corroborate the results found here an interlaboratory crosscheck of fewer tree rings would be crucial.

References cited here and also in the original article:

Hua, Q., Barbetti, M., and Rakowski, A. Z.: Atmospheric Radiocarbon for the Period 1950–2010, Radiocarbon, 55, 2059-2072, doi:10.2458/azu_js_rc.v55i2.16177, 2013

Richardson, A. D., Carbone, M. S., Keenan, T. F., Czimczik, C. I., Hollinger, D. Y., Murakami, P., Schaberg, P. G., and Xu, X.: Seasonal dynamics and age of stemwood nonstructural carbohydrates in temperate forest trees, New Phytologist, 197, 850-861, doi:10.1111/nph.12042, 2013

---

## Referee Comment (RC1) · Guaciara Santos (Referee) · 1 Feb 2020

Associate Editor, Dr Aninda Mazumdar, asked me to act as a Referee of this article, although I had already placed a short comment earlier. I am happy to be identified as a reviewer and have my comments passed on to the authors. Therefore, I am expending my previous short comment here. I hope that these further comments and suggestions will help the authors to improve their work.

General comments

[Figure]

The paper presents annual 14C data from an African baobab (Adansonia digitata) tree from Oman, for the interval AD 1941 to 2005. This work is important in that it provides a fairly detailed pre/post-bomb 14C time-series for a region that has not yet being part of the atmospheric 14C global compilation. This is actually one of the main goals of the manuscript. The authors have also improved the quality of the data set by providing intra-annual analyses of $\delta$13C and $\delta$18O, as well as F14C for the calendar years of 1962 and 1963.

While the high number of consecutive single tree-rings measured by radiocarbon allowed confirming the annual nature of the baobab species, a significant mismatch between the baobab F14C values and the post-bomb atmospheric curve NH3 was detected. This mismatch prompted an alternative explanation, i.e. mixed pool of slow-turnover non-structural carbon (NSC) into the structural ring cellulose fraction - a strong functional trait of parenchyma-rich tree species (maybe ?!).

The Baobab terminal parenchyma bands F14C values presented here, definitely demonstrate that a large percent of the parenchyma in this tree species is relatively young, and as such, it provides valuable perspectives in the field of plant physiology. On the other hand, mixed carbon pools in putative structural ring cellulose fraction (in this case, slow-turnover NSC residue in holocellulose extracts) put into question the use of tree rings of this group of woody plant when reconstructing atmospheric 14C.

I appreciate that in view of the perplexing results of the 14C data of the baobab tree rings, an alternative explanation should be considered. However, for the mixed pool NSC-ring cellulose assumption works, all other possible bias must be carefully ruled out. Robust methodologies must be properly done and explained in detail, as well as the use of reference materials/internal standard, or equivalent (i.e. interlaboratory measurements), and the use of further chemical extractions. All of those are missing here.

Given the absence of an independent benchmark, e.g. a short F14C sequence of con-

secutive single tree-rings from a non-parenchyma-rich woody plant in Oman, I cannot tell whether slow-turnover NSC detected in holocellulose extracts of baobab tree rings is a feasible explanation for the 14C offset observed here or not. For starters, 14C analysis of incomplete single tree rings (material that do not represent a full growing season) could contribute in 14C offsets (see specific comments/suggestions). Furthermore, we had to keep in view that other factors must also play some role in those 14C offsets (atmospheric circulation and carbon dioxide from human activities, for example). Previous records across zones NH3, SH3 and SH1-2 are very scarce. Therefore, the possibility of multiple sources of air-14CO2 influencing Oman should be discussed. One cannot ignore the fact that during the assembly of the atmospheric post-AD 1950 14C global compilation by Hua et al. (2013) some datasets were disregarded due to its mismatches with other regional datasets. Therefore, a thorough evaluation of possible external effects should also be offered.

Finally, procedures described here need further explanations and details. The result and discussion part is quite jumpy and very tricky to follow. It does not quite convey the ideas of the underlying assumption offered to explain the baobab tree F14C offsets. I recommend a complete re-organization of the manuscript, by focusing on placing the absolutely necessary data, figures and tables (for the purpose of the paper) in the main text. The stable isotope findings were not particularly striking. Although important, they are currently creating a lot of distraction. I strongly suggest moving them (most of its description, associated material and discussions) to a supplementary text or appendix.

Specific Comments/Suggestions

I am going to focus here on just major topics that are in need of clarification to verify the fitness of the data shown.

- p4, l111. It is stated that 10 trees were sampled by increment cores from four different orientations (NE, SE, SW, and NW). Do you mean four radii were collected per tree?! If yes, random tree rings were used for 14C analysis or just one tree and radii's? Please,

clarify.

- p5, l148 to l55. How the tree specimen selected was dendrochronologically-secure? How the chronosequence of tree rings (prior 14C dating) was obtained without a master chronology for Baboab species?! The passage selected here describes just figure 2. Later (at p.6, l190 to 194), it is explained that no dated tree-ring width chronology from the study region is currently available. Therefore to anchor the chronosequence of tree rings (prior counting of all baobab tree rings) the F14C of the TPBs and Oxcal was used instead. Is this correct?!... If yes, this explanation should appear early on in the text. The fitness of the chronosequence is the backbone of the atmospheric 14C record production using tree rings. Plus, add what type of juniper species you are referring to.

- p5&6, l159 to l65, and l177 to l178. Passages explaining the wood material used for radiocarbon and stable isotope analysis are confusing, and very troubling. It appears that the full dataset was produced in two phases, a pre-screening phase with 5 calendar years or so, where just 1/3 of the tree ring (cut parallel to the fiber orientation, in radial direction from the cambial zone) was used. In a second phase, in order to measure the remaining calendar years, just 2/3 of the single tree ring was used for 14C dating. The remaining material was then used for 13C. This description gives the idea that the tree ring cutting for isotopic analysis was selective, before chemical extractions took place.

Normally a homogenized cellulose-extract of a full single whole-ring (from early- to late-wood) is used to reconstruct atmospheric 14C data. It is understandable that since the baobab contains 69-88 % parenchyma cells, mostly concentrated at the terminal parenchyma bands (TPBs) or late-wood, this portion was removed. But if the remaining material was further sub-divided by removing wood material representative of the growth season (Figure 2), unexpected 14C offsets would then be expected, especially at the slopes of the bomb peak. Accurate cutting of the tree rings is paramount for the reconstruction of atmospheric 14C data. This was already demonstrated by the intra-annual analyses of F14C for the calendar years of 1962 and 1963 shown here. Moreover, if the wood cutting was indeed selective (prior chemical extractions,

as mentioned above), I do not understand how the ms can assert at the abstract that "considerable autocorrelation was found in the d13C series, confirming incorporation of previous years' carbon significantly affecting the average age of derived wood", if the wood material tested was not the same. Analyses of $\delta$13C, $\delta$18O, as well as 14C should be done from homogenized cellulose-extracts from the same wood aliquots. Please, clarify.

-p5, l164 & 165. Some of the TPBs removed were selected for 14C dating in phase one (4 or 5 samples). There is no mentioning of the chemical treatment they were subjected to prior sample processing for 14C-AMS. Please, explain...

-p.6, l182 to 190. This portion is very confusing. The TPBs F14C and OxCal were used to anchor the chronosequence. This would give a general idea of the calendar ages of these tree rings, which is ok. But since no chemical extraction appear to have being applied to TPB samples (no description of such is offered), I do not understand why one should expect that they would match with the NH3. Please, rephrase statements.

Regarding figure 3, and text portion between l187 to 190. What do you mean w/ "the baobab samples' position on the time axis is relative to their position within the tree ring of a growing season lasting from June until September"? Were the calendar years in the "x-axis" of figures 3 (and figure 8, as well) adjusted to match w/ the growing season of the baobab species as shown in Fig. 1C (June to September)? It is hard to see if such adjustment was applied in figure 3, as the figure is small. But I think that this adjustment was not applied to figure 8, as it should, and therefore the entire baobab F14C values are too far to the left. Have you take this monthly shift in account in the modelling as well? This calendar year adjustment should also appear at Table 2, second column to avoid confusing between growth date and dendro-date.

On figure 3A, I am left unsure (without checking all records in Hua et al. 2013 supplementary material) the main differences in uncertainties between SH3, SH1-2 and NH3 records beyond about 1972 (orange shaded area). Why is this shaded area particularly

different from all others, when the SH3 record (based on Muna Island data) stopped in 1979? Beyond this calendar year most records assume no differences between hemispheres due to scarcity of data in the tropics. Please, explain.

Figure 3B, I appreciate the effort of showing F14C values between the calendar year of 1962 to 1964, but further discussions on air mass circulation (as mentioned earlier) are still lacking. Since the citation of Nydal & Lovseth 1983 is already listed in the article, all other records in the same zonal band in this article should be added to the plot. Second, most of the citations in this figure legend are not in the reference list. Third, replace Turnball et al. 2017 by Turnbull et al. 2017.

-p.9, section 3.1. I do not understand why one should expect that the TPbs would match with the NH3, or even match with the TRs (holocellulose extracts, Table 2). I don't see how this part is relevant. Most importantly would be comparisons between F14C data of TRs and alternative alpha-cellulose treatments that target the removal of starches and sugars (e.g., "Soxhlet"-type extractions using solvents). Note that the alpha-cellulose extraction described here was attained by adding an extra step of 17.5% NaOH to the holocellulose procedure. Incomplete removal of resinous compounds during chemical pretreatment of tree rings biasing 14C data has been shown by others (Cain and Suess 1976, Westbrook et al. 2006, for example).

-p12, section 4.1. I found this section highly speculative; especially when no 14C dating targeting starch extracts from the baobab parenchyma-dominated wood was attempted. Richardson et al. (2013), cited in this section, indeed found direct evidence for 'fast' and 'slow' cycling reserves in stemwood. However, Richardson et al. (2013) also stated that even though aboveground temperate forest trees contained very old pools of starch and sugars, stressed trees would still use up first all available present-day fast cycling carbon pool to support growth and metabolism. This would include even the most recently added starch molecules. Therefore, the usage of "older" NSC reserves was set for times of stress. Richardson et al. (2013) did not mentioned that ring cellulose 14C results were inaccurate after direct comparison with the northern

hemisphere atmospheric record, just the NSCs extracts (sugars and starches) were. Those compound fractions were chemically extracted separately by standard protocols for the purpose of 14C dating. I think it will be important to make this distinction in the text to avoid misleading the reader.

While I do not think that it is completely impossible that the baobab tree species incorporate slow-turnover NSC into its ring cellulose structural carbon fraction year-after-year (regardless of the environment stress conditions surrounding it), I think that the ms fail to: 1) clearly demonstrate this phenomenon, 2) properly justify it, and discuss external bias for the 14C offsets. A short sequence of a non-parenchyma-dominated wood chronosequence of tree rings dated by 14C bomb peak from this region, such as the juniper species mentioned in text, should resolve most (if not all ) the issues raised here. Therefore, I cannot see how this manuscript can be published without major revisions, or even further demonstrations.

Reference list cited here:

Cain and Suess 1976. Journal of Geophysical Research 81(21):3688 Hua et al. 2013. Journal of Geophysical Research: Oceans 88(C6): 3621. Richardson et al. 2013. New Phytologist, 197: 850. Turnbull et al. 2017. Atmospheric Chemistry and Physics 17(2). Westbrook et al. 2006, IAWA Journal 27(2): 193
* * *

---

## Referee Comment (RC2) · Supriyo Chakraborty (Referee) · 12 Feb 2020

Radiocarbon analysis of the annual rings of trees has been carried out by several investigators to study a variety of natural processes. Such kinds of records, especially in the extra-tropical region of the northern hemisphere are widely available. The tropical region, however, is not well represented. To fill this gap Slotta et al. attempted to reconstruct atmospheric 14C records from southern Oman based on the radiocarbon analysis of tree rings. The atmospheric radiocarbon activity showed anomalous enrichment during the early to mid-1960s, which is well documented in various atmospheric

measurements as well as observed in several tree ring-based proxy records. The authors have made a high-frequency sampling of a baobab tree during the bomb peak interval in order to study the nature of the 14C variability and the underlying mechanism that caused the observed variability. One of the main observations of their analysis is that the 14C variability in this region is characterized by a significantly low value (ca 9%) compared to the expected value across the similar latitudinal belt. The authors opine that the internal cause, such as plant physiological processes are primarily responsible for this depletion. Apparently, they ignore the external factors, such as the fossil fuel dilution of atmospheric 14C variability, which may also produce such kind of anomalous signal. I would suggest the authors discuss this aspect as well to systematically rule out this possibility before coming to a definitive conclusion. My concerns are given below.

Section 4.1 The authors observed 14C activity in their tree ring that was noticeably lower than the NH3 and SH3 around the bomb peak (1964-1967). The authors explain that the anomalously low values were driven by plant physiological activities, the carbohydrate turn over time. But this hypothesis suffers from some limitations because such kind of low tree ring 14C activities has been reported by some investigators in the Asian region without invoking the tree physiological process. For example, Kikata et al. observed a bomb peak around D14C = 692‰ in Vietnam. Hua et al. (2000) found 694‰ in northern Thailand. Chakraborty et al. (1994) found 630‰ in an urban area in west India. Murphy et al. observed a slightly higher value of 705‰ in central India, which was also supported by Chakraborty et al (2008)'s observation of 708‰ in another site in central India.

Some of these authors have attributed the lower value of atmospheric 14C activity in a specific region in terms of fossil fuel dilution. For example, Chakraborty et al. (1994) analyzed a teak sample from a western Indian urban area and found a somewhat low value of 630‰ but the same species of another teak sample obtained from a central Indian but forested environment showed a bomb peak of 708%. The lowering

of bomb peak (approx. 11%) in the urban area was attributed to fossil fuel dilution of atmospheric 14C. Chakraborty et al. (2008) did not invoke the idea of tree physiological process in this case, though the possibility, in principle, may not be ruled out. But, the occurrence of two different 14C values in the same tree species at two different places seem to be driven by an external factor(s) rather than the tree physiological processes.

There may be other reasons to doubt the tree physiological process affecting the tree ring 14C activity. The mechanism explained by the authors involves the incorporation of previous year's carbon that significantly affects the average age of the current year wood. If that be the case, then a similar effect should have been observed by other investigators. Hua et al. (2003) analyzed a Pinus Radiata tree sample collected at Armidale in New South Wells, Australia and found excellent agreement with the atmospheric observation for the period of 1952 to 1967. But these authors observed higher 14C values in their Armidale tree ring samples for the period of 1968-1975. Obviously, an increase in radiocarbon activity cannot be explained by tree physiological processes. So either an increase or a decrease in 14C activity is likely to be driven by external factors.

Using a numerical exercise and auto-correlation analysis of d13C data Slotta et al. estimate that approx 85% of fast cycling carbon pools and 15% of slow-cycling carbon pools are contributing to the lower values of the bomb peak. If this explanation is true, then this effect should have been manifested in the entire record the authors have reported, which is not apparent from their results. Rather, the authors admit that the baobab F14C values for 1945, 1952-1954, 1956 and 1957 are indeed higher than the calculated range. This observation casts doubt in their explanation of the old carbon turn over mechanism in explaining the negative excursion of 14C activity during the bomb peak period. There may be another explanation of lower 14C activity in this region. Cember (1989) analyzed coral 14C from across the Red Sea to estimate the gas exchange rate. Cember observed a very high air-sea exchange process over the Red Sea region. If this process is also operative in this region which is not very far from

the Red Sea, then a viable explanation of anomalous 14C activity in the atmosphere may be provided.

2.4 Radiocarbon dating The analytical description provided by the authors is not up to the mark. For example, radiocarbon dating requires 13C correction and age correction; there is no mention of whether such kind of corrections has been done. The reporting of 14C activity, especially in the case of sequential samples (tree ring, corals) is typically done in cap delta notation (D14C), but the authors have preferred normalized activity. For comparison purposes with the published records, the authors are suggested to report the 14C activity in cap delta notation. Finally, the error in 14C measurement should be mentioned in terms of cap delta as well as the corresponding temporal value.

Minor issues: Line 92: The rainfall amount and its isotopes usually show a weak inverse correlation. Pls, provide reference for evidence of "strong" correlation.

Line 141: very heavy rainfall in a single day producing high negative d18O "due to amount effect" is not technically right. Many studies (Lawrence and Gedzelman, 1996; Gedzelman et al., 2003; Lawrence et al. 1998; Chakraborty et al., 2016; Xu et al., 2019), showed that extreme precipitation events such as cyclonic activities produced very low values of precipitation d18O.

Line 181: pls provide a zoomed figure for the 1962-63 record of bomb 14C.

Line 199: What are the reference materials used?

Line 202: Please provide the permil sign after 0.15 and 0.25.

Line 217: 'weakening' should be "weaken".

Line 251-252: How the interpretation of the F14C data was confirmed by visual and statistical comparison of the TRW chronology with precipitation data should be explained in detail.

Line 255: 'shallow' should be replaced by "gentle".

Line 268: 'radiocarbon' should be followed by "analysis".

Line 275: 'considerably declining' meaning is not clear.

Line 280: What is the physical basis of getting a strong correlation between d18O and RWI? Also mentioned in Line 334. Please provide the value of correlation and state the sample number.

Line 309: the lag between cyclonic events and the corresponding d18Omin should be provided on a monthly time scale.

Line 355: 'extend' should be replaced by "extent".

Line 494: "evaporative enrichment in 18O...".Please provide supportive evidence of enhanced soil evaporation, say by means of observed or reanalysis data in support of this speculation.

Line 505: "Vapor pressure deficit ...18O enrichment in leaf water", and "lower stomatal conductance...13C discrimination to decline" require supporting literature.

Line 514: the authors argue that the decline in d18O...might be due to the previous year's October precipitation. If so, then d18O is also expected to be auto-correlated.

Line 524: likely in 'would have likely..." should be deleted.

References:

Cember 1989 Bom radiocarbon in the Red Sea: a medium scales gas exchange experiment. JGR Ocean 94:2111-2123.

Lawrence, J. L. & Gedzelman, S. D. Low stable isotope ratios of tropical cyclone rains. Geophys. Res. Lett. 23, 527–530 (1996). Gedzelman, S., Lawrence, J., Gamache, J., Black, M., Hindman, E., Black, R., Dunion, J., Willoughby, H., Zhang, X., 2003. Probing hurricanes with stable isotopes of rain and water vapor. Mon. Weather. Rev. 131 (6), 1112–1127.

Hua, Q., Barbetti, M., Zoppi, U., Chapman, D. M., and Thomson, B. 2003 Bomb Radiocarbon in Tree Rings from Northern New South Wales, Australia: Implications for Dendrochronology, Atmospheric Transport, and Air-Sea Exchange of $CO_2$, Radiocarbon, 45, 431-447.

Xu et al. 2019 Stable isotope ratios of typhoon rains in Fuzhou, Southeast China, during 2013–2017.

Kikata Y, Yonenobu H, Morishita F, Hattori Y. 1992. 14C concentrations in tree stems. Bulletin of the Nagoya University Furukawa Museum 8:41–6. In Japanese.

---

## Author Comment (AC2) · 6 Nov 2020

Dear Dr. Mazumdar, dear referees,

We gratefully acknowledge the efforts of Dr. Santos and Dr. Chakraborty for their comprehensive reviews of our manuscript. Their questions, criticism and thoughtful comments are very appreciated. In this response – prior to the editor's decision - we provide additional input and perspective that make some of the referees' concerns less pressing and may dispel others altogether. In the revised MS minor comments will

thoroughly be considered, grammar and spelling mistakes corrected.

The most critical comments of the referees can be summarized three key points: 1) our paper does not discuss in sufficient detail all potential causes of the observed F14C values of tree-ring cellulose being significantly lower than in the corresponding atmospheric $CO_2$ for the period around the bomb peak. Referees question our arguments for significant tree-physiological causes and particularly criticize the lack of discussion of external reasons. 2) our methodological and analytical description is lacking detail, unclear, ambiguous or not up to the mark. 3) The stable isotope data, our analysis of climate-proxy relationships and the climatological interpretation has raised almost no interest of the referees at all.

If given the opportunity by the editor to revise our MS we will particularly focus on addressing these three issues. We will consider and incorporate the literature suggested by the referees. In view of this we will carefully re-write our methods section exemplifying that sample preparation and mass spectrometric analyses (IRMS and AMS) were performed in compliance with international standards and good scientific practice. In the discussion we will be taking into special consideration the potential external causes of 14C change during the period of the bomb peak (e.g. fossil fuel burning, atmospheric circulation etc. as put forward by the referees). In the following, we outline the key changes we intend to incorporate into the revised manuscript to consider the individual key points raised by the referees.

Comments to Referee #1: General remarks

The paper presents annual 14C data from an African baobab (Adansonia digitata) tree from Oman, for the interval AD 1941 to 2005. This work is important in that it provides a fairly detailed pre/post-bomb 14C time-series for a region that has not yet being part of the atmospheric 14C global compilation. This is actually one of the main goals of the manuscript. The authors have also improved the quality of the data set by providing intra-annual analyses of $\delta$13C and $\delta$18O, as well as F14C for the calendar years of 1962

and 1963. Response: We thank the reviewer for considering this work as important for a region that is underrepresented in terms of 14C data.

While the high number of consecutive single tree-rings measured by radiocarbon allowed confirming the annual nature of the baobab species, a significant mismatch between the baobab F14C values and the post-bomb atmospheric curve NH3 was detected. This mismatch prompted an alternative explanation, i.e. mixed pool of slow-turnover non-structural carbon (NSC) into the structural ring cellulose fraction - a strong functional trait of parenchyma-rich tree species (maybe ?!). Response: We agree with this summarizing statement of our observations. However, we do not just explain the aberrant F14C values by incorporation of carbon from a mixed pool of slow and fast turnover of NSC. In the MS we propose an additional potential cause: namely the huge difference in longevity of wood forming plant cells: while parenchyma cells can live up to approx. 20 years, wood fibres live up only from a few weeks to a season. This means that parenchyma tissue can undergo changes in its 14C for over several years, whereas the 14C of fibre tissue is always assigned to a certain year. Since in baobabs parenchyma occurs not only as bands but is also diffusely distributed within a tree ring, varying proportions of parenchyma and fibers can cause variations in F14C of a tree ring that can be, to a certain extent, unrelated to the specific date of the tree ring. Please note, that baobabs are unique in this regard. To our knowledge, no other tree species shows similarly high parenchyma contents than baobabs (69-88%). Changes in the manuscript: We will rewrite introduction and discussion to make these two tree physiological aspects clearer. In addition, we will discuss in more detail potential external casues of the observed 14C trends (see responses to further comments below).

Comment: The Baobab terminal parenchyma bands F14C values presented here, definitely demonstrate that a large percent of the parenchyma in this tree species is relatively young, and as such, it provides valuable perspectives in the field of plant physiology. On the other hand, mixed carbon pools in putative structural ring cellulose fraction (in this case, slow-turnover NSC residue in holocellulose extracts) put into

question the use of tree rings of this group of woody plant when reconstructing atmospheric 14C. Response: Thank you for supporting our conclusion that our F14F data points to future perspectives in plant physiological research (in particular on baobabs, which are widely distributed in Africa and potentially threatened by global change). Our data set contributes a fairly detailed pre/post-bomb 14C time-series for a region that has not yet being part of the atmospheric 14C global compilation. However, it was NOT the main purpose of our MS to reconstruct atmospheric 14C from this data set. As written in the introduction, we primarily intended to use the 14C bomb peak to validate the counting/dating of tree rings. As mentioned by the referee above, the unexpected significant mismatch between the baobab F14C values and the post-bomb atmospheric curve NH3 prompted for some reasonable interpretation. In this case we suggested and still suggest that it is from a mixed carbon pool in conjunction with the extraordinary high content and longevity of parenchyma tissue relative to short lived fibre tissue that constitute the tree rings of baobabs. Nonetheless, the referee is right. Tree species with such a high content of parenchyma should not be used for reconstructing atmospheric14C. This raises particular issue for tropical regions, where tree angiosperm species show about 36% of parenchyma on average. In contrast, angiosperm tree species in temperate zones have a content of about 21%, only. Changes in the manuscript: We will rewrite introduction to clarify the original purpose of our 14C analyses and we will add and detail the aspects outlined above in the discussion of the revised MS.

Comment: I appreciate that in view of the perplexing results of the 14C data of the baobab tree rings, an alternative explanation should be considered. However, for the mixed pool NSC-ring cellulose assumption works, all other possible bias must be carefully ruled out. Response: Thank you very much for this valuable comment. This point, i.e. other potential causes for the observed bias, has also been raised by referee #2. Changes in the manuscript: As suggested, we will add a paragraph tackling other, external effects on atmospheric 14C to the MS and also rephrase the parts in the manuscript referring to this.

Comment: Robust methodologies must be properly done and explained in detail, as well as the use of reference materials/internal standard, or equivalent (i.e. interlaboratory measurements), and the use of further chemical extractions. All of those are missing here. Response: Thank you very much for pointing to the lack of description of our methodology. This point has partly also been raised by referee #2. Changes to the manuscript: As suggested, we will carefully rewrite the methods section and explain in more detail the process of tree ring dissection, cellulose extraction and mass spectrometric analyses.

Comment: Given the absence of an independent benchmark, e.g. a short F14C sequence of consecutive single tree-rings from a non-parenchyma-rich woody plant in Oman, I cannot tell whether slow-turnover NSC detected in holocellulose extracts of baobab tree rings is a feasible explanation for the 14C offset observed here or not. For starters, 14C analysis of incomplete single tree rings (material that do not represent a full growing season) could contribute in 14C offsets (see specific comments/suggestions). Response: Good point. Unfortunately, project resources were limited and did not allow analyses of other tree species than baobabs. 14C analyses were done on the 2/3 of a tree ring. The last 1/3 including each terminal parenchyma band was discarded in order to minimize negative effects from long living parenchyma tissue. Since the transition from wood to the terminal parenchyma band cannot be determined precisely, we decided to also skip some of the wood. We believe that contamination from parenchyma causes larger bias than the seasonal effects. Baobab tree rings in Oman are largely formed between May and October each year; our samples from may represent the growing period from May to August or early September. Changes to the MS: We will exemplify our sampling strategy in more detail.

Comment: Furthermore, we had to keep in view that other factors must also play some role in those 14C offsets (atmospheric circulation and carbon dioxide from human activities, for exam- ple). Previous records across zones NH3, SH3 and SH1-2 are very scarce. Therefore, the possibility of multiple sources of air-14CO2 influencing Oman

should be discussed. Response: Yes, we admit that these factors have not been discussed in detail. This point has been raised by referee #2 as well. Changes to the MS: We will carefully add and discuss these aspects.

Comment: One cannot ignore the fact that during the assembly of the atmospheric post-AD 1950 14C global compilation by Hua et al. (2013) some datasets were disregarded due to its mismatches with other regional datasets. Therefore, a thorough evaluation of possible external effects should also be offered. Response: Yes, once more, we admit that these factors have not been discussed in detail. Changes to the MS: We will carefully add and discuss these aspects.

Comment: Finally, procedures described here need further explanations and details. Response: Yes, thank you for this comment. Changes to the MS: We will exemplify all our procedures in much more detail.

Comment: The result and discussion part is quite jumpy and very tricky to follow. It does not quite convey the ideas of the underlying assumption offered to explain the baobab tree F14C offsets. I recommend a complete re-organization of the manuscript, by focusing on placing the absolutely necessary data, figures and tables (for the purpose of the paper) in the main text. Response: Thank you for this comment. Apparently, there is some misunderstanding concerning the purpose of this paper. It is not ment to present a reconstruction of atmospheric 14C (see earlier response comments above).

Changes to the MS: We will rephrase the MS to make our intentions much clearer.

The stable isotope findings were not particularly striking. Although important, they are currently creating a lot of distraction. I strongly suggest moving them (most of its description, associated material and discussions) to a supplementary text or appendix. Response: The intra-annual stable isotope data, its climate related analysis and interpretation is of valuable interest to the dendro- and palaeclimate community, which may not be the major audience of BG and the referee of this MS. In particular, stable isotope data points to distinct relations between intra-seasonal stable isotope patterns and pre- and post monsoon cyclones. Furthermore, tree-ring stable isotope records from regions like Oman are still scarce and knowledge about the climatic significance of baobab tree-ring parameters is important. According to her expertise, the referee feels distracted while focusing on the radiocarbon part, only. Note, there seems to be some misunderstanding concerning the purpose of this paper, which was not ment to present a reconstruction of atmospheric 14C (see response comments above) Changes to the MS: We do not intend to make changes to the MS in this regard. However, if advised by the editor, we may remove all stable isotope related aspects from the MS and change the title as follows: "High resolution 14C bomb-peak dating of tree rings of an African baobab from Oman"

Specific Comments/Suggestions I am going to focus here on just major topics that are in need of clarification to verify the fitness of the data shown. - p4, l111. It is stated that 10 trees were sampled by increment cores from four different orientations (NE, SE, SW, and NW). Do you mean four radii were collected per tree?! If yes, random tree rings were used for 14C analysis or just one tree and radii's? Please, clarify. Response: One tree, and one radii for 14C and stable isotope analysis Changes to the MS: We will be more specific the revised MS.

- p5, l148 to l55. How the tree specimen selected was dendrochronologically-secure? How the chronosequence of tree rings (prior 14C dating) was obtained without a master chronology for Baboab species?! The passage selected here describes just figure 2. Later (at p.6, l190 to 194), it is explained that no dated tree-ring width chronology from the study region is currently available. Response: One tree, and one radii. No chronology from several trees has been developed for this case study and no chronology from any other tree species from this site or region has been published. Changes to the MS: This will be added to the text.

- Therefore to anchor the chronosequence of tree rings (prior counting of all baobab tree rings) the F14C of the TPBs and Oxcal was used instead. Is this correct?!...

Response: No. One tree, and one radii was analysed. No chronology from several trees has been developed for this case study. Tree rings were counted and 14C data was used to validate the ring count. Changes to the MS: This will be added to the text. - If yes, this explanation should appear early on in the text. The fitness of the chronosequence is the backbone of the atmospheric 14C record production using tree rings. Response: One tree, and one radii. No chronology from several trees has been developed for this case study. Tree rings were counted and 14C data was used to validate the ring count. Changes to the MS: This will be added to the text.

- Plus, add what type of juniper species you are referring to. Response: Juniperus seravschanica Changes to the MS: This will be added to the text.

- p5&6, l159 to l65, and l177 to l178. Passages explaining the wood material used for radiocarbon and stable isotope analysis are confusing, and very troubling. It appears that the full dataset was produced in two phases, a pre-screening phase with 5 calendar years or so, where just 1/3 of the tree ring (cut parallel to the fiber orientation, in radial direction from the cambial zone) was used. In a second phase, in order to measure the remaining calendar years, just 2/3 of the single tree ring was used for 14C dating. The remaining material was then used for 13C. This description gives the idea that the tree ring cutting for isotopic analysis was selective, before chemical extractions took place. Response: As mentioned above, the tree rings of the baobab sample were counted. Since missing rings and miscounting was expected 14C bomb peak wiggle matching should helb to validate the ring counting. In the first phase 5 individual tree rings covering individual, not consecutive years of the bomb peak period (as defined by ring counting) were selected for narrowing down the period. Changes to the MS: this will be exemplified in more detail in the revised text.

Normally a homogenized cellulose-extract of a full single whole-ring (from early- to late-wood) is used to reconstruct atmospheric 14C data. Response: Cellulose has been extracted for all samples of this study. Here, all tree-ring material, except from terminal parenchyma bands has been used for a 14C analysis. Changes to the MS: This will be

added to the text.

It is understandable that since the baobab contains 69-88 % parenchyma cells, mostly concentrated at the terminal parenchyma bands (TPBs) or late-wood, this portion was removed. Response: Yes, parenchyma bands (TPBs) were seperated from tree-ring material prior to 14C and stable isotope analysis. But, no further selective sampling was performed for 14C analysis. Tree rings of 1961-1963 and 2005 were sub-divided, but all sub-division were then analysed (see fig. 8) Changes to the MS: This will be clarified in the text and figs. 3 a,b will be modified to make this easier to comprehend.

But if the remaining material was further sub-divided by removing wood material representative of the growth season (Figure 2), unexpected 14C offsets would then be expected, especially at the slopes of the bomb peak. Accurate cutting of the tree rings is paramount for the reconstruction of atmospheric 14C data. This was already demonstrated by the intra-annual analyses of F14C for the calendar years of 1962 and 1963 shown here. Moreover, if the wood cutting was indeed selective (prior chemical extractions,as mentioned above), Response: This concern is irrelevant. Tree rings were sub-divided for intra-annual stable isotope analysis and detection of pre- and post monsoon hurricanes. Except from separating the TPBs no sub-division of tree-rings has been made for 14C analysis, except for 1961-1963 and 2005. Changes to the MS: This will be clarified the text.

I do not understand how the ms can assert at the abstract that "considerable autocorrelation was found in the d13C series, confirming incorporation of previous years' carbon significantly affecting the average age of derived wood", if the wood material tested was not the same. Analyses of $\delta$13C, $\delta$18O, as well as 14C should be done from homogenized cellulose-extracts from the same wood aliquots. Please, clarify. Response: autocorrelation was found in min, max and mean of $\delta$13C values of intra-ring sub-divisions. Tree-rings were sampled by dissecting sub-divisions of approx. 0.5mm. From each sub-division aliquots (=same mass (weight) from each sub-division) were taken, pooled together and homogenized for 14C analysis. The rest of each subdivision was used for stable isotope analyses. Cellulose extraction has been performed on each individual sample for 14C and stable isotope analysis, respectively. Changes to MS: This will be clarified and exemplified in the revised MS.

-p5, l164 & 165. Some of the TPBs removed were selected for 14C dating in phase one (4 or 5 samples). There is no mentioning of the chemical treatment they were subjected to prior sample processing for 14C-AMS. Please, explain. . . Response: Cellulose extraction has been performed on each individual sample for 14C and stable isotope analysis, respectively. It has been mentioned in the abstract and introduction that measurements were performed on cellulose. In paragraph 2.4 we outline that all "110 samples were holocellulose extracted with a base-acid-base-acid-bleaching procedure after (Němec et al., 2010). Ten samples were further purified to alpha-cellulose with an additional base treatment (17.5 % NaOH for 2 h at room temperature) followed by washing and freeze-drying." Co-authors and project leaders Lukas Wacker and Gerhard Helle are well aware of the need for and importance of cellulose extraction and sample homogenization for stable isotope and 14C analyses. Various methodologies of cellulose extraction and sample homogenization are established in their laboratories at ETH, Zürich and GFZ, Potsdam since decades. The methods applied here are well adopted by the international scientific community and all methods used here were previously published in peer-reviewed scientific journals. We have indicated which specific procedure has been used and have referenced the corresponding papers (e.g. Wacker et al. 2010a, b; Nemec et al 2010; Wieloch et al. 2011; Laumer et al. 2009 etc.) that describe these methods in detail. We initially thought that this might by sufficient and maintains the reading flow of the MS. Changes to MS: If given the opportunity by the editor we will be more specific and detailed in the revised MS.

-p.6, l182 to 190. This portion is very confusing. The TPBs F14C and OxCal were used to anchor the chronosequence. This would give a general idea of the calendar ages of these tree rings, which is ok. But since no chemical extraction appear to have being applied to TPB samples (no description of such is offered), Response: We disagree! As

mentioned above, the MS unambiguously mentions that cellulose extraction has been performed and cites descriptions of the methods used. Cellulose extraction has been performed on each individual sample for 14C and stable isotope analysis, respectively. Hence, TPBs underwent cellulose extraction as well. Changes to MS: If given the opportunity by the editor we will be more specific and detailed in the revised MS.

I do not understand why one should expect that they would match with the NH3. Please, rephrase statements. Response: The study site is located in the NH3 zone as outlined by Hua et al. 2013. Hence, one could expect or hypothesize that all plant material containing carbon photosynthesized at this site should have the isotopic signature (14C and 13C) of atmospheric CO2 prevailing at this site. Changes to MS: We will attempt to be more specific and clearer in the revised MS.

Regarding figure 3, and text portion between l187 to 190. What do you mean w/ "the baobab samples' position on the time axis is relative to their position within the tree ring of a growing season lasting from June until September"? Were the calendar years in the "x-axis" of figures 3 (and figure 8, as well) adjusted to match w/ the growing season of the baobab species as shown in Fig. 1C (June to September)? It is hard to see if such adjustment was applied in figure 3, as the figure is small. But I think that this adjustment was not applied to figure 8, as it should, and therefore the entire baobab F14C values are too far to the left. Have you take this monthly shift in account in the modelling as well? This calendar year adjustment should also appear at Table 2, second column to avoid confusing between growth date and dendro-date. Response: We will more clearly write how we positioned our samples on the x-axis. We think however, that the positioning was properly done, but maybe not easy to read in the graph. That is why we did a zoom-in on figure 3 (where you can better see the positioning in the relevant part). Nevertheless, we will check everything once again.

On figure 3A, I am left unsure (without checking all records in Hua et al. 2013 supplementary material) the main differences in uncertainties between SH3, SH1-2 and NH3 records beyond about 1972 (orange shaded area). Why is this shaded area particularly

different from all others, when the SH3 record (based on Muna Island data) stopped in 1979? Beyond this calendar year most records assume no differences between hemispheres due to scarcity of data in the tropics. Please, explain. Response: We disagree! We are convinced that the data set published by Hua et al. 2013 is correct and was cited by us correctly. We kindly ask for your understanding that we cannot recap in all detail the paper by Hua et al. 2013. Please do note that SH3 is not different from all others, it just caries larger uncertainties, likely due to the fact, that Muna Island stops in 1973.

Figure 3B, I appreciate the effort of showing F14C values between the calendar year of 1962 to 1964, but further discussions on air mass circulation (as mentioned earlier) are still lacking. Since the citation of Nydal & Lovseth 1983 is already listed in the article, all other records in the same zonal band in this article should be added to the plot. Response: Thank you for this comment. Changes to MS: We will add a discussion on potential effects due to atmospheric circulation and also add the data to fig. 3b in the revised MS.

Second, most of the citations in this figure legend are not in the reference list. Third, replace Turnball et al. 2017 by Turnbull et al. 2017. Response: Thank you for this comment. Changes to MS: We will correct the references and the reference list in the revised MS.

-p.9, section 3.1. I do not understand why one should expect that the TPbs would match with the NH3, or even match with the TRs (holocellulose extracts, Table 2). I don't see how this part is relevant. Response: As mentioned above: our study site is located in the NH3 zone as outlined by Hua et al. 2013. Hence, one could expect or hypothesize that all plant material containing carbon photosynthesized at this site should have the isotopic signature (14C and 13C) of atmospheric $CO_2$ prevailing at this site. Since our data show a clear mismatch we think this is important to mention. In other words, how can one expect plant material (like wood or specific wood cells like TPBs) to not show the atmospheric carbon signature of the region it was growing?

Changes to MS: We suggest no changes to the MS in this regard.

Most importantly would be comparisons between F14C data of TRs and alternative alpha-cellulose treatments that target the removal of starches and sugars (e.g., "Soxhlet"-type extractions using solvents). Note that the alpha-cellulose extraction described here was attained by adding an extra step of 17.5% NaOH to the holocellulose procedure. Incomplete removal of resinous compounds during chemical pretreatment of tree rings biasing 14C data has been shown by others (Cain and Suess 1976, Westbrook et al. 2006, for example). Response: As mentioned above: Cellulose extraction has been performed according to well established methods (cf. citations in MS: Wacker et al. 2010a, b; Nemec et al 2010; Wieloch et al. 2011; Laumer et al. 2009 etc.). Please do note that angiosperm baobab wood does hardly contain resinous compounds and if so resins, as well as other extractives (e.g. starch, sugars etc.) would have been completely removed by the procedures applied (for review of established methods, including methods applied in this MS please cf. Helle, G., Pauly, M., Heinrich, I., Schollaen, K. (2020). Stable isotope signatures of wood, its constituents and methods of cellulose extraction. In: Siegwolf, R., Brooks, J.R., Roden, J., Saurer, M. (eds.). Springer: Tree Physiology Book Series.) Changes to MS: We will try to rephrase our methods section in order to be clearer and more specific.

-p12, section 4.1. I found this section highly speculative; especially when no 14C dating targeting starch extracts from the baobab parenchyma-dominated wood was attempted. Response: Solvent extractives (starch sugars, etc.) were all removed. Changes to MS: We will be more specific in the revised MS.

Richardson et al. (2013), cited in this section, indeed found direct evidence for 'fast' and 'slow' cycling reserves in stemwood. However, Richardson et al. (2013) also stated that even though aboveground temperate forest trees contained very old pools of starch and sugars, stressed trees would still use up first all available present-day fast cycling carbon pool to support growth and metabolism. This would include even the most recently added starch molecules. Therefore, the usage of "older" NSC reserves

was set for times of stress. Richardson et al. (2013) did not mentioned that ring cellulose 14C results were inaccurate after direct comparison with the northern. Response: We disagree that "older" NSC reserves are used for times of stress only. Deciduous trees like baobabs do need carbohydrate reserves for bud growth, leaf emergence and maintanence metabolism during the dormant period. That is why we have assumed a contribution of "old carbon" of 15%. Please do note, with the simple model considerations presented here we do not intend to precisely describe or capture temporal changes of carbohydrate pools in baobab trees. We can show, however, that the F14C values of baobab tree-ring cellulose can better be explained when assuming a mixing of pools of carbohydrate precursors of cellulose. Changes to MS: We will be clearer and give more details on this aspect in the revised MS.

---

## Author Comment (AC3) · 6 Nov 2020

Dear Dr. Mazumdar, dear referees,

We gratefully acknowledge the efforts of Dr. Santos and Dr. Chakraborty for their comprehensive reviews of our manuscript. Their questions, criticism and thoughtful comments are very appreciated. In this response – prior to the editor's decision - we provide additional input and perspective that make some of the referees' concerns less pressing and may dispel others altogether. In the revised MS minor comments will

thoroughly be considered, grammar and spelling mistakes corrected.

The most critical comments of the referees can be summarized three key points: 1) our paper does not discuss in sufficient detail all potential causes of the observed F14C values of tree-ring cellulose being significantly lower than in the corresponding atmospheric $CO_2$ for the period around the bomb peak. Referees question our arguments for significant tree-physiological causes and particularly criticize the lack of discussion of external reasons. 2) our methodological and analytical description is lacking detail, unclear, ambiguous or not up to the mark. 3) The stable isotope data, our analysis of climate-proxy relationships and the climatological interpretation has raised no particular interest of the referees. If given the opportunity by the editor to revise our MS we will particularly focus on addressing these three issues. We will consider and incorporate the literature suggested by the referees. In view of this we will carefully re-write our methods section exemplifying that sample preparation and mass spectrometric analyses (IRMS and AMS) were performed in compliance with international standards and good scientific practice. In the discussion we will be taking into special consideration the potential external causes of 14C change during the period of the bomb peak (e.g. fossil fuel burning, atmospheric circulation etc. as put forward by the referees). In the following, we outline the key changes we intend to incorporate into the revised manuscript to consider the individual key points raised by the referees. Comments to Referee #2:

General remarks Radiocarbon analysis of the annual rings of trees has been carried out by several in- vestigators to study a variety of natural processes. Such kinds of records, especially in the extra-tropical region of the northern hemisphere are widely available. The trop- ical region, however, is not well represented. To fill this gap Slotta et al. attempted to reconstruct atmospheric 14C records from southern Oman based on the radiocarbon analysis of tree rings. The atmospheric radiocarbon activity showed anomalous enrichment during the early to mid-1960s, which is well documented in various atmospheric measurements as well as observed in several tree ring-based proxy

records. The authors have made a high-frequency sampling of a baobab tree during the bomb peak interval in order to study the nature of the 14C variability and the underlying mechanism that caused the observed variability. One of the main observations of their analysis is that the 14C variability in this region is characterized by a significantly low value (ca 9%) compared to the expected value across the similar latitudinal belt. The authors opine that the internal cause, such as plant physiological processes are primarily responsible for this depletion.

Apparently, they ignore the external factors, such as the fossil fuel dilution of atmospheric 14C variability, which may also produce such kind of anomalous signal. I would suggest the authors discuss this aspect as well to systematically rule out this possibility before coming to a definitive conclusion. Response: Thank you very much for this identifying this significant weakness of our MS. Changes to MS: If given the opportunity by the editor, in the revised MS we will address the issue of external factors diluting bomb-induced 14C variabilty in the atmosphere and give more details on the aspects raised by the referee below.

Section 4.1 The authors observed 14C activity in their tree ring that was noticeably lower than the NH3 and SH3 around the bomb peak (1964-1967). The authors explain that the anomalously low values were driven by plant physiological activities, the carbohydrate turn over time. But this hypothesis suffers from some limitations because such kind of low tree ring 14C activities has been reported by some investigators in the Asian region without invoking the tree physiological process. For example, Kikata et al. observed a bomb peak around D14C = 692‰ in Vietnam. Hua et al. (2000) found 694‰ in northern Thailand. Chakraborty et al. (1994) found 630‰ in an urban area in west India. Murphy et al. observed a slightly higher value of 705‰ in central India, which was also supported by Chakraborty et al (2008)'s observation of 708‰ in another site in central India. Response: Thank you very much. We were unaware of the details of papers you mention. Changes to MS: We will incorporate the results to our MS, carefully change our discussion and cite the above mentioned literature

accordingly.

Some of these authors have attributed the lower value of atmospheric 14C activity in a specific region in terms of fossil fuel dilution. For example, Chakraborty et al. (1994) analyzed a teak sample from a western Indian urban area and found a somewhat low value of 630‰ but the same species of another teak sample obtained from a central Indian but forested environment showed a bomb peak of 708%. The lowering of bomb peak (approx. 11%) in the urban area was attributed to fossil fuel dilution of atmospheric 14C. Response: Thank you very much for this information. Changes to MS: We will evaluate our data with respect to potential effects of fossil fuel dilution of atmospheric 14C.

Chakraborty et al. (2008) did not invoke the idea of tree physiological process in this case, though the possibility, in principle, may not be ruled out. But, the occurrence of two different 14C values in the same tree species at two different places seem to be driven by an external factor(s) rather than the tree physiological processes. Response: We understand, that the teak tree species invested by Chakraborty et al. 2008 is much different in wood anatomy and tree ecophysiology than the baobabs studied in our MS. Wood anatomy and wood cell types and proportions of fibre vs. parenchyma tissue is closer to those of angiosperms from temperate zones. Although the proportion of parenchyma tissue of teak is higher than those of angiosperm tree species of temperate zones, which might partly (besides dilution by fossil fuel burining and others)e xplain why lower 14C ratios were found as well in teak. Changes to MS: We will evaluate our data with respect to potential effects of fossil fuel dilution of atmospheric 14C and compare and discuss our data with those of the above suggested publications.

There may be other reasons to doubt the tree physiological process affecting the tree ring 14C activity. The mechanism explained by the authors involves the incorporation of previous year's carbon that significantly affects the average age of the current year wood. If that be the case, then a similar effect should have been observed by other investigators. Response: No, we disagree! No other tree species invested for 14C during

the bomb peak period has such high proportion of long-living parenchyma. As mentioned by the referee #2 above also authors have found lower values of atmospheric 14C activity as expected.

Hua et al. (2003) analyzed a Pinus Radiata tree sample collected at Armidale in New South Wells, Australia and found excellent agreement with the atmospheric observation for the period of 1952 to 1967. But these authors observed higher 14C values in their Armidale tree ring samples for the period of 1968-1975. Obviously, an increase in radiocarbon activity cannot be explained by tree physiological processes. So either an increase or a decrease in 14C activity is likely to be driven by external factors. Response: We admit that observed variability of 14C values in tree rings cannot be explained by tree physiological processes alone, however, and in return we cannot follow the argument that only external factors cause 14C variability. Tress are living organisms, they react to changes of their environment in many ways. For example, they modify the stomatal aperture of their leaves/needle in response to moisture availability or CO2 concentration of the atmosphere. Related to this isotopic fractionation occurs that can lead to very individual or site specific 13C and 14C contents in tree organic matter. Changes to MS: We will stress the importance of external factors and will detail them in the revised MS. However, we do not see striking arguments why our suggestion to consider tree physiological effects to some extent should be discarded.

Using a numerical exercise and autocorrelation analysis of d13C data Slotta et al. estimate that approx 85% of fast cycling carbon pools and 15% of slow-cycling carbon pools are contributing to the lower values of the bomb peak. If this explanation is true, then this effect should have been manifested in the entire record the authors have reported, which is not apparent from their results. Rather, the authors admit that the baobab F14C values for 1945, 1952-1954, 1956 and 1957 are indeed higher than the calculated range. This observation casts doubt in their explanation of the old carbon turn over mechanism in explaining the negative excursion of 14C activity during the bomb peak period. Response: As mentioned in response to referee#1 above, the unexpected significant mismatch between the baobab F14C values and the post-bomb atmospheric curve NH3 prompted for some reasonable interpretation. Hence, we suggested and still suggest that it is from a mixed carbon pool in conjunction with the extraordinary high content and longevity of parenchyma tissue relative to short lived fibre tissue that constitute the tree rings of baobabs. With the simple model considerations presented here we do not intend to precisely describe or capture temporal changes of carbohydrate pools in baobab trees. We can show, however, that the F14C values of baobab tree-ring cellulose can better be explained when assuming a mixing of pools of carbohydrate precursors of cellulose instead of assuming direct transfer to atmospheric carbon into organic matter of tree rings. Changes in the manuscript: We will rewrite ithe MS to stress the uncertainties involved with our interpretation.

There may be another explanation of lower 14C activity in this region. Cember (1989) analyzed coral 14C from across the Red Sea to estimate the gas exchange rate. Cember observed a very high air-sea exchange process over the Red Sea region. If this process is also operative in this region which is not very far from the Red Sea, then a viable explanation of anomalous 14C activity in the atmosphere may be provided. Response: Thank you very much for this comment. Changes in the manuscript: We will pick this up and incorporate the potential air-sea exchange processes into our interpretation and discussion.

2.4 Radiocarbon dating The analytical description provided by the authors is not up to the mark. For example, radiocarbon dating requires 13C correction and age correction; there is no mention of whether such kind of corrections has been done. The reporting of 14C activity, especially in the case of sequential samples (tree ring, corals) is typically done in cap delta notation (D14C), but the authors have preferred normalized activity. Response: By utilizing F14C we follow the suggestion of Reimer, P. J., Brown, T. A. and Reimer, R. W. (2004). "Discussion: Reporting and calibration of post-bomb C-14 data." Radiocarbon 46(3): 1299-1304. In this paper the advantages of using F14C in studies of the bomb peak period are well described.

For comparison purposes with the published records, the authors are suggested to report the 14C activity in cap delta notation. Finally, the error in 14C measurement should be mentioned in terms of cap delta as well as the corresponding temporal value. Changes in the manuscript: We are inclined not to make changes to our nomenclature (cf. above). However, if requested by the editor we can provide del14C values for the most prominent excursions of our baobab record in the revised MS. As indicated in the figure captions, errors are smaller than the symbols of the figures. However, we will specify the numbers in the methods section of the revised MS.

Minor issues: Line 92: The rainfall amount and its isotopes usually show a weak inverse correlation. Pls, provide reference for evidence of "strong" correlation. Line 141: very heavy rainfall in a single day producing high negative d18O "due to amount effect" is not technically right. Many studies (Lawrence and Gedzelman, 1996; Gedzelman et al., 2003; Lawrence et al. 1998; Chakraborty et al., 2016; Xu et al., 2019), showed that extreme precipitation events such as cyclonic activities produced very low values of precipitation d18O. Response: Thank you very much for this comment. Changes in the manuscript: We will correct this in the revised MS and will be technically more specific.

Line 181: pls provide a zoomed figure for the 1962-63 record of bomb 14C. Line 199: What are the reference materials used? Response: Thank you very much for this comment. Changes in the manuscript: We will be more specific in the revised MS.

Line 202: Please provide the permil sign after 0.15 and 0.25. Line 217: 'weakening' should be "weaken". Response: Thank you very much for this comment. Changes in the manuscript: We will correct these mistakes in the revised MS.

Line 251-252: How the interpretation of the F14C data was confirmed by visual and sta- tistical comparison of the TRW chronology with precipitation data should be explained in detail. Response: Thank you very much for this comment. Changes in the manuscript: We will be more specific in the revised MS.

Line 255: 'shallow' should be replaced by "gentle". Changes in the manuscript: We will do in the revised MS.

Line 268: 'radiocarbon' should be followed by "analysis". Line 275: 'considerably declining' meaning is not clear. Changes in the manuscript: We will be more specific in the revised MS.

Line 280: What is the physical basis of getting a strong correlation between d18O and RWI? Also mentioned in Line 334. Please provide the value of correlation and state the sample number. Response: Thank you very much for this comment. Changes in the manuscript: We will be more specific and provide correlation coefficients in the revised MS.

Line 309: the lag between cyclonic events and the corresponding d18Omin should be provided on a monthly time scale. Line 355: 'extend' should be replaced by "extent". Changes in the manuscript: Thanks, we will correct this in the revised MS.

Line 494: "evaporative enrichment in 18O...".Please provide supportive evidence of enhanced soil evaporation, say by means of observed or reanalysis data in support of this speculation. Changes in the manuscript: Thanks, corresponding papers will be cited in the revised MS.

Line 505: "Vapor pressure deficit ...18O enrichment in leaf water", and "lower stomatal conductance...13C discrimination to decline" require supporting literature. Changes in the manuscript: Thanks, corresponding papers will be cited in the revised MS. For instance: Barbour MM (2007) Stable oxygen isotope composition of plant tissue: a review. Funct Plant Biol 34 (2):83-94. doi:https://doi.org/10.1071/fp06228

Line 514: the authors argue that the decline in d18O...might be due to the previous year's October precipitation. If so, then d18O is also expected to be auto-correlated. Response: This is correct, thank you very much for this comment. Changes in the manuscript: We will delete this speculation in the revised MS.

Line 524: likely in 'would have likely..." should be deleted. Changes in the manuscript: Thanks, we will correct this in the revised MS.

References: Cember 1989 Bom radiocarbon in the Red Sea: a medium scales gas exchange ex- periment. JGR Ocean 94:2111-2123. Lawrence, J. L. & Gedzelman, S. D. Low stable isotope ratios of tropical cyclone rains. Geophys. Res. Lett. 23, 527–530 (1996). Gedzelman, S., Lawrence, J., Gamache, J., Black, M., Hindman, E., Black, R., Dunion, J., Willoughby, H., Zhang, X., 2003. Probing hurricanes with stable isotopes of rain and water vapor. Mon. Weather. Rev. 131 (6), 1112–1127. Hua, Q., Barbetti, M., Zoppi, U., Chapman, D. M., and Thomson, B. 2003 Bomb Radiocarbon in Tree Rings from Northern New South Wales, Australia: Implications for Dendrochronology, Atmospheric Transport, and Air-Sea Exchange of $CO_2$, Radiocarbon, 45, 431-447. Xu et al. 2019 Stable isotope ratios of typhoon rains in Fuzhou, Southeast China, during 2013–2017. Kikata Y, Yonenobu H, Morishita F, Hattori Y. 1992. 14C concentrations in tree stems. Bulletin of the Nagoya University Furukawa Museum 8:41–6. In Japanese. Changes in the manuscript: Thanks, corresponding papers will be cited in the revised MS.

---

## Author Response (AR1)

Dear Dr. Mazumdar, dear referees,

We gratefully acknowledge the efforts of Dr. Santos and Dr. Chakraborty for their comprehensive reviews of our manuscript. Their questions, criticism and thoughtful comments are very appreciated. In this response we provide additional input and perspective that hopefully make some of the referees' concerns less pressing and may dispel others altogether. The original MS has been re-written to address two major critical key points raised:

1) our methodological and analytical description is lacking detail, unclear, ambiguous or not up to the mark.

2) our paper does not discuss in sufficient detail all potential causes of the observed $F^{14}C$ values of tree-ring cellulose being significantly lower than in the corresponding atmospheric $CO_2$ for the period around the bomb peak. Referees question our arguments for significant tree-physiological causes and particularly criticize the lack of discussion of external reasons.

Preliminary remarks:

We re-wrote the introduction and refined the aims of this study, which were primarily NOT striving at reconstructing the atmospheric $^{14}C$ curve from baobab tree rings as assumed by the referees.

The stable isotope data, our analysis of climate-proxy relationships and the climatological interpretation apparently raised little interest of Dr. Santos. Hence, apart from carefully considering the minor comments of Dr. Chakraborty on the stable isotope part we did not make any changes to it.

All additional literature suggested by the referees has been incorporated.

Grammar and spelling mistakes have been corrected. Minor comments have been considered in the revised MS.

Ad 1) We re-wrote our methods section and added schematic drawings (Fig. 2C, D) exemplifying the specific wood anatomy of *A. digitata* and the intra-annual sampling scheme applied in this study. Furthermore, we added some more detail about sample preparation and mass spectrometric analyses (IRMS and AMS) and cited additional peer-reviewed literature describing in detail the methods that have been applied in this study and which are well established in the laboratories of Lukas Wacker and Gerhard Helle at ETH Zurich and GFZ Potsdam.

Ad2)

We re-wrote the discussion and conclusions listing potential external factors of atmospheric $^{14}C$ dilution and discussing why we think that they are less likely responsible for the observed offset between baobab $F^{14}C$ and NH3 (and even SH3) atmospheric $F^{14}C$. We offer two explanations that are specifically related to tree physiological aspects related to the unique wood anatomical characteristics of *A. digitata*. Wood of this species can consist of up to more than 80% of parenchyma. Parenchyma cells are the major storage organs for carbohydrate reserves and they have the ability to divide throughout their lifespan over several years. Hence, we suggest that carbohydrate turnover and the high abundance of parenchyma are the major source for the observed offset. Published results from other sites and other tree species are acknowledged, however none of them is comparable to the specific framework of our baobab study.

Response to Comments of Referee Dr. Santos:

General remarks

Comment: The paper presents annual 14C data from an African baobab (Adansonia digitata) tree from Oman, for the interval AD 1941 to 2005. This work is important in that it provides a fairly detailed pre/post-bomb 14C time-series for a region that has not yet being part of the atmospheric 14C global compilation. This is actually one of the main goals of the manuscript. The authors have also improved the quality of the data set by providing intra-annual analyses of $\delta$13C and $\delta$18O, as well as F14C for the calendar years of 1962 and 1963.

Response: We thank the reviewer for considering this work as important for a region that is underrepresented in terms of 14C data. However, it has not been our major goal to reconstruct atmospheric $^{14}$C from baobab tree rings. The original purpose of this study has been re-written and clearified in the introduction of the revised MS.

Comment: While the high number of consecutive single tree-rings measured by radiocarbon al- lowed confirming the annual nature of the baobab species, a significant mismatch be- tween the baobab F14C values and the post-bomb atmospheric curve NH3 was de- tected. This mismatch prompted an alternative explanation, i.e. mixed pool of slow- turnover non-structural carbon (NSC) into the structural ring cellulose fraction - a strong functional trait of parenchyma-rich tree species (maybe ?!).

Response: We agree with this summarizing statement of our observations. However, we do not just explain the aberrant F14C values by incorporation of carbon from a mixed pool of slow and fast turnover of NSC. In the MS we propose an additional potential cause: namely the huge difference in longevity of wood forming plant cells: while parenchyma cells can live up to approx. 20 years, wood fibres live up only from a few weeks to a season. This means that parenchyma tissue can undergo changes in its 14C for over several years, whereas the 14C of fibre tissue is always assigned to a certain year. Since in baob- abs parenchyma occurs not only as bands but is also diffusely distributed within a tree ring, varying proportions of parenchyma and fibers can cause variations in F14C of a tree ring that can be, to a certain extent, unrelated to the specific date of the tree ring. Please note, that baobabs are unique in this regard. To our knowledge, no other tree species shows similarly high parenchyma contents than baobabs (69-88%).

Changes in the manuscript: We re-wrote introduction and discussion to make these two tree physiological aspects clearer. In addition, we have added details on potential external causes for the observed 14C trends (see responses to further comments below).

Comment: The Baobab terminal parenchyma bands F14C values presented here, definitely demonstrate that a large percent of the parenchyma in this tree species is relatively young, and as such, it provides valuable perspectives in the field of plant physiology. On the other hand, mixed carbon pools in putative structural ring cellulose fraction (in this case, slow-turnover NSC residue in holocellulose extracts) put into question the use of tree rings of this group of woody plant when reconstructing atmo- spheric 14C.

Response: Thank you for supporting our conclusion that our F14C data points to future perspectives in plant physiological research (in particular on baobabs, which are widely distributed in Africa and potentially threatened by global change). Our data set contributes a fairly detailed pre/post-bomb 14C time-series for a region that has not yet being part of the atmospheric 14C global compilation. However, it was NOT the main purpose of our MS to reconstruct atmospheric 14C from this data set. As written in the introduction, we primarily intended to use the 14C bomb peak to validate the counting/dating of tree rings. As mentioned by the referee above, the unexpected significant mismatch between the baobab F14C values and the post-bomb atmospheric curve NH3 prompted for some reasonable interpretation. In this case we suggested and

still suggest that it is from a mixed carbon pool in conjunction with the extraordinary high content and longevity of parenchyma tissue relative to short lived fibre tissue that constitute the tree rings of baobabs. Nonetheless, the referee is right. Tree species with such a high content of parenchyma should not be used for reconstructing atmospheric14C. This may raise particular issue for tropical regions, where tree angiosperm species show about 36% of parenchyma on average. In contrast, angiosperm tree species in temperate zones have a content of about 21%, only.

Changes to the manuscript: We re-wrote the introduction to clarify the original purpose of our 14C analyses and we added the aspects outlined above in the discussion of the revised MS.

Comment: I appreciate that in view of the perplexing results of the 14C data of the baobab tree rings, an alternative explanation should be considered. However, for the mixed pool NSC-ring cellulose assumption works, all other possible bias must be carefully ruled out.

Response: Thank you very much for this valuable comment. This point, i.e. other potential causes for the observed bias, has also been raised by referee #2.

Changes in the manuscript: As suggested, we add a paragraph tackling other, external effects on atmospheric 14C to the discussion of the MS and also rephrased the parts in the manuscript referring to this.

Comment: Robust methodologies must be properly done and explained in detail, as well as the use of reference materials/internal standard, or equivalent (i.e. interlaboratory measurements), and the use of further chemical extractions. All of those are missing here.

Response: Thank you very much for pointing to the lack of description of our methodology. This point has partly also been raised by referee #2.

Changes to the manuscript: As suggested, we have re-written the methods section and explain in little more detail the methods applied and provide additional literature describing the procedures established in our laboratories.

Comment: Given the absence of an independent benchmark, e.g. a short F14C sequence of consecutive single tree-rings from a non-parenchyma-rich woody plant in Oman, I cannot tell whether slow-turnover NSC detected in holocellulose extracts of baobab tree rings is a feasible explanation for the 14C offset observed here or not. For starters, 14C analysis of incomplete single tree rings (material that do not represent a full growing season) could contribute in 14C offsets (see specific comments/suggestions). Response: Good point. Unfortunately, project resources were limited and did not allow analyses of other tree species than baobabs. 14C analyses were done on the 2/3 of a tree ring. The last 1/3 including each terminal parenchyma band was discarded in order to minimize negative effects from long living parenchyma tissue that would smear the $^{14}$C signal. Since the transition from wood to the terminal parenchyma band cannot be separated precisely to 100%, we decided to also skip some of the wood. We believe that contamination from parenchyma causes larger bias than the seasonal effects. Baobab tree rings in Oman are largely formed between May and October each year; our samples from may represent the growing period from May to August or early September.

Comment: Furthermore, we had to keep in view that other factors must also play some role in those 14C offsets (atmospheric circulation and carbon dioxide from human activities, for example). Previous records across zones NH3, SH3 and SH1-2 are very scarce. Therefore, the possibility of multiple sources of air-14CO2 influencing Oman

should be discussed.

Response: Yes, we admit that these factors have not been addressed in the previous version of the MS. This point has been raised by referee #2 as well. Changes to the MS: We have added and discussed these aspects in the revised MS.

Comment: One cannot ignore the fact that during the assembly of the atmospheric post-AD 1950 14C global compilation by Hua et al. (2013) some datasets were disregarded due to its mismatches with other regional datasets. Therefore, a thorough evaluation of possible external effects should also be offered.

Response: Yes, once more, we admit that these factors have not been discussed previously. Changes to the MS: We have added and discussed these aspects in the revised MS.

Comment: Finally, procedures described here need further explanations and details. Response: Yes, thank you for this comment. Changes to the MS: We have detailed our procedures and provide additional literature on them.

Comment: The result and discussion part is quite jumpy and very tricky to follow. It does not quite convey the ideas of the underlying assumption offered to explain the baobab tree F14C offsets. I recommend a complete re-organization of the manuscript, by focusing on placing the absolutely necessary data, figures and tables (for the purpose of the paper) in the main text. Response: Thank you for this comment. Indeed, this paper is complex, as it presents and discusses ${}^{14}$C and stable isotope data sets. Apparently, there is some misunderstanding concerning the purpose of this paper. It is not ment to present a reconstruction of atmospheric 14C (see earlier response comments above).

Changes to the MS: We have partly rephrased the MS to make our intentions much clearer.

The stable isotope findings were not particularly striking. Although important, they are currently creating a lot of distraction. I strongly suggest moving them (most of its description, associated material and discussions) to a supplementary text or appendix.

Response: We kindly ask Dr. Santos to not insist on moving the stable isotope part to

pretation is of valuable interest to the dendro- and palaeclimate community. In particular, stable isotope data points to distinct relations between intra-seasonal stable isotope patterns and pre- and post monsoon cyclones. Furthermore, tree-ring stable isotope records from regions like Oman are still scarce and knowledge about the climatic significance of baobab tree-ring parameters is important. According to her expertise, the referee feels distracted while focusing on the radiocarbon part, only. Note, there seems to be some misunderstanding concerning the purpose of this paper, which was not ment to present a reconstruction of atmospheric 14C (see response comments above) Changes to the MS: We did not do any changes to the MS in this regard.

Specific Comments/Suggestions I am going to focus here on just major topics that are in need of clarification to verify the fitness of the data shown.

- p4, l111. It is stated that 10 trees were sampled by increment cores from four different orientations (NE, SE, SW, and NW). Do you mean four radii were collected per tree?! If yes, random tree rings were used for 14C analysis or just one tree and radii's? Please, clarify. Response: One tree, and one radii for 14C and stable isotope analysis Changes to the MS: We are more specific the revised MS.

- p5, l148 to l55. How the tree specimen selected was dendrochronologically-secure? How the chronosequence of tree rings (prior 14C dating) was obtained without a master chronology for Baboab species?! The passage selected here describes just figure 2. Later (at p.6, l190 to 194), it is explained that no dated tree-ring width chronology from the study region is currently available.

- Response: One tree, and one radii. No chronology from several trees has been developed for this case study and no chronology from any other tree species from this site or region has been published. This has been written the Materials and Methods part of the MS.

- Therefore to anchor the chronosequence of tree rings (prior counting of all baobab tree rings) the F14C of the TPBs and Oxcal was used instead. Is this correct?!...

Response: No. One tree, and one radii was analysed. No chronology from several trees has been developed for this case study. Tree rings were counted and 14C data was used to validate the ring count.

- If yes, this explanation should appear early on in the text. The fitness of the chronosequence is the backbone of the atmospheric 14C record production using tree rings.

Response: One tree, and one radii. No chronology from several trees has been developed for this case study. Tree rings were counted and 14C data was used to validate the ring count. This has been written the text.

- Plus, add what type of juniper species you are referring to. Response: Juniperus seravschanica

- J. excelsa. Changes to the MS: We have added the corresponding citation to the text.

- p5&6, l159 to l65, and l177 to l178. Passages explaining the wood material used for radiocarbon and stable isotope analysis are confusing, and very troubling. It appears that the full dataset was produced in two phases, a pre-screening phase with 5 calendar years or so, where just 1/3 of the tree ring (cut parallel to the fiber orientation, in radial direction from the cambial zone) was used. In a second phase, in order to measure the remaining calendar years, just 2/3 of the single tree ring was used for 14C dating. The remaining material was then used for 13C. This description gives the idea that the tree ring cutting for isotopic analysis was selective, before chemical extractions took place.

- Response: As mentioned above, the tree rings of the baobab sample were counted. Since missing rings and miscounting was expected 14C bomb peak wiggle matching should helb to validate the ring counting. In the first phase 5 individual tree rings covering individual, not consecutive years of the bomb peak period (as defined by ring counting) were selected for narrowing down the period. Changes to the MS: this is exemplified by an additional figure and in more detail in the revised text.

Normally a homogenized cellulose-extract of a full single whole-ring (from early- to late-wood) is used to reconstruct atmospheric 14C data.

Response: Cellulose has been extracted for all samples of this study. Here, all tree-ring material, except from terminal parenchyma bands has been used for a 14C analysis. This will been written in the text.

It is understandable that since the baobab contains 69-88 % parenchyma cells, mostly concentrated at the terminal parenchyma bands (TPBs) or late-wood, this portion was removed.

Response: Yes, parenchyma bands (TPBs) were seperated from tree-ring material prior to 14C and stable isotope analysis. But, no further selective sampling was performed for 14C analysis. Tree rings of 1961-1963 and 2005 were sub-divided, but all sub-division were then analysed (see fig. 8) Changes to the MS: This has been clarified in the text.

But if the remaining material was further sub-divided by removing wood material representative of the growth season (Figure 2), unexpected 14C offsets would then be expected, especially at the slopes of the bomb peak. Accurate cutting of the tree rings is paramount for the reconstruction of atmospheric 14C data. This was already demonstrated by the intra-annual analyses of F14C for the calendar years of 1962 and 1963 shown here. Moreover, if the wood cutting was indeed selective (prior chemical extractions, as mentioned above), Response: This concern is irrelevant. Tree rings were sub-divided for intra-annual stable isotope analysis and detection of pre- and post monsoon hurricanes. Except from separating the TPBs no sub-division of tree-rings has been made for 14C analysis, except for 1961-1963 and 2005.

I do not understand how the ms can assert at the abstract that "considerable autocorrelation was found in the d13C series, confirming incorporation of previous years' carbon significantly affecting the average age of derived wood", if the wood material tested was not the same. Analyses of $\delta$13C, $\delta$18O, as well as 14C should be done from homogenized cellulose-extracts from the same wood aliquots. Please, clarify.

Response: autocorrelation was found in min, max and mean of $\delta$13C values of intra-ring sub-divisions. Tree-rings were sampled by dissecting sub-divisions of approx. 0.5mm. From each sub-division aliquots (=same mass (weight) from each sub-division) were taken, pooled together and homogenized for 14C analysis. The rest of each subdivision was used for stable isotope analyses. Cellulose extraction has been performed on each individual sample for 14C and stable isotope analysis, respectively.

-p5, l164 & 165. Some of the TPBs removed were selected for 14C dating in phase one (4 or 5 samples). There is no mentioning of the chemical treatment they were subjected to prior sample processing for 14C-AMS. Please, explain. . .

Response: Cellulose extraction has been performed on all samples for 14C and stable isotope analysis, respectively. It is mentioned in the abstract and introduction that measurements were performed on cellulose. In paragraph 2.4 we outline that all "110 samples were holocellulose extracted with a base-acid-base-acid-bleaching procedure after (Němec et al., 2010). Ten samples were further purified to alpha-cellulose with an additional base treatment (17.5 % NaOH for 2 h at room temperature) followed by washing and freeze-drying."

-p.6, l182 to 190. This portion is very confusing. The TPBs F14C and OxCal were used to anchor the chronosequence. This would give a general idea of the calendar ages of these tree rings, which is ok. But since no chemical extraction appear to have being applied to TPB samples (no description of such is offered),

Response: We disagree! As mentioned above, the MS unambiguously mentions that cellulose extraction has been performed and cites descriptions of the methods used. Cellulose extraction has been performed on each individual sample for 14C and stable isotope analysis, respectively. Hence, TPBs underwent cellulose extraction as well.

I do not understand why one should expect that they would match with the NH3. Please, rephrase statements. Response: The study site is located in the NH3 zone as outlined by Hua et al. 2013. Hence, one could expect or hypothesize that all plant material containing carbon photosynthesized at this site should have the isotopic signature (14C and 13C) of atmospheric $CO_2$ prevailing at this site. The revised MS hopefully is more specific and clearer on this.

Regarding figure 3, and text portion between l187 to 190. What do you mean w/ "the baobab samples' position on the time axis is relative to their position within the tree ring of a growing season lasting from June until September"? Were the calendar years in the "x-axis" of figures 3 (and figure 8, as well) adjusted to match w/ the growing season of the baobab species as shown in Fig. 1C (June to September)? It is hard to see if such adjustment was applied in figure 3, as the figure is small. But I think that this adjustment was not applied to figure 8, as it should, and therefore the entire baobab F14C values are too far to the left. Have you take this monthly shift in account in the modelling as well? This calendar year adjustment should also appear at Table 2, second column to avoid confusing between growth date and dendro-date. Response: The positioning has been checked and was properly done, but maybe not easy to read in the graph. That is why we did a zoom-in on figure 3 (where you can better see the positioning in the relevant part).

On figure 3A, I am left unsure (without checking all records in Hua et al. 2013 supplementary material) the main differences in uncertainties between SH3, SH1-2 and NH3 records beyond about 1972 (orange shaded area). Why is this shaded area particularly different from all others, when the SH3 record (based on Muna Island data)

stopped in 1979? Beyond this calendar year most records assume no differences between hemispheres due to scarcity of data in the tropics. Please, explain.

Response: We disagree! We are convinced that the data set published by Hua et al. 2013 is correct and was cited by us correctly. We kindly ask for your understanding that we cannot recap in all detail the paper by Hua et al. 2013. Please do note that SH3 is not different from all others, it just caries larger uncertainties, likely due to the fact, that Muna Island stops in 1973.

Figure 3B, I appreciate the effort of showing F14C values between the calendar year of 1962 to 1964, but further discussions on air mass circulation (as mentioned earlier) are still lacking. Since the citation of Nydal & Lovseth 1983 is already listed in the article, all other records in the same zonal band in this article should be added to the plot.

Response: Thank you for this comment. Changes to MS: We have made no changes to the MS regarding potential effects due to atmospheric circulation, because this distracts from the original purpose of the MS and is beyond the task. The mentioned data will be added to a future paper comparing baobab $^{14}$C with data from well-dendro-dated J. excelsa from Oman.

Second, most of the citations in this figure legend are not in the reference list. Third, replace Turnball et al. 2017 by Turnbull et al. 2017.

Response: Thank you for this comment. Changes to MS: We will correct the references and the reference list in the revised MS.

-p.9, section 3.1. I do not understand why one should expect that the TPbs would match with the NH3, or even match with the TRs (holocellulose extracts, Table 2). I don't see how this part is relevant.

Response: As mentioned above, our study site is located in the NH3 zone as outlined by Hua et al. 2013. Hence, one could expect or hypothesize that all plant material containing carbon photosynthesized at this site should have the isotopic signature (14C and 13C) of atmospheric $CO_2$ prevailing at this site. Since our data show a clear mismatch we think this is important to mention. In other words, how can one expect plant material (like wood or specific wood cells like TPBs) to not show the atmospheric carbon signature of the region it was growing?

Changes to MS: We suggest no changes to the MS in this regard.

Most importantly would be comparisons between F14C data of TRs and alternative alpha-cellulose treatments that target the removal of starches and sugars (e.g., "Soxhlet"-type extractions using solvents). Note that the alpha-cellulose extraction described here was attained by adding an extra step of 17.5% NaOH to the holocellulose procedure. Incomplete removal of resinous compounds during chemical pretreatment of tree rings biasing 14C data has been shown by others (Cain and Suess 1976, Westbrook et al. 2006, for example). Response: Thank you for this valuable advice. As mentioned above: Cellulose extraction has been performed according to well established and approved methods (cf. citations in MS: Wacker et al. 2010a, b; Nemec et al 2010; Wieloch et al. 2011; Laumer et al. 2009 etc.). Please consider that Baker, Santos et al. 2017 have applied very similar procedures (e.g. Wieloch et al. 2011). Please also note that angiosperm baobab wood does hardly contain resinous compounds and if so resins, as well as other extractives (e.g. starch, sugars etc.) would have been completely removed by the procedures applied (for review of established methods, including methods applied in this MS please cf. Helle, G. et al., (2021): Stable isotope signatures of wood, its constituents and methods of cellulose extraction. In: Siegwolf, R., Brooks, J.R., Roden, J., Saurer, M. (eds.). Springer: Tree Physiology Book Series.)

Changes to MS: We attempted to rephrase our methods section in order to be clearer and more specific.

-p12, section 4.1. I found this section highly speculative; especially when no 14C dating targeting starch extracts from the baobab parenchyma-dominated wood was attempted. Response: Solvent extractives (starch sugars, etc.) were all removed.

Richardson et al. (2013), cited in this section, indeed found direct evidence for 'fast' and 'slow' cycling reserves in stemwood. However, Richardson et al. (2013) also stated that even though aboveground temperate forest trees contained very old pools of starch and sugars, stressed trees would still use up first all available present- day fast cycling carbon pool to support growth and metabolism. This would include even the most recently added starch molecules. Therefore, the usage of "older" NSC reserves was set for times of stress. Richardson et al. (2013) did not mentioned that ring cellulose 14C results were inaccurate after direct comparison with the northern.

Response: We disagree that "older" NSC reserves are used for times of stress only. Deciduous trees like baobabs do need carbohydrate reserves for bud growth, leaf emergence and maintanence metabolism during the dormant period. That is why we have assumed a contribution of "old carbon" of 15%. Please do note, with the simple model considerations presented here we do not intend to describe in detail or capture temporal changes of carbohydrate pools in baobab trees. We can show, however, that the F14C values of baobab tree-ring cellulose can better be explained when assuming a mixing of pools of carbohydrate precursors of cellulose. Further investigations involving $^{14}$C analysis of various cellulose precursors will improve on these aspects. Changes to MS: We have been more careful with formulating our interpretation in the revised MS.

Comments to Referee #2:
General remarks Radiocarbon analysis of the annual rings of trees has been carried out by several in- vestigators to study a variety of natural processes. Such kinds of records, especially in the extra-tropical region of the northern hemisphere are widely available. The trop- ical region, however, is not well represented. To fill this gap Slotta et al. attempted to reconstruct atmospheric 14C records from southern Oman based on the radiocarbon analysis of tree rings. The atmospheric radiocarbon activity showed anomalous enrichment during the early to mid-1960s, which is well documented in var- ious atmospheric measurements as well as observed in several tree ring-based proxy records. The authors have made a high-frequency sampling of a baobab tree during the bomb peak interval in order to study the nature of the 14C variability and the underlying mechanism that caused the observed variability. One of the main observations of their analysis is that the 14C variability in this region is characterized by a significantly low value (ca 9%) compared to the expected value across the similar latitudinal belt. The authors opine that the internal cause, such as plant physiological processes are primarily responsible for this depletion.
Apparently, they ignore the external factors, such as the fossil fuel dilution of atmo-spheric 14C variability, which may also produce such kind of anomalous signal. I would suggest the authors discuss this aspect as well to systematically rule out this possibil- ity before coming to a definitive conclusion.
Response: Thank you very much for this identifying this significant weakness of our MS. Changes to MS: in the revised MS we have addressed the issue of external factors diluting bomb-induced 14C variability in the atmosphere and give more details on the aspects raised by the referee below.
Section 4.1 The authors observed 14C activity in their tree ring that was noticeably lower than the NH3 and SH3 around the bomb peak (1964-1967). The authors explain that the anomalously low values were driven by plant physiological activities, the car- bohydrate turn over time. But this hypothesis suffers from some limitations because such kind of low tree ring 14C activities has been reported by some investigators in the Asian region without invoking the tree physiological process. For example, Kikata et al. observed a bomb peak around D14C = 692‰ in Vietnam. Hua et al. (2000) found 694‰ in northern Thailand. Chakraborty et al. (1994) found 630‰ in an urban area in west India. Murphy et al. observed a slightly higher value of 705‰ in central India, which was also supported by Chakraborty et al (2008)'s observation of 708‰ in another site in central India.

Response: Thank you very much. We were unaware of the details of papers you mention. Changes to MS: We will incorporate the results to our MS, carefully change our discussion and cite the above mentioned literature accordingly. Some of these authors have attributed the lower value of atmospheric 14C activity in a specific region in terms of fossil fuel dilution. For example, Chakraborty et al. (1994) analyzed a teak sample from a western Indian urban area and found a somewhat low value of 630‰ but the same species of another teak sample obtained from a cen- tral Indian but forested environment showed a bomb peak of 708%. The lowering of bomb peak (approx. 11%) in the urban area was attributed to fossil fuel dilution of atmospheric 14C.

Response: Thank you very much for this information. Changes to MS: We discuss our data with respect to potential effects of fossil fuel dilution of atmospheric 14C.

Chakraborty et al. (2008) did not invoke the idea of tree physiological process in this case, though the possibility, in principle, may not be ruled out. But, the occurrence of two different 14C values in the same tree species at two different places seem to be driven by an external factor(s) rather than the tree physiological processes.

Response: We understand, that the teak tree species invested by Chakraborty et al. 2008 is much different in wood anatomy and tree ecophysiology than the baobabs studied in our MS. Wood anatomy and wood cell types and proportions of fibre vs. parenchyma tissue  is closer to those of angiosperms from temperate zones. Although the proportion of parenchyma tissue of teak is higher than those of angiosperm tree species of temperate zones, which might partly (besides dilution by fossil fuel burning and others) explain why lower 14C ratios were found as well in teak.

Changes to MS: We evaluate our data with respect to potential effects of fossil fuel dilution of atmospheric 14C and compare and discuss our data with those of the above suggested publications.

There may be other reasons to doubt the tree physiological process affecting the tree ring 14C activity. The mechanism explained by the authors involves the incorporation of previous year's carbon that significantly affects the average age of the current year wood. If that be the case, then a similar effect should have been observed by other investigators.

Response: No, we disagree! No other tree species invested for 14C during the bomb peak period has such high proportion of long-living parenchyma as A. digitata (baobab).

Hua et al. (2003) analyzed a Pinus Radiata tree sample collected at Armidale in New South Wells, Australia and found excellent agreement with the atmospheric observation for the period of 1952 to 1967. But these authors observed higher 14C values in their Armidale tree ring samples for the period of 1968-1975. Obviously, an increase in radiocarbon activity cannot be explained by tree physiological processes. So either an increase or a decrease in 14C activity is likely to be driven by external factors.

Response: We admit that observed variability of 14C values in tree rings cannot be explained by tree physiological processes alone, however, and in return we cannot follow the argument that only external factors cause 14C variability. Tress are living organisms, they react to changes of their environment in many ways. For example, they modify the stomatal aperture of their leaves/needle in response to moisture availability or CO2 concentration of the atmosphere. Related to this isotopic fractionation occurs that can lead to very individual or site specific 13C and 14C contents in tree organic matter. Changes to MS: We discuss the external factors in the revised MS. However, we do not see striking arguments why our suggestion to consider tree physiological effects to some extent should be discarded.

Using a numerical exercise and autocorrelation analysis of d13C data Slotta et al. estimate that approx 85% of fast cycling carbon pools and 15% of slow-cycling carbon pools are contributing to the lower values of the bomb peak. If this explanation is true, then this effect should have been manifested in the entire record the authors have re- ported, which is not apparent from their results. Rather, the authors admit that the baobab F14C values for 1945, 1952-1954, 1956 and 1957 are indeed higher than the calculated range. This observation casts doubt in their explanation of the old carbon turn over mechanism in explaining the negative excursion of 14C activity during the bomb peak period.

Response: As mentioned in response to referee#1 above, the unexpected significant mismatch between the baobab F14C values and the post-bomb atmospheric curve NH3 prompted for some reasonable interpretation. Hence, we suggested and still suggest that it is from a mixed carbon pool in conjunction with the extraordinary high content and longevity of parenchyma tissue relative to short lived fibre tissue that constitute the tree rings of baobabs. With the simple model considerations presented here we do not intend to precisely describe or capture temporal changes of carbohydrate pools in baobab trees. We can show, however, that the F14C values of baobab tree-ring cellulose can better be explained when assuming a mixing of pools of carbohydrate precursors of cellulose instead of assuming direct transfer to atmospheric carbon into organic matter of tree rings. Changes in the manuscript: We rewrote the MS to stress the uncertainties involved with our interpretation.

There may be another explanation of lower 14C activity in this region. Cember (1989) analyzed coral 14C from across the Red Sea to estimate the gas exchange rate. Cem- ber observed a very high air-sea exchange process over the Red Sea region. If this process is also operative in this region which is not very far from the Red Sea, then  a viable explanation of anomalous 14C activity in the atmosphere may be provided.

Response: Thank you very much for this comment. Changes in the manuscript: We considered this aspect and came to the conclusion that air-sea exchange processes cannot be responsible for a 8.8% dilution of atmospheric $^{14}$C.

2.4 Radiocarbon dating The analytical description provided by the authors is not up to the mark. For example, radiocarbon dating requires 13C correction and age correction; there is no mention of whether such kind of corrections has been done. The reporting of 14C activity, especially in the case of sequential samples (tree ring, corals) is typically done in cap delta notation (D14C), but the authors have preferred normalized activity.

Response: By utilizing F14C we follow the suggestion of Reimer, P. J., Brown, T. A. and Reimer, R. W. (2004). "Discussion: Reporting and calibration of post-bomb C-14 data." Radiocarbon 46(3): 1299-1304. In this paper the advantages of using F14C in studies of the bomb peak period are well described.

For comparison purposes with the published records, the authors are suggested to report the 14C activity in cap delta notation. Finally, the error in 14C measurement should be mentioned in terms of cap delta as well as the corresponding temporal value.

Changes in the manuscript: We are not inclined to make changes to our nomenclature (cf. above). However, all data is available for download at Pangaea.de. Any interested party can do recalculation from these data.

Minor issues: Line 92: The rainfall amount and its isotopes usually show a weak inverse correlation. Pls, provide reference for evidence of "strong" correlation.
Done!

Line 141: very heavy rainfall in a single day producing high negative d18O "due to amount effect" is not technically right. Many studies (Lawrence and Gedzelman, 1996; Gedzelman et al., 2003; Lawrence et al. 1998; Chakraborty et al., 2016; Xu et al., 2019), showed that extreme precipitation events such as cyclonic activities produced very low values of precipitation d18O.

Response: Thank you very much for this comment. Changes  in the manuscript: We have corrected this in the revised MS.

Line 181: pls provide a zoomed figure for the 1962-63 record of bomb 14C. Line 199: What are the reference materials used?

Response: Thank you very much for this comment. Figure 3B provide some more detail, highlighting the $^{14}$C spread of different species and sites. More detail is not of value, because our baobab data cannot be used to refine the atmospheric $^{14}$C record.

Line 202: Please provide the permil sign after 0.15 and 0.25. Line 217: 'weakening' should be "weaken". Response: Thank you very much for this comment. Changes in the manuscript: Done!

Line 251-252: How the interpretation of the F14C data was confirmed by visual and sta- tistical comparison of the TRW chronology with precipitation data should be explained in detail.

Response: Thank you very much for this comment. We are more specific in the

revised MS (COFECHA has been used).
Line 255: 'shallow' should be replaced by "gentle". Done!
Line 268: 'radiocarbon' should be followed by "analysis".
Done!
Line 275: 'considerably declining' meaning is not clear.
We are more specific in the revised MS.
Line 280: What is the physical basis of getting a strong correlation between d18O and RWI? Also mentioned in Line 334. Please provide the value of correlation and state the sample number.
Response: Thank you very much for this comment. We made changes to the MS in order to be more specific; correlation coefficients are provided in tables and figures
Line 309: the lag between cyclonic events and the corresponding d18Omin should be provided on a monthly time scale.
Response: Unfortunately, no precise date can be given for the timing of d18Omin. The trees do record the effects of cyclonic rainfall, but we cannot extract information on exact time lag.
Line 355: 'extend' should be replaced by "extent". Done!
Line 494: "evaporative enrichment in 18O...".Please provide supportive evidence of enhanced soil evaporation, say by means of observed or reanalysis data in support of this speculation.
Changes in the manuscript: Thanks, corresponding papers are cited in the revised MS.
Line 505: "Vapor pressure deficit ...18O enrichment in leaf water", and "lower stomatal conductance...13C discrimination to decline" require supporting literature. For instance: Barbour MM (2007) Stable oxygen isotope composition of plant tissue: a review. Funct Plant Biol 34 (2):83-94. doi:https://doi.org/10.1071/fp06228
Changes in the manuscript: Thanks, a highly valuable paper by Treydte et al 2014 is now cited in the revised MS.
Line 514: the authors argue that the decline in d18O...might be due to the previous year's October precipitation. If so, then d18O is also expected to be auto-correlated.
Response: We changed the MS accordingly.
Line 524: likely in 'would have likely..." should be deleted.
Changes in the manuscript: Thanks, we have corrected this in the revised MS.
References: Cember 1989 Bom radiocarbon in the Red Sea: a medium scales gas exchange ex- periment. JGR Ocean 94:2111-2123. Lawrence, J. L. & Gedzelman, S. D. Low stable isotope ratios of tropical cyclone rains. Geophys. Res. Lett. 23, 527–530 (1996). Gedzelman, S., Lawrence, J., Gamache, J., Black, M., Hindman, E., Black, R., Dunion, J., Willoughby, H., Zhang, X., 2003. Probing hurricanes with stable isotopes of rain and water vapor. Mon. Weather. Rev. 131 (6), 1112–1127. Hua, Q., Barbetti, M., Zoppi, U., Chapman, D. M., and Thomson, B. 2003 Bomb Ra- diocarbon in Tree Rings from Northern New South Wales, Australia: Implications for Dendrochronology, Atmospheric Transport, and Air-Sea Exchange of CO2, Radiocar- bon, 45, 431-447. Xu et al. 2019 Stable isotope ratios of typhoon rains in Fuzhou, Southeast China, during 2013–2017. Kikata Y, Yonenobu H, Morishita F, Hattori Y. 1992. 14C concentrations in tree stems. Bulletin of the Nagoya University Furukawa Museum 8:41–6. In Japanese.
Changes in the manuscript: Thanks, corresponding papers are now cited in the revised MS.

---

## Author Response (AR2)

Dear Dr. Mazumdar, dear Dr. Chakraborty,

We gratefully acknowledge the efforts of Dr. Chakraborty for re-evaluating our manuscript. In accordance with his advice, we have improved the language and corrected mistakes with support from a native speaker. Furthermore, we improved the quality of Fig. 7.

We do hope that the revised MS can be accepted now.

Sincerely,

Gerhard Helle